# Score Distillation Beyond Acceleration: Generative Modeling from Corrupted Data

**Yasi Zhang**[*,1], **Tianyu Chen**[*,2,3,‡], **Zhendong Wang**[3], **Ying Nian Wu**[1],
**Mingyuan Zhou**[†,2,3], **Oscar Leong**[†,1]
[*]Equal contribution     [†]Equal advising
[1]University of California, Los Angeles     [2]University of Texas at Austin
[3]Microsoft AI Superintelligence     [‡] Work done during an internship in Microsoft
Correspondence to `yasminzhang@ucla.edu`, `tianyuchen@utexas.edu`

## Abstract

Learning generative models directly from corrupted observations is a long-standing challenge across natural and scientific domains. We introduce *Restoration Score Distillation (RSD)*, a unified framework for learning high-fidelity, one-step generative models using **only** degraded data of the form $y = \mathcal{A}(x) + \sigma\varepsilon, x \sim p_X, \varepsilon \sim \mathcal{N}(0, I_m)$, where the mapping $\mathcal{A}$ may be the identity or a non-invertible corruption operator (e.g., blur, masking, subsampling, Fourier acquisition). RSD first pretrains a *corruption-aware diffusion teacher* on the observed measurements, then *distills* it into an efficient one-step generator whose samples are statistically closer to the clean distribution $p_X$. The framework subsumes identity corruption (denoising task) as a special case of our general formulation.

Empirically, RSD consistently reduces Fréchet Inception Distance (FID) relative to corruption-aware diffusion teachers across noisy generation (CIFAR-10, FFHQ, CELEBA-HQ, AFHQ-V2), image restoration (Gaussian deblurring, random inpainting, super-resolution, and mixtures with additive noise), and multi-coil MRI—*without access to any clean images*. The distilled generator inherits one-step sampling efficiency, yielding up to $30\times$ speedups over multi-step diffusion while surpassing the teachers after substantially fewer training iterations. These results establish score distillation as a practical tool for generative modeling from corrupted data, *not merely for acceleration*. We provide theoretical support for the use of distillation in enhancing generation quality in the Appendix. The code is available at `https://github.com/TianyuCodings/RSD`.

## 1 Introduction

Learning from corrupted data is central to many scientific and engineering domains where clean observations are scarce or costly, including astronomy Roddier (1988); Lin et al. (2024), medical imaging Reed et al. (2021); Jalal et al. (2021), and seismology Nolet (2008); Rawlinson et al. (2014). For instance, fully sampled MRI acquisitions are time-consuming and uncomfortable for patients Knoll et al. (2020); Zbontar et al. (2018), motivating methods that recover the structure of the underlying clean distribution from corrupted measurements alone.

**Problem Statement.** We study *generative modeling from corrupted observations*. Let $x \in \mathbb{R}^d$ be drawn from an unknown clean distribution $p_X$. We observe only

$$y = \mathcal{A}(x) + \sigma\epsilon, \qquad \epsilon \sim \mathcal{N}(0, I_m), \tag{1}$$

where $\mathcal{A} : \mathbb{R}^d \to \mathbb{R}^m$ is a (known) non-invertible corruption operator and $\sigma$ is the noise level. The operator may be identity (i.e., denoising), a deterministic linear map (blur, downsampling), a random mask (inpainting), or a Fourier-domain undersampling pattern (MRI). Our goal is to learn a generator whose samples follow $p_X$ using only a dataset of $N$ corrupted datapoints $\{y^{(i)}\}_{i=1}^N$ of the form Eq 1.

**Background and limitations.** Diffusion models Sohl-Dickstein et al. (2015); Ho et al. (2020), also known as score-based generative models Song and Ermon (2019); Song et al. (2021b), achieve

Figure 1: Overview of *Restoration Score Distillation* (RSD) and some qualitative results across diverse operators: Gaussian deblurring, random inpainting, and super-resolution. Additional examples appear in Appendix A and K.

state-of-the-art results in high-dimensional image synthesis Dhariwal and Nichol (2021); Ho et al. (2022); Ramesh et al. (2022); Rombach et al. (2022); Peebles and Xie (2023); Zheng et al. (2024); Zhang et al. (2025); Chang et al. (2025); Chen et al. (2025). When only measurements are available, *corruption-aware* training adapts diffusion objectives to the forward operator: Ambient Diffusion for masking Daras et al. (2023b), Ambient Tweedie for additive noise Daras et al. (2025), and Fourier-space variants for MRI Aali et al. (2025). These methods, however, inherit the sampling cost of multi-step reverse processes. EM-Diffusion Bai et al. (2024) offers broader operator coverage but requires a few clean images, is computationally expensive, and can struggle under severe corruption.

**Distillation beyond acceleration.** Score distillation transfers a pretrained diffusion teacher into a one-step generator while largely preserving fidelity (Poole et al., 2022; Wang et al., 2024b; Luo et al., 2024; Yin et al., 2024b; Zhou et al., 2024; Xie et al., 2024; Xu et al., 2025; Yin et al., 2024a). Recent reports indicate that distilled generators can even outperform their diffusion teachers (Zhou et al., 2024), though gains under clean-data training are typically modest. In contrast, under *corrupted-data–only* training, we observe substantially larger improvements over the teacher (Sec. 4), underscoring distillation's particular advantage in challenging regimes.

**Restoration Score Distillation (RSD).** We introduce *RSD*, a unified framework for learning high-fidelity one-step generators directly and only from corrupted observations, with denoising ($\mathcal{A}{=}I$) as a special case. RSD proceeds in two stages: (i) *corruption-aware diffusion pretraining*, where a teacher is trained on measurements using an objective matched to the forward operator $\mathcal{A}$ (Sec. 3.1); and (ii) *score distillation*, which transfers the teacher into a single-step generator while explicitly *respecting the measurement operator* during training (Sec. 3.2). Concretely, we synthesize measurements by applying the same corruption pipeline to generator outputs and then align the induced generator scores with the teacher's scores under a divergence (e.g., Fisher or KL). The distillation phase, which includes a corruption-respecting procedure, consistently improves generation quality upon the diffusion teacher across both denoising and more general operators (Fig. 1).

**Contributions.** Our work makes three primary contributions. **Unified framework for diverse corruptions:** We propose *RSD*, a unified approach that learns generators directly from diverse corrupted measurements $y = \mathcal{A}(x) + \sigma\epsilon$. This formulation encompasses denoising ($\mathcal{A} = I$) as well as more general operators, including blur, downsampling, random masking, and Fourier undersampling, under both noisy and noiseless regimes, and achieves state-of-the-art performance across these settings. **Modular training:** Our training pipeline is organized into two phases. Phase I accommodates a variety of corruption-aware techniques, such as standard diffusion, diffusion for denoising, random inpainting, and masked Fourier-space (F.S.) transformations. Phase II distills the teacher model into

Table 1: Summary of corruption-aware diffusion objectives used for pretraining. Our framework can be seamlessly integrated with existing advanced corruption-aware diffusion objectives.

| Suitable Scenario | Algorithm | Operator | Domain | Pretrain Objective | Notation |
|---|---|---|---|---|---|
| Noiseless Corruption | Alg(2) | $\mathcal{A}(x)$ | Image | $\mathcal{L}_{\text{SD}} = \mathbb{E}\big[\lambda(t)\|f_\phi(y + \sigma_t\varepsilon, t) - y\|_2^2\big]$ | $y$: corrupted data |
| Noisy Corruption | Alg(3) | $\mathcal{A}(x) + \sigma\epsilon$ | Image | $\mathcal{L}_{\text{N}} = \mathbb{E}\big[\|\frac{\tilde{\sigma}_t^2 - \sigma^2}{\tilde{\sigma}_t^2}f_\phi(y_t, t) + \frac{\sigma^2}{\tilde{\sigma}_t^2}y_t - y\|_2^2\big]$ | $\sigma$: data noise level, $\tilde{\sigma}_t = \max\{\sigma_t, \sigma\}$ |
| Random Inpainting | Alg(5) | $Mx$ | Image | $\mathcal{L}_{\text{RI}} = \mathbb{E}\big[\|M(f_\phi(\tilde{M}, \tilde{M}y_t, t) - y)\|_2^2\big]$ | $\tilde{M}$: further corrupted mask $M$ |
| Masked F.S. | Alg(6) | $M\mathcal{F}x$ | Fourier | $\mathcal{L}_{\text{FS}} = \mathbb{E}\big[\|\mathcal{A}f_\phi(\tilde{M}, \tilde{y}_t, t) - y\|_2^2\big]$ | $\tilde{y}_t$: further corrupted $y_t$, $\mathcal{A}$ defined in Appx.H.4 |

a one-step generator while retaining the corruption pipeline used during training. This modularity makes it straightforward to plug in new forward operators or training objectives. We also provide theoretical analysis that explains why the distillation phase can enhance generation quality. **Extensive experiments:** We conduct comprehensive evaluations on natural-image benchmarks (CIFAR-10, CelebA-HQ, FFHQ, AFHQ-v2), restoration tasks (denoising with $\sigma \in \{0.1, 0.2, 0.4\}$, Gaussian deblurring, random inpainting with $p \in \{0.6, 0.8, 0.9\}$, and $2\times$ super-resolution), and multi-coil MRI with acceleration factors $R \in \{4, 6, 8\}$. Across all settings, RSD consistently improves FID over corruption-aware diffusion teachers while offering substantial speedups via one-step generation (Sec. 4). Additional ablations (Sec. 4.5) on unknown corruption types and different data size further underscore the robustness of our framework. Moreover, we demonstrate that the learned clean-image prior (the generator) can be directly leveraged for downstream conditional inverse problems, achieving good performance (Sec. 4.7).

## 2 BACKGROUND

### 2.1 DIFFUSION MODELS

Diffusion models Sohl-Dickstein et al. (2015); Ho et al. (2020), also known as score-based generative models Song and Ermon (2019); Song et al. (2021b), consist of a forward process that gradually perturbs data with noise and a reverse process that denoises this signal to recover the data distribution $p_X(x)$. Specifically, the forward process defines a family of conditional distributions over noise levels $t \in (0, 1]$, given by $q_t(x_t \mid x) = \mathcal{N}(\alpha_t x, \sigma_t^2 I)$, with marginals $q_t(x_t)$. We adopt a variance-exploding (VE) process Song and Ermon (2019) by setting $\alpha_t = 1$, yielding the simple form $x_t = x + \sigma_t\varepsilon$, where $\varepsilon \sim \mathcal{N}(0, I_d)$. To model the reverse denoising process, one typically trains a time-dependent denoising autoencoder (DAE) $f_\phi(\cdot, t) : \mathbb{R}^d \times [0, 1] \to \mathbb{R}^d$ Vincent (2011), parameterized by a neural network, to approximate the posterior mean $\mathbb{E}[x \mid x_t]$. This is achieved by minimizing the following standard diffusion loss:

$$\mathcal{L}_{\text{SD}}(\phi; \{x^{(i)}\}_{i=1}^N) := \mathbb{E}_{x,\sigma_t,\varepsilon}\big[\lambda(t)\|f_\phi(x_t, t) - x\|_2^2\big] \tag{2}$$

where $x_t = x + \sigma_t\varepsilon$ and $\{x^{(i)}\}_{i=1}^N$ denotes the dataset. One can also apply the loss to the corrupted data $\{y^{(i)}\}_{i=1}^N$ to directly learn the distribution $p_Y(y)$. For clarity, we omit the diffusion training schedule $\lambda(t)$, $p(\sigma_t)$ and the noise term $\varepsilon \sim \mathcal{N}(0, I_m)$; full details are deferred to Appendix H.1, Alg 2. The corresponding objectives are summarized in Table 1, with additional discussion below.

### 2.2 DIFFUSION MODELS FOR CORRUPTIONS

Recent advances have extended diffusion models to address diverse forms of data corruption. We present several variants of the diffusion loss tailored to different corruption settings, including objectives for noisy data, random inpainting in the image domain and Fourier domain.

**Diffusion for Noisy Corruptions.** Daras et al. (2025) generalizes score matching to noisy observations $y = x + \sigma\varepsilon$, $\varepsilon \sim \mathcal{N}(0, I)$. Here, $\sigma > 0$ is a known noise level in the measurements. One can incorporate the known corruption by minimizing the following diffusion for noisy corruptions

$$\mathcal{L}_{\text{N}}(\phi; \{y^{(i)}\}_{i=1}^N) = \mathbb{E}_{y,\tilde{\sigma}_t,\epsilon}\left[\lambda(t)\left\|\frac{\tilde{\sigma}_t^2 - \sigma^2}{\tilde{\sigma}_t^2}f_\phi(y_t, t) + \frac{\sigma^2}{\tilde{\sigma}_t^2}y_t - y\right\|_2^2\right], \tag{3}$$

where $\tilde{\sigma}_t = \max\{\sigma_t, \sigma\}$, $y_t = y + \tilde{\sigma}_t$, and $p(\sigma_t)$ and $\lambda(t)$ arise from the diffusion training schedule. This formula models the distribution of the $x_t := x + \sigma_t\varepsilon$ where $\sigma_t \geq \sigma$. Full details are deferred to Appendix H.2, the training algorithm in Algorithm 3 and the sampling algorithm in Algorithm 4.

**Diffusion for Image-Space Random Masked Corruptions.** Daras et al. (2023b) learn $p_X(x)$ from randomly inpainted measurements by incorporating mask into the diffusion objective. Given observations $y = Mx$ for a binary mask $M$, the teacher is trained with a random-inpainting loss

$$\mathcal{L}_{\mathrm{RI}}(\phi; \{y^{(i)}\}_{i=1}^N) = \mathbb{E}_{(y,M),\,\tilde{M},\,\sigma_t,\,\varepsilon}\left[\left\| M\left(f_\phi(\tilde{M},\,\tilde{M}y_t,\,t) - y\right)\right\|_2^2\right], \tag{4}$$

where $y_t = y + \sigma_t\varepsilon$ is from diffusion schedule, $\varepsilon \sim \mathcal{N}(0, I)$, and $\tilde{M}$ is a secondary mask that further erases pixels based on $M$. Details of the diffusion schedule $\lambda(t)$, $p(\sigma_t)$, and the full training procedure are provided in Appendix H.3, Algorithm 5.

**Diffusion for Fourier-Space Random Masked Corruptions.** (Aali et al., 2025) extends diffusion models to handle frequency-domain measurements with random masking, which are used in scientific imaging (e.g., MRI). Specifically, observations take the form $y = M\mathcal{F}x$, where $\mathcal{F}$ denotes the Fourier transform operator and $M$ is a sampling mask. Formal definitions of the measurement process and the corresponding training algorithm are provided in Appendix H.4 and Algorithm 6.

These specialized frameworks enable generative modeling from realistic scenarios such as noisy, missing data, or frequency-domain degradation. All objective functions are summarized in Table 1.

### 2.3 SCORE DISTILLATION FOR GENERATIVE MODELING

Score distillation compresses multi-step diffusion models into efficient one-step generators. Originally proposed for text-to-3D generation Poole et al. (2022); Wang et al. (2024b) and later extended to image synthesis Luo et al. (2024); Yin et al. (2024b); Zhou et al. (2024); Xie et al. (2024), it transfers knowledge from a pretrained diffusion teacher $f_\phi$ to a generator $G_\theta : \mathbb{R}^d \to \mathbb{R}^d$. To bridge the two, a fake diffusion model $f_\psi$ approximates the distribution induced by $G_\theta(\cdot)$ across diffusion noise levels. Training encourages consistency between $f_\psi$ and $f_\phi$ over time:

$$\mathcal{L}_{\mathrm{distill}}(\theta) = \mathbb{E}_{\sigma_t,\,z\sim\mathcal{N}(0,I_d)}\mathbb{E}_{x_\theta=G_\theta(z)}\left[\|f_\phi(x_t,t) - f_\psi(x_t,t)\|_2^2\right], \tag{5}$$

where $x_t = x_\theta + \sigma_t\epsilon$ and the loss corresponds to Fisher divergence, following SiD Zhou et al. (2024). This divergence of true and fake distribution can be replaced by KL divergence; ablation results are reported in Section 4.5 and Appendix D.5. Intuitively, the fixed teacher $f_\phi$ represents the true data distribution, while $f_\psi$ captures the generator's induced distribution. Updating $G_\theta$ to minimize $\mathcal{L}_{\mathrm{distill}}(\theta)$ aligns generator samples with the true data. Notably, distilled generators can even outperform their teachers: for instance, Zhou et al. (2024) reports that on FFHQ the teacher achieves an FID of 2.39, whereas its distilled generator attains 1.55. In our setting, we find that such improvements are further amplified when distillation is performed under corrupted-data training. Details of the distillation training schedules are deferred to Appendix I.

## 3 RESTORATION SCORE DISTILLATION

**Problem Statement.** Suppose we are given a finite corrupted dataset of size $N$, denoted by $\{y^{(i)}\}_{i=1}^N$. Each corrupted observation is generated as $y^{(i)} = \mathcal{A}(x^{(i)}) + \sigma\varepsilon^{(i)}$, where $\sigma$ is a known noise level (with extensions available for the unknown-$\sigma$ setting, see Section 4.5), and $\varepsilon^{(i)} \overset{\text{i.i.d.}}{\sim} \mathcal{N}(0, I_m)$. Crucially, the clean data $\{x^{(i)}\}_{i=1}^N$ is never accessible. In certain scenarios, such as random inpainting, the corruption operator $\mathcal{A}$ may vary across samples, drawn from a common distribution. In this case, each corrupted observation takes the form $y^{(i)} = \mathcal{A}^{(i)}(x^{(i)}) + \sigma\varepsilon^{(i)}$.

RSD is a two-phase framework for learning generative models solely from corrupted observations. It decouples the process into: (1) a flexible corruption-aware diffusion pretraining stage, and (2) a distillation stage that compresses the pretrained model into a single-step generator and further improves generation quality. The overall procedure is summarized in Algorithm 1. We summarize different types of pretrained diffusion models in Table 1, and then describe how they can be seamlessly integrated into RSD to further enhance performance. Notably, even when a suitable corruption-aware pretrained model is unavailable, employing a standard diffusion model (Tab 1) to learn the corrupted data distribution still yields substantial improvements after distillation (Tab 4).

## 3.1 PHASE I: FLEXIBLE CORRUPTION-AWARE PRETRAINING

The first phase trains a teacher diffusion model $f_\phi(\cdot, t)$ directly on corrupted observations $y = \mathcal{A}(x) + \sigma\epsilon$, where $\mathcal{A}$ is a potentially non-invertible corruption operator. A straightforward approach is to train on corrupted data $y$ using the standard diffusion loss (Eq. 2). In practice, we find that this naive strategy already yields strong performance. To further improve performance across diverse corruption operators, we incorporate existing corruption-aware diffusion training methods, as summarized in Table 1. Our choice of pretraining objectives for different corruption scenarios is as follows: 1) Standard Diffusion is effective under **noiseless corruptions**, directly modeling the distribution of $y = \mathcal{A}(x)$. 2) **Noisy Corruption** Daras et al. (2025) targets additive noise settings, learning the distribution of $\mathcal{A}(x)$ from noisy observations $y = \mathcal{A}(x) + \sigma\epsilon$, with denoising as a special case when $\mathcal{A} = I$. 3) **Random Inpainting** Daras et al. (2023b) addresses learning from partial observations $Mx$, where $M$ is a random inpainting mask, and learns the distribution of $x$. 4) **Masked F.S** Aali et al. (2025) extends inpainting to Fourier-domain corruptions, well-suited for frequency-based degradation such as accelerated MRI, learning from masked Fourier observations $M\mathcal{F}x$.

## 3.2 PHASE II: ONE-STEP GENERATOR DISTILLATION

After pretraining, we distill the teacher diffusion model $f_\phi(\cdot, t)$ into a one-step generator $G_\theta$ that maps latent noise $z \sim \mathcal{N}(0, I_d)$ directly to clean samples. To facilitate this distillation, we introduce an auxiliary fake diffusion model $f_\psi(\cdot, t)$, initialized from $f_\phi$ and trained on corrupted output of $G_\theta(z)$, which is also initialized from $f_\phi$. Note that this is common practice in the distillation literature to facilitate and stabilize training. In this case, $G_\theta, f_\phi, f_\psi$ share the same network structure; see Appendix I for further discussion. By encouraging $f_\psi$ to match the teacher's time-dependent dynamics (e.g., score fields parameterized by $f_\phi$), we align the behavior of $G_\theta$ with the data distribution learned by $f_\phi$, thereby narrowing the gap between the generated and clean data distributions.

We adopt the SiD Zhou et al. (2024) framework for distillation and optimize the distillation loss $\mathcal{L}_{\text{distill}}(\theta)$ (Eq. 5). An ablation of loss choices is provided in Sec. 4.5. Beyond standard score distillation, we further corrupt the generated samples $x_g = G_\theta(z)$ into $\tilde{y}$ using the same corruption pipeline as the training data (Algorithm 1, Line 6). These corrupted samples are then used to train a fake diffusion model $f_\psi(\cdot, t)$. In Algorithm 1, Lines 2 and 7 allow any of the pretraining objectives summarized in Table 1 (or new objectives tailored to specific corruptions), but *the same* objective must be used for both lines; mixing objectives destabilizes training and can lead to divergence. To complement the empirical results in Section 4, Appendix F offers a theoretical analysis establishing conditions under which distillation yields improved sample quality.

---

**Algorithm 1** Restoration Score Distillation (RSD)

---

1: **procedure** RESTORATION SCORE DISTILLATION($\{y^{(i)}\}_{i=1}^N$)
2:     $f_\phi \leftarrow$ DIFFUSIONTRAINING($\{y^{(i)}\}_{i=1}^N$, Obj)                    ▷ Diffusion training with Obj ∈ Tab. 1
3:     Initialize $f_\psi \leftarrow f_\phi, G_\theta \leftarrow f_\phi$
4:     **for** $j = 1, \ldots, K$ **do**
5:         $x_g \leftarrow G_\theta(z), \quad z \sim \mathcal{N}(0, I_d)$                    ▷ Generate fake clean images
6:         $\tilde{y} = \mathcal{A}(\text{stopgrad}(x_g)) + \sigma\varepsilon, \varepsilon \sim \mathcal{N}(0, I_d)$     ▷ Corrupt $x_g$ same way as observation
7:         *# If corruption contains noise, choose an Obj for $f_\phi$, $f_\psi$ that is denoising-aware*
8:         $f_\psi \leftarrow$ DIFFUSIONTRAINING($\{\tilde{y}^{(i)}\}_{i=1}^N$, Obj)                    ▷ **Same Obj** as above
9:         *# Do NOT inject additional noise during distillation when updating $\theta$*
10:        $y_\theta \leftarrow \mathcal{A}(G_\theta(z)), \quad z \sim \mathcal{N}(0, I_d)$
11:        Update $\theta$ by distillation loss $\mathcal{L}_{\text{distill}}(\theta)$ with generated $x_\theta = y_\theta$                    ▷ Eq. 5
12:     **end for**
13: **end procedure**

---

## 4 EXPERIMENTS

We conduct comprehensive experiments to evaluate the effectiveness, flexibility, and generality of our framework, RSD, across both natural and scientific imaging domains. We first demonstrate its performance through a 2D toy task and qualitative visualizations, providing intuition into the

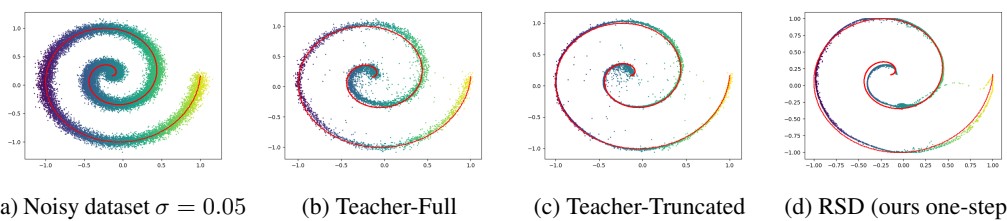

| (a) Noisy dataset $\sigma = 0.05$ | (b) Teacher-Full | (c) Teacher-Truncated | (d) RSD (ours one-step) |

Figure 2: **A toy example of learning from a noisy dataset with $\sigma = 0.05$.**

underlying mechanism. Then we consider four challenging image restoration tasks—denoising, Gaussian deblurring, random inpainting, and super-resolution—on CIFAR-10 ($32 \times 32$), FFHQ ($64 \times 64$), and CelebA-HQ ($64 \times 64$) Liu et al. (2015); Karras et al. (2017), under both noiseless ($\sigma = 0.0$) and noisy ($\sigma = 0.2$) regimes. We further show that RSD can be distilled directly from pretrained teacher models trained with corruption-aware objectives, including diffusion for denoising Daras et al. (2025), diffusion for random inpainting Daras et al. (2023a), and diffusion for Fourier-space inpainting Aali et al. (2025), highlighting the framework's modularity and flexibility. Sec. 4.2 benchmarks denoising across multiple datasets and noise levels. Sec. 4.3 evaluates general corruption operators in both noiseless and noisy settings, including a sweep over random-inpainting missing rates $p \in \{0.6, 0.8, 0.9\}$.

To assess real-world applicability, we evaluate RSD on multi-coil Magnetic Resonance Imaging (MRI) using the FastMRI dataset Zbontar et al. (2018); Knoll et al. (2020), a setting where fully sampled ground truth is often unavailable. Across tasks and corruption regimes, RSD consistently surpasses teacher diffusion models in generative quality as measured by FID Heusel et al. (2017) without access to clean data (Sec. 4.4).

Finally, RSD exhibits strong *distillation–data efficiency* (Sec. 4.6)—it quickly surpasses the teacher—and offers substantial inference-time gains from its one-step design (up to $30\times$; see Sec. 4.6). Our ablations (Sec. 4.5) show that the framework accommodates unknown noise levels $\sigma$ and is robust with respect to the training-data size; we also examine alternative distillation losses in an ablation study. Separately, Sec. 4.7 demonstrates that the trained generator achieves superior performance on downstream conditional inverse problems, indicating broad applicability.

## 4.1 A 2D TOY EXAMPLE

In Fig. 2, the observation is a noisy spiral dataset with $\sigma = 0.05$. The core issue with teacher diffusion models trained on noisy datasets, i.e., (b) Teacher-Full and (c) Teacher-Truncated (Daras et al. (2024), Algorithm 4), is that they force the approximating distribution to spread its probability mass across all regions, making the learned density overly diffuse. In contrast, (d) RSD excels at denoising the original dataset, producing a narrow, concentrated, and sharp approximation. This effect occurs *without* providing the clean data itself during training.

## 4.2 DENOISING AS A SPECIAL CASE OF RESTORATION SCORE DISTILLATION

We begin with the denoising special case $y = x + \sigma\epsilon$ where the forward operator $\mathcal{A} = I$. We compare RSD against three groups of baselines (Table 2). **(1) Teacher diffusion models.** A teacher trained with the noisy corruption loss in Eq. 3 serves as a strong generative baseline. We adopt the two sampling schedules of Daras et al. (2024)—**Teacher-Full** and **Teacher-Truncated** (Algorithm 4)—and additionally evaluate the **Teacher-consistency** variant Daras et al. (2025). There may exist better sampling schedules for denoising; however, to the best of our knowledge, the schedules we adopt achieve strongest

Table 3: **RSD vs. teachers on CIFAR-10/FFHQ/AFHQ-v2.** RSD (distilled) consistently surpasses teacher diffusion (*Full, Truncated*) across datasets and noise levels. (FID).

| Methods Data noise | CIFAR-10 | | | FFHQ | AFHQ-v2 |
|---|---|---|---|---|---|
| | $\sigma=0.1$ | $\sigma=0.2$ | $\sigma=0.4$ | $\sigma=0.2$ | $\sigma=0.2$ |
| Observation | 73.74 | 127.22 | 205.52 | 110.83 | 51.51 |
| Teacher-Full | 25.55 | 60.73 | 124.28 | 41.52 | 17.93 |
| Teacher-Truncated | 7.55 | 12.21 | 22.12 | 14.67 | 9.82 |
| **RSD (Distilled)** | **3.98** | **4.77** | **21.63** | **6.29** | **5.42** |

Table 2: **Denoising results on CIFAR-10 and CelebA-HQ at** $\sigma = 0.2$. Rows with $\sigma$=0.0 are clean-data upper bounds. Few-shot methods use 50 clean images. Baseline numbers are from the original papers or Bai et al. (2024); Lu et al. (2025). The distilled student (RSD, one-step) improves over teacher models (Full/Truncated/Consistency) and surpasses all few-shot baselines.

| Methods | CIFAR-10 (32×32) | | | CelebA-HQ (64×64) | | |
|---|---|---|---|---|---|---|
| | $\sigma$ | Type | FID | $\sigma$ | Type | FID |
| DDPM Ho et al. (2020) | 0.0 | Full-Shot | 4.04 | 0.0 | Full-Shot | 3.26 |
| DDIM Song et al. (2021a) | 0.0 | Full-Shot | 4.16 | 0.0 | Full-Shot | 6.53 |
| EDM Karras et al. (2022) | 0.0 | Full-Shot | **1.97** | – | – | – |
| SURE-Score Aali et al. (2023) | 0.2 | Few-Shot | 132.61 | – | – | – |
| Ambient Diffusion Daras et al. (2023a) | 0.2 | Few-Shot | 114.13 | – | – | – |
| EM-Diffusion Bai et al. (2024) | 0.2 | Few-Shot | 86.47 | – | – | – |
| TweedieDiff Daras et al. (2024) | 0.2 | Few-Shot | 65.21 | 0.2 | Few-Shot | 58.52 |
| SFBD Lu et al. (2025) | 0.2 | Few-Shot | **13.53** | 0.2 | Few-Shot | **6.49** |
| TweedieDiff Daras et al. (2024) | 0.2 | Zero-Shot | 167.23 | 0.2 | Zero-Shot | 246.95 |
| Teacher-Full Daras et al. (2025) | 0.2 | Zero-Shot | 60.73 | 0.2 | Zero-Shot | 61.14 |
| Teacher-Truncated Daras et al. (2025) | 0.2 | Zero-Shot | 12.21 | 0.2 | Zero-Shot | 13.90 |
| Teacher-Consistency Daras et al. (2025) | 0.2 | Zero-Shot | 11.93 | 0.2 | Zero-Shot | 12.97 |
| **RSD (Ours, One-Step)** | 0.2 | Zero-Shot | **4.77** | 0.2 | Zero-Shot | **6.48** |

Table 4: **CelebA-HQ restoration under noiseless/noisy settings.** Baselines are taken from original papers when available, otherwise reproduced. EM-Diffusion uses 50 clean images for initialization, while RSD uses none. Best results are highlighted.

| Methods | $\sigma$ | Gaussian Deblurring | Random Inpainting ($p = 0.9$) | Super-Resolution (×2) |
|---|---|---|---|---|
| Observation | | 72.83 | 396.14 | 23.94 |
| Teacher Diffusion | 0.0 | 94.40 | 25.53 | 23.28 |
| EM-Diffusion *(Few-Shot)* | | 56.69 | 104.68 | 58.99 |
| **RSD (Ours, One-Step)** | | **31.90** | **16.86** | **12.99** |
| Observation | | 264.37 | 419.92 | 200.04 |
| Teacher Diffusion | 0.2 | 99.19 | 319.34 | 23.92 |
| EM-Diffusion *(Few-Shot)* | | **51.33** | 165.60 | 57.31 |
| **RSD (Ours, One-Step)** | | 76.98 | **79.48** | **22.00** |

performance reported in prior work. **(2) Few-shot methods. EM-Diffusion** Bai et al. (2024) alternates DPS-based reconstructions (E-step) with model refinement (M-step), while **SFBD** Lu et al. (2025) casts the problem as density deconvolution; both use 50 clean images for initialization. **(3) Clean-data diffusion (upper bound). DDPM** Ho et al. (2020), **DDIM** Song et al. (2021a), and **EDM** Karras et al. (2022) are trained on clean data ($\sigma$=0) and serve as upper bounds for any method trained purely on corrupted observations. Across CIFAR-10 and CelebA-HQ at $\sigma$=0.2, RSD (one-step) outperforms all zero-shot and few-shot baselines and improves over its teachers (Table. 2). For FFHQ and AFHQ-v2 where prior few-shot results are not reported, RSD also surpasses teacher models (Table 3). Finally, we sweep noise levels $\sigma \in \{0.1, 0.2, 0.4\}$ on CIFAR-10 and observe consistent gains over teachers. Quality examples are provided in Appendix A and K.

### 4.3 GENERAL CORRUPTIONS MIXED WITH NOISE

We now move beyond denoising to the general measurement model $y = \mathcal{A}(x) + \sigma\epsilon$, where $\mathcal{A}$ may be non-invertible. Using CelebA-HQ as a running example, we study both noiseless ($\sigma = 0.0$) and noisy ($\sigma = 0.2$) regimes and instantiate $\mathcal{A}$ as (i) Gaussian deblurring, (ii) random inpainting, and (iii) 2× super-resolution. Aggregate results across tasks and noise levels are reported in Table 4. For random inpainting, we further sweep the missing rate $p \in \{0.6, 0.8, 0.9\}$; see Table 5. Across settings, RSD—trained *without* clean images—consistently improves over its teacher diffusion models and is competitive with, or exceeds, few-shot methods that rely on clean initialization. Qualitative examples are provided in Appendix A and K.

### 4.4 BEYOND NATURAL IMAGES: MULTI-COIL MAGNETIC RESONANCE IMAGING (MRI)

Table 5: **CelebA-HQ random inpainting vs. missing rate** $p$**.** Teacher diffusion is trained as in Daras et al. (2023b).

| Method | $p = 0.6$ | $p = 0.8$ | $p = 0.9$ |
|---|---|---|---|
| Observation | 275.04 | 383.82 | 396.14 |
| Teacher Diffusion | 6.08 | 11.19 | 25.53 |
| **RSD** | **4.44** | **7.10** | **16.86** |

Table 6: **FID across acceleration levels** $R$ **in multi-coil MRI.** Teacher diffusion is trained as in Aali et al. (2024).

| Method | $R = 2$ | $R = 4$ | $R = 6$ | $R = 8$ |
|---|---|---|---|---|
| L1-EDM | 18.55 | 27.64 | 51.43 | 102.98 |
| Teacher Diffusion | 30.34 | 32.31 | 31.50 | 48.15 |
| **RSD** | **12.95** | **10.71** | **14.64** | **22.51** |

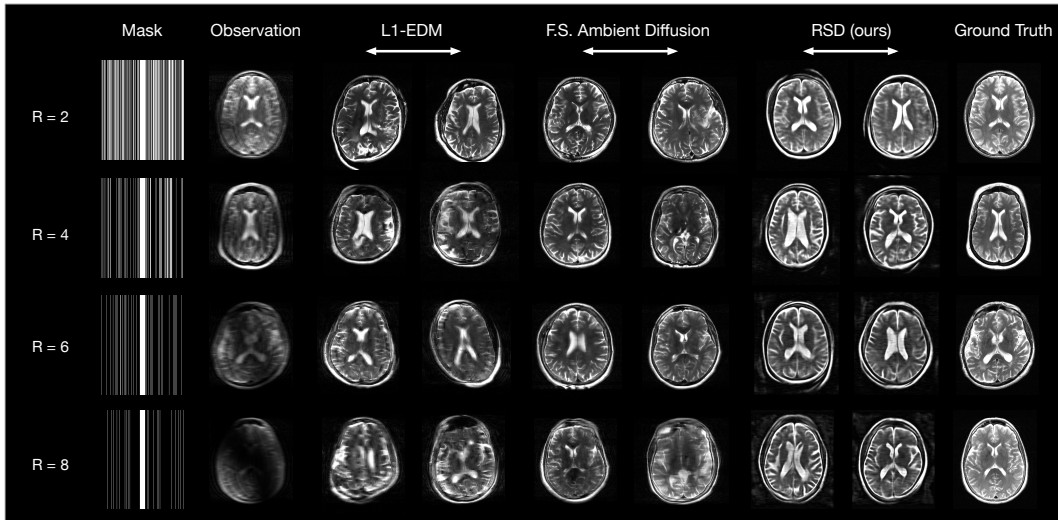

Figure 3: **Multi-coil MRI:** qualitative reconstructions from **RSD** across acceleration levels $R$. See Tab. 6 for corresponding FID trends.

We next apply RSD to a practical medical-imaging setting where fully sampled data are often unavailable due to time and cost constraints Knoll et al. (2020); Zbontar et al. (2018); Tibrewala et al. (2023); Desai et al. (2022). This case study demonstrates: (i) the flexibility of RSD to integrate with advanced corruption-aware diffusion techniques, (ii) robustness across acceleration factors $R$, and (iii) an extension from real- to complex-valued signals, $x \in \mathbb{C}^d$. Table 6 reports FID versus acceleration

Table 7: Comparison of RSD with teacher diffusion and baselines on CIFAR-10. Metrics for DDPM and Rectified Flow are from [34].

| Method | $\sigma$ | FID↓ | IS↑ | Prec.↑ | Rec.↑ | KID↓ |
|---|---|---|---|---|---|---|
| Teacher | 0.2 | 12.21 | 8.31 | 0.59 | 0.41 | .0059 |
| RSD | 0.4 | 21.63 | 7.93 | 0.53 | 0.38 | .0127 |
| RSD | 0.2 | 4.77 | 9.16 | **0.65** | 0.56 | .0025 |
| RSD | 0.1 | 3.98 | 9.34 | 0.64 | **0.57** | .0015 |
| DDPM | 0.0 | 3.21 | 9.46 | N/A | 0.57 | N/A |
| RF (ODE) | 0.0 | **2.58** | **9.60** | N/A | 0.57 | N/A |

$R \in \{2, 4, 6, 8\}$ comparing RSD, a teacher trained via **Fourier-Space Ambient Diffusion** Aali et al. (2025), and a baseline **L1-EDM** that trains EDM Karras et al. (2022) on L1–Wavelet reconstructions Lustig et al. (2007). Across all acceleration levels, RSD improves over the teacher; notably, it also outperforms L1-EDM at low acceleration ($R$=2), suggesting that distillation-based regularization is more effective than handcrafted L1 priors in the wavelet domain. Qualitative examples are shown in Fig. 3, with full algorithmic details in Appendix H.4 and I.

## 4.5 Ablations: Data Scale, Distillation Loss Choice, and More

**Data scale.** We next vary the training set size on CelebA-HQ to test data efficiency. RSD maintains strong performance even with 10% of the data and improves as more data become available, outperforming both Teacher-Full and Teacher-Truncated across settings. The results are shown in Table 8.

Table 8: **Data-size ablation on CelebA-HQ.**

| Data Size | Teacher-Full | Teacher-Trunc. | RSD |
|---|---|---|---|
| 10% | 62.25 | 14.36 | **10.53** |
| 50% | 56.09 | 17.19 | **9.76** |
| 100% | 61.14 | 13.90 | **6.48** |

**Comprehensive data quality metrics.** While FID is the primary metric for assessing data quality in image generation, we incorporate additional metrics for a more holistic evaluation, as detailed in Table 7. The results demonstrate that our distilled RSD generator achieves both superior density modeling (low FID) and robust mode coverage (high Recall), even when compared to generative models trained on clean datasets.

**Choice of distillation loss.** We compare several popular distillation *losses* used to compress diffusion teachers into one-step generators, including SDS Poole et al. (2022), DMD Yin et al. (2024b), and SiD Zhou et al. (2024). Unless otherwise noted, we use the *default* hyperparameters from the original papers (and public repos) without task-specific tuning. Under these settings, SiD consistently yields the strongest FID across our datasets, while SDS and DMD underperform. We report the full per-dataset and per-$\sigma$ breakdown in Table 18 (Appendix D.5). We emphasize that we did not perform a hyperparameter sweep; consequently, better-tuned configurations of SDS or DMD may close the gap to SiD. Empirically, we find SiD's default hyperparameters at different setting is a stable choice in our RSD pipeline.

More ablations can be found in the Sec. D of the Appendix.

## 4.6 SAMPLING AND TRAINING EFFICIENCY FOR DISTILLATION

After distillation, RSD attains markedly higher inference throughput than its diffusion teacher. On CIFAR-10 at $\sigma = 0.2$, producing 50,000 samples drops from **10 minutes** (teacher) to **20 seconds** (generator), a $\sim$**30$\times$** reduction in wall-clock time. Training is likewise efficient: the *Phase II* distillation stage surpasses the teacher's FID in roughly **4 hours**, compared to **48 hours** for *Phase I* teacher pretraining. Thus, once the teacher is available, running RSD adds only a minor computational budget yet yields substantial quality and speed gains. Figure 4 summarizes both sampling and training efficiency. All wall-clock times were measured on $8\times$ RTX A6000 (32 GB). Additional quantitative statistics and efficiency examples are provided in Appendix C and Table 12.

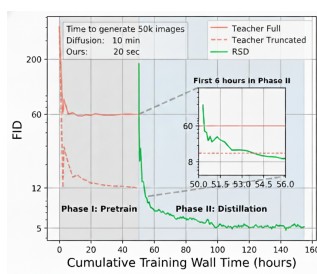

Figure 4: **Efficiency of RSD (CIFAR-10, $\sigma = 0.2$).**

## 4.7 AFTER DISTILLATION: SOLVING CONDITIONAL INVERSE PROBLEMS

Table 9: Conditional inverse problem results of denoising on CIFAR10 at $\sigma = 0.2$. Results for the baselines are taken from [5]. We follow [5] and sample 250 test images and compute the average PSNR and LPIPS.

| Method | Type | PSNR↑ | LPIPS↓ |
|---|---|---|---|
| Observations | | 18.05 | 0.047 |
| DPS w/ clean prior [12] | Full-Shot | 25.91 | 0.010 |
| SURE-Score [1] | Few-Shot | 22.42 | 0.138 |
| AmbientDiffusion [15] | Few-Shot | 21.37 | 0.033 |
| EM-Diffusion [5] | Few-Shot | 23.16 | 0.022 |
| Noise2Self [6] | Zero-Shot | 21.32 | 0.227 |
| RSD (ours) | Zero-Shot | **24.11** | **0.025** |

Because our framework yields a high-quality clean image generator—which naturally serves as a prior in conditional inverse solvers Chung et al. (2022); Zhang et al. (2024); Zhu et al. (2024); Zhang and Leong (2025); Hertrich et al. (2025)—a direct extension is to evaluate its utility on downstream inverse problems. We report a denoising task performance in Table 9, where RSD substantially outperforms prior methods and achieves performance comparable to few-shot approaches such as EM-Diffusion. In our implementation, we solve $\min_z \left\| \mathcal{A}(G_\theta(z)) - y \right\|_2^2$ for 1000 steps using Adam Kingma (2014) with a learning rate of 0.05. Exploring alternative strategies for inverse problems with one-step generators is an exciting direction. Additional results are provided in Appendix D.3.

## 4.8 MODEL SELECTION CRITERION UNDER CORRUPTION: PROXIMAL FID

Prior baselines Daras et al. (2023a); Bai et al. (2024) report FID scores yet do not specify how to perform *model selection* under corruption during training. To address this gap, we introduce *Proximal FID*, a model-selection metric tailored to corrupted-data regimes. Concretely, we generate 50k clean samples $\{x^{(i)}\}_{i=1}^{50k}$ from the current generator, corrupt them to match the training noise

and operator—yielding $\left\{\mathcal{A}\big(x^{(i)}\big) + \sigma\,\epsilon^{(i)}\right\}_{i=1}^{50\mathrm{k}}$, and compute FID against the corrupted training set $\{y^{(i)}\}_{i=1}^{n}$. As shown on CIFAR-10 in Fig. 5 and Tab. 10, Proximal FID tracks the true FID closely throughout distillation and attains near-optimal *true* FID across datasets. More results can be found in the Appendix. Taken together, these results support Proximal FID as a practical and reliable proxy for model selection when clean data are unavailable.

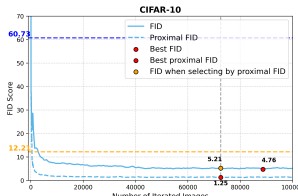

Figure 5: **CIFAR-10 (denoising, $\sigma = 0.2$):** Proximal FID closely tracks true FID and selects a near-optimal checkpoint.

Table 10: **Model selection by Proximal FID.** Best true FID obtained by selecting checkpoints via Proximal FID vs. selecting via true FID (oracle). The red $(+\cdot)$ indicates the gap to the best achievable true FID during training.

| Dataset | Proximal FID (selected) | Best true FID (oracle) |
|---|---|---|
| CIFAR-10 | 5.21 (+0.45) | 4.76 |
| FFHQ | 6.12 (+0.04) | 6.08 |
| CelebA-HQ | 6.90 (+0.54) | 6.36 |
| AFHQ-v2 | 5.45 (+0.06) | 5.39 |

## 4.9 WHY SCORE DISTILLATION BEYOND ACCELERATION?

Score distillation is traditionally viewed as a means of *accelerating* sampling by compressing a multi-step diffusion process into a single forward pass. However, our paper aims to reveal a more fundamental **conceptual shift**: In corrupted or low-quality data regimes, score distillation serves as *a principled mechanism* for improving the sample quality of teacher diffusion models.

**Intuitive Understanding:** Diffusion models are trained using the ELBO objective for maximum likelihood, which minimizes the *forward* KL divergence, a mode-covering objective. As a result, teacher diffusions are incentivized to place probability mass on *all* plausible locations suggested by noisy data, even if these regions do not correspond to the true clean distribution.

In contrast, the *reverse* distillation objective (Eq. 5) takes its expectation over the *student generator's* distribution, which naturally induces *mode-seeking* behavior: the generator does not need to cover all regions where the teacher assigns non-zero mass. Instead, it is encouraged to place probability only where it will actually sample from. Importantly, we stress that *encouraging mode seeking does not imply mode collapse*. This is also evidenced by the FID and Recall in Tab. 7, qualitative results in Fig. 2, and extensive qualitative examples in Appendix K. This change in objective allows the one-step generator to concentrate its probability mass on the *high-density, well-supported regions* implied by the teacher's score field, while discarding diffuse or low-density regions that the teacher includes.

**Theoretical Support:** Detailed analysis is provided in the appendix. Sec. F.1 first focuses on a linear Gaussian setting, providing a direct quantitative analysis and error bounds (Eq. 9). These findings are extended to general linear corruption in Sec. F.2. Sec. F.3 derives performance bounds for the distilled generator under a more general setting. We posit that a deeper theoretical analysis of the surprising effects of score distillation under corruption is a promising research direction.

## 5 CONCLUSION

In this work, we introduced Restoration Score Distillation (RSD) to learn clean data distribution from a broad class of corruption types. Our empirical results on natural images and scientific MRI datasets show consistent improvements over existing baselines. Moreover, beyond standard diffusion objectives, the RSD framework is compatible with several corruption-aware training techniques, enabling flexible integration with recent advances in diffusion modeling. Together, our contributions highlight the potential of score distillation as a powerful mechanism for robust generative learning in real-world settings where clean data are scarce or unavailable. A detailed discussion of limitations is provided in Appendix E.

ETHICS STATEMENT

This work develops methods for training generative models from corrupted data without requiring access to clean ground truth. The primary applications we target are scientific and medical imaging domains where clean acquisitions are expensive or infeasible. Our framework does not involve human subjects, personal data, or harmful content, and thus poses minimal ethical risks. Nevertheless, as with all generative models, there is potential for misuse in creating synthetic content; to mitigate this, we emphasize the intended use of our approach in scientific and restoration contexts.

REPRODUCIBILITY STATEMENT

We provide complete algorithmic and training details in the Appendix. To ensure reproducibility, we will release all code and model checkpoints upon acceptance of this manuscript.

ACKNOWLEDGMENTS

Ying Nian Wu was supported in part by NSF DMS-2415226, DARPA W912CG25CA007, and research gifts from Amazon and Qualcomm. Tianyu Chen and Mingyuan Zhou acknowledge the support of NSF-IIS 2212418 and NIH-R37 CA271186. We gratefully acknowledge Microsoft for providing the computing resources used in this project.

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

## A   QUALITATIVE SNAPSHOTS OF GENERATED RESULTS

In this section, we present visual examples highlighting the quality of outputs produced by our model across various tasks and corruption settings. A full version of qualitative examples are in Appendix K.

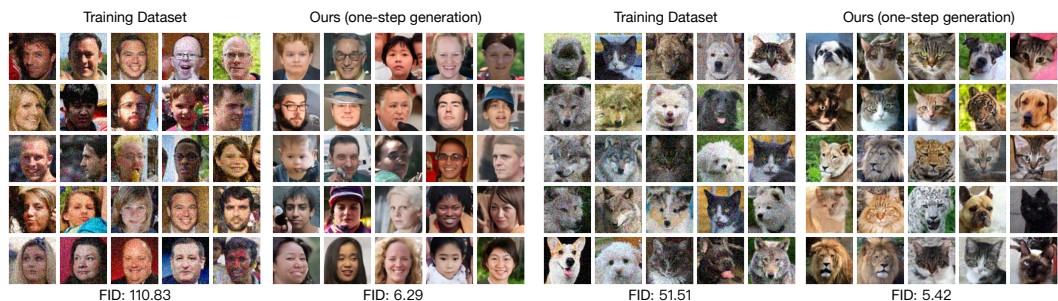

Figure 6: Qualitative results for the Denoising task. Each pair shows the corrupted input and the generation output from our RSD at $\sigma = 0.2$. The left two panels are from FFHQ, while the right two are from AFHQ-v2.

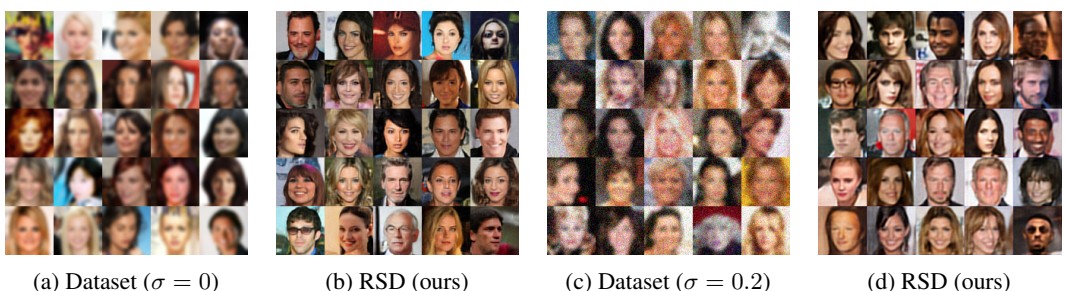

(a) Dataset ($\sigma = 0$)    (b) RSD (ours)    (c) Dataset ($\sigma = 0.2$)    (d) RSD (ours)

Figure 7: Qualitative results for the Gaussian blur task. Each pair shows the corrupted input and the generation output from our RSD.

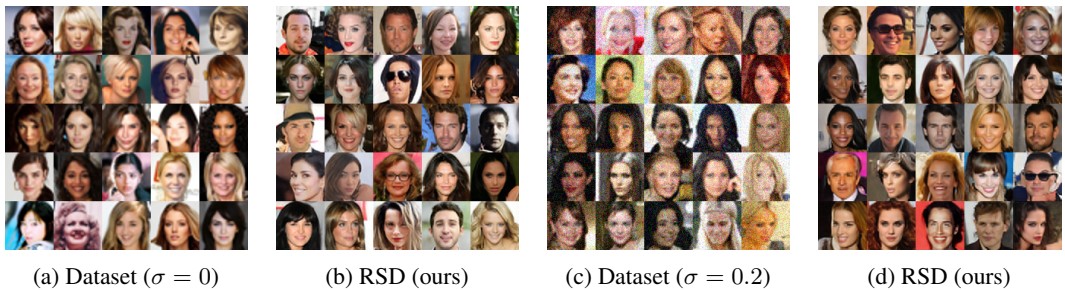

(a) Dataset ($\sigma = 0$)    (b) RSD (ours)    (c) Dataset ($\sigma = 0.2$)    (d) RSD (ours)

Figure 8: Qualitative results for the Super Resolution task. Each pair shows the corrupted input and the generation output from our RSD.

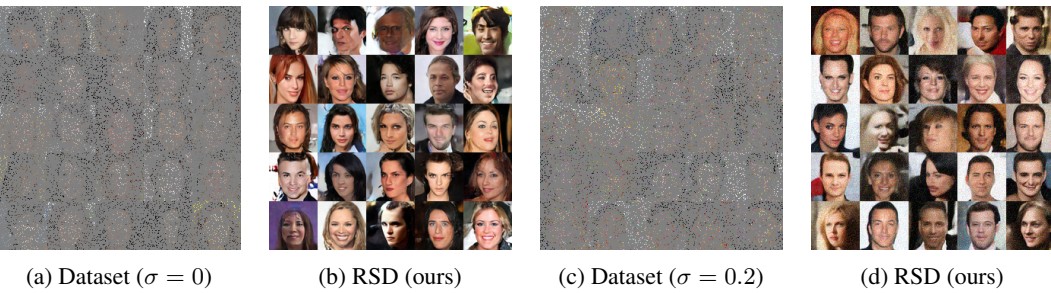

(a) Dataset ($\sigma = 0$)    (b) RSD (ours)    (c) Dataset ($\sigma = 0.2$)    (d) RSD (ours)

Figure 9: Qualitative results for the Random Inpainting task with missing probability ($p = 0.9$). Each pair shows the corrupted input and the generation output from our RSD.

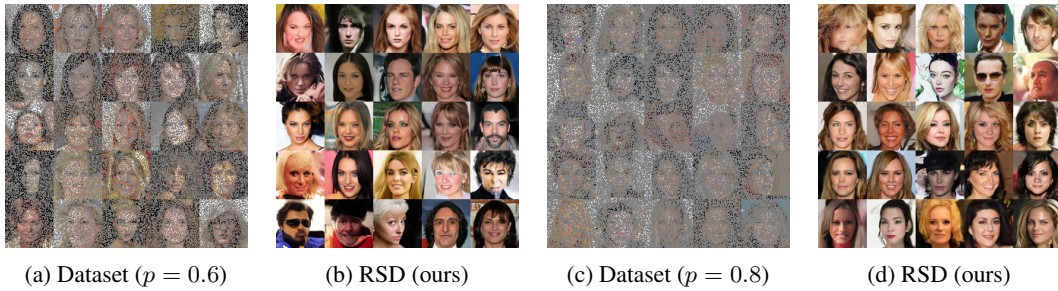

| (a) Dataset ($p = 0.6$) | (b) RSD (ours) | (c) Dataset ($p = 0.8$) | (d) RSD (ours) |
|---|---|---|---|

Figure 10: Qualitative results for the Random Inpainting task with different missing probability ($p = 0.6, p = 0.8$). Each pair shows the corrupted input and the generation output from our RSD.

## B  PROXIMAL FID

When only corrupted data are available, model selection must proceed without clean references. In standard image generation, FID is the de facto criterion, but in our setting computing FID against clean ground-truth images is infeasible.

Prior baselines—Ambient Diffusion Daras et al. (2023a) and EM-Diffusion Bai et al. (2024)—report FID scores yet do not specify how to perform *model selection* under corruption during training. To address this gap, we introduce *Proximal FID*, a model-selection metric tailored to corrupted-data regimes. Concretely, we generate 50k clean samples $\{x^{(i)}\}_{i=1}^{50k}$ from the current generator, corrupt them to match the training noise and operator—yielding

$$\left\{\mathcal{A}\left(x^{(i)}\right) + \sigma\,\epsilon^{(i)}\right\}_{i=1}^{50k},$$

and compute FID against the corrupted training set $\{y^{(i)}\}_{i=1}^{n}$. As shown on CIFAR-10 in Fig. 5, Proximal FID tracks the true FID closely throughout distillation. Quantitatively, Table 10 shows that the model chosen by Proximal FID attains near-optimal *true* FID across datasets (e.g., 6.12 vs. best 6.08 on FFHQ) for denoising task, and Table 11 for general corruption task. We further visualize the dynamics on FFHQ, CelebA-HQ, and AFHQ-v2 in Fig. 11, and extend the analysis to multiple corruption operators in Fig. 12. Taken together, these results support Proximal FID as a practical and reliable proxy for model selection when clean data are unavailable.

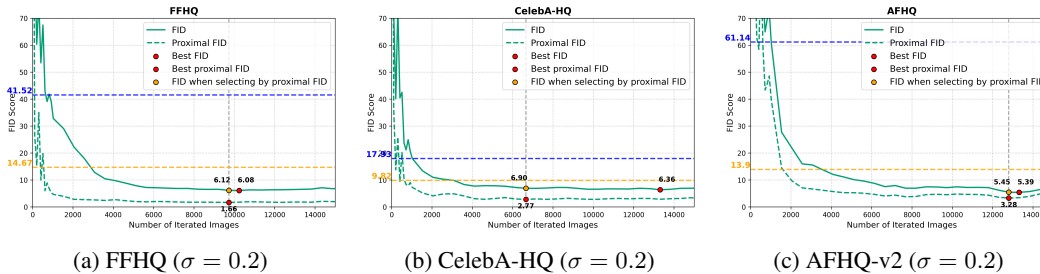

| (a) FFHQ ($\sigma = 0.2$) | (b) CelebA-HQ ($\sigma = 0.2$) | (c) AFHQ-v2 ($\sigma = 0.2$) |
|---|---|---|

Figure 11: **Across datasets:** True FID vs. *Proximal FID* during distillation on FFHQ, CelebA-HQ, and AFHQ-v2 (denoising, $\sigma = 0.2$). In all cases, Proximal FID reliably identifies checkpoints with near-optimal true FID.

## C  TRAINING AND INFERENCE EFFICIENCY

We use the denoising setting with $\sigma = 0.2$ as a running example to quantify efficiency. Our approach improves not only accuracy but also end-to-end efficiency in both training and inference. All experiments were conducted on a Linux system with $8\times$NVIDIA RTX A6000 GPUs unless otherwise stated.

**Training.** The additional distillation phase introduces only a minor overhead: the FID of the one-step generator rapidly decreases and *surpasses* the Teacher (Teacher-Truncated) within **4 hours**.

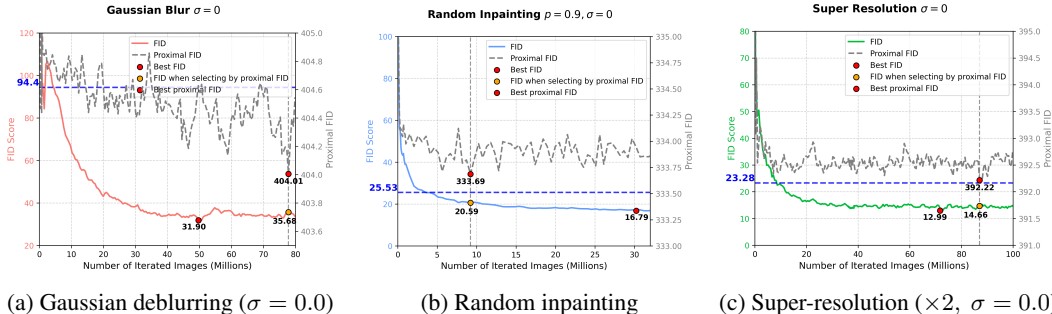

(a) Gaussian deblurring ($\sigma = 0.0$)  (b) Random inpainting  (c) Super-resolution ($\times 2$, $\sigma = 0.0$)

Figure 12: **Across corruption operators:** True FID vs. *Proximal FID* during training for Gaussian deblurring, random inpainting, and super-resolution. Proximal FID consistently tracks the true FID and supports reliable model selection across operators.

Table 11: Comparison between the best true FID and the FID of the model selected by proximal FID for the general corruption task.

| Noise ($\sigma$) | Task | Best FID | FID Selected by Proximal FID |
|---|---|---|---|
| $\sigma = 0.0$ | Gaussian Deblurring | 31.90 | 35.68 |
| | Random Inpainting ($p = 0.9$) | 16.79 | 20.59 |
| | Super Resolution ($\times 2$) | 12.99 | 14.66 |
| $\sigma = 0.2$ | Gaussian Deblurring | 76.98 | 88.29 |
| | Random Inpainting ($p = 0.9$) | 79.48 | 83.98 |
| | Super Resolution ($\times 2$) | 22.00 | 27.42 |

Table 12: **Training and inference efficiency of our method.** During training, the additional distillation phase introduces only a minor overhead, as FID decreases rapidly and surpasses the teacher diffusion model, Teacher-Truncated, within just 4 hours. For inference, our one-step generator enables the generation of 50k images in only 20 seconds, achieving a $30\times$ speedup.

| Datasets | Pretraining Time | Distillation Time to Achieve the Same FID as | | | Time to Generate 50k Images | |
|---|---|---|---|---|---|---|
| | | Teacher-Full | Teacher-Truncated | Best | Diffusion | RSD |
| CIFAR-10 | | 7 minutes | ~3 hours | ~3 days | 10 minutes | 20 seconds |
| FFHQ | ~2 days | 56 minutes | ~3 hours | ~9 hours | | |
| CelebA-HQ | | 34 minutes | ~2 hours | ~13 hours | 15 minutes | 30 seconds |
| AFHQ-v2 | | 80 minutes | ~3 hours | ~13 hours | | |

Representative wall-clock times across datasets are summarized in Table 12. During distillation, we employ early stopping when the validation FID begins to diverge.

**Inference.** Our one-step generator produces 50k images in $\sim$20 s on 4$\times$NVIDIA RTX A6000 GPUs with batch size 1024, compared to $\sim$10 min for the diffusion teacher—yielding a $30\times$ speedup. Inference wall-clock measurements are reported in the rightmost columns of Table 12.

# D  ADDITIONAL EXPERIMENTS

## D.1  MORE METRICS

The detailed results show that our distilled RSD generator achieves both strong density modeling (low FID) and robust mode coverage (high Recall).

Table 13: Comparison of RSD with teacher diffusion and baselines on CIFAR-10. *Metrics for DDPM and Rectified Flow (ODE) are copied from the Rectified Flow Liu et al. (2022) paper's Table 1.

| Method | $\sigma$ | FID ($\downarrow$) | IS ($\uparrow$) | Precision ($\uparrow$) | Recall ($\uparrow$) | KID ($\downarrow$) |
|---|---|---|---|---|---|---|
| **Teacher** | 0.2 | 12.21 | 8.312 | 0.595 | 0.416 | 0.00593 |
| **RSD** | 0.4 | 21.63 | 7.934 | 0.536 | 0.384 | 0.01270 |
| **RSD** | 0.2 | 4.77 | 9.165 | **0.650** | 0.564 | 0.00252 |
| **RSD** | 0.1 | 3.98 | 9.346 | 0.643 | **0.578** | **0.00157** |
| **DDPM** | 0.0 | 3.21 | 9.46 | N/A | 0.57 | N/A |
| **Rectified Flow (ODE)** | 0.0 | 2.58 | 9.60 | N/A | 0.57 | N/A |

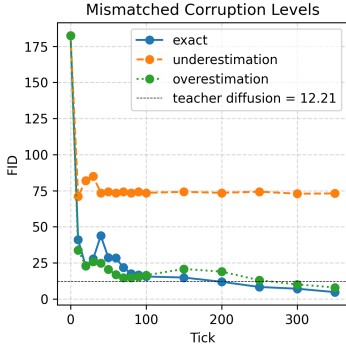

Figure 13: **Mismatched Corruption Levels during Distillation.** Our method remains robust even when the assumed corruption level is mismatched during distillation.

## D.2 UNKNOWN $\sigma$.

When $\sigma$ is unavailable at training time, we adopt a simple strategy: (1) estimate per-image noise level using the off-the-shelf estimator implemented via `skimage.restoration.estimate_sigma` Donoho and Johnstone (1994); (2) for reliability, select the *maximum* estimated $\hat{\sigma}$ over a small calibration set. Concretely, we sample 100 noisy CIFAR-10 images, compute $\hat{\sigma}$ per image across 100 independent trials, and use the trial-wise maximum as the working $\hat{\sigma}$. As shown in Table 14, this deliberately *slightly overestimates* the noise, a regime where RSD remains stable (see experiments in Table 15) and, in line with blind denoising results Zhang et al. (2018; 2017); Dabov et al. (2007), helps avoid under-regularization.

Table 14: **Unknown noise: true $\sigma$ vs. $\hat{\sigma}$ on CIFAR-10.**

Table 15: **Misspecification study on CIFAR-10.** RSD is robust to slight overestimation when the true $\sigma = 0.2$.

| $\sigma$ | $\hat{\sigma}$ | 95% CI |
|---|---|---|
| 0.05 | 0.07 | [0.073, 0.074] |
| 0.10 | 0.12 | [0.119, 0.121] |
| 0.20 | 0.22 | [0.215, 0.216] |
| 0.30 | 0.31 | [0.311, 0.313] |

| | Teacher-Full | Teacher-Trunc. | RSD |
|---|---|---|---|
| $\hat{\sigma} = 0.15$ (under) | 80.60 | 49.78 | 103.55 |
| $\hat{\sigma} = 0.20$ (true) | 60.73 | 12.21 | **4.77** |
| $\hat{\sigma} = 0.25$ (over) | 42.99 | 88.11 | **16.07** |

**Effect of misspecified $\sigma$.** To isolate the impact of noise misspecification, we fix the ground-truth level at $\sigma = 0.2$ on CIFAR-10 and evaluate three settings: (i) *underestimation*, $\hat{\sigma} = 0.15$; (ii) *correct*, $\hat{\sigma} = 0.20$; and (iii) *overestimation*, $\hat{\sigma} = 0.25$. The same $\hat{\sigma}$ is used consistently in both Phase I pretraining (noisy corruption objective) and Phase II distillation. As shown in Table 15 and Fig. 13, RSD attains its best accuracy under correct specification and remains competitive under mild overestimation, whereas underestimation is substantially more harmful—echoing observations in blind denoising Dabov et al. (2007); Zhang et al. (2018; 2017). Thus, when $\sigma$ is unknown, slight overestimation is a robust practical choice, consistent with the strategy shown in Table 14. A

Table 16: $\times 2$ **super-resolution on CelebA-HQ (100 images).** We compare a pseudo-inverse baseline $(\mathcal{A}^\dagger y)$, EM-Diffusion (few-shot) with DPS, Teacher Diffusion with DPS, and our RSD prior with latent optimization (Eq. 6). Higher PSNR/SSIM and lower LPIPS/FID are better.

| Method | PSNR ↑ | SSIM ↑ | LPIPS ↓ | Prior FID ↓ |
|---|---|---|---|---|
| $\mathcal{A}^\dagger y$ | 24.065 | 0.816 | 0.125 | 23.94 |
| EM-Diffusion (few-shot, DPS) | 25.678 | 0.839 | 0.056 | 58.99 |
| Teacher Diffusion (DPS) | 23.810 | 0.778 | 0.056 | 23.28 |
| **RSD (ours, latent opt.)** | **27.803** | **0.909** | **0.047** | **12.99** |

complementary 2D toy example in Appendix G corroborates this conclusion. We further include an ablation where the noise level is drawn from a distribution, i.e., $\sigma \sim p(\sigma)$. Our method remains robust under this setting as well; see Appendix D.4 for details.

### D.3 MORE CONDITIONAL INVERSE PROBLEM RESULTS

Our primary goal is to learn a strong generative *prior* solely from corrupted data. Once such a prior is obtained via RSD, it is agnostic to the forward operator, enabling conditional generation under arbitrary measurement models. A natural approach with our one-step generator is to solve

$$\min_z \ \left\| \mathcal{A}(G_\theta(z)) - y \right\|_2^2, \tag{6}$$

which enforces data consistency through the measurement process.

**Setup.** Beyond the denoising task discussed in Section 4.5, we consider an additional conditional inverse problem. Specifically, we use the generator trained on the $\times 2$ super-resolution task and evaluate it in a conditional inverse setting where $\mathcal{A}$ corresponds to a $\times 2$ down-sampling operator. We conduct experiments on 100 CelebA-HQ images and compare RSD against EM-Diffusion (few-shot) and the Teacher Diffusion prior. For a fair comparison, RSD optimizes Eq. 6 for 1000 steps using Adam with a learning rate of 0.1, while both EM-Diffusion and Teacher Diffusion adopt DPS Chung et al. (2022), a classical solver for diffusion priors, with 1000 diffusion sampling steps. LPIPS is computed using AlexNet.

**Results.** Table 16 reports PSNR, SSIM, LPIPS, and the prior's FID. RSD attains the best performance across all metrics (e.g., PSNR = 27.803, LPIPS = 0.047), and its learned prior achieves a strong FID of 12.99, highlighting the benefit of high-quality priors for conditional generation.

### D.4 RANDOM NOISE ABLATION

We conducted an additional experiment in a more challenging setting where each sample is corrupted with a different noise level. Specifically, we adopt the corruption model:

$$y = x + \sigma\epsilon, \quad \sigma \sim p(\sigma)$$

where $p(\sigma) = \text{Uniform}[0.15, 0.25]$. This naturally introduces variability across samples both in terms of noise level and corruption behavior.

To accommodate this setup, we made a minimal modification to Algorithm 1—replacing Lines 6 from the fixed-noise setting ($\tilde{y} = x_g + \sigma\epsilon$) to the sample-dependent corruption ($\tilde{y} = x_g + \sigma\epsilon, \sigma \sim p(\sigma)$). With this simple change, RSD can be applied directly to heterogeneous corruption settings.

We tested this on the CIFAR-10 dataset, and the results are reported below in Table 17.

### D.5 CHOICE OF DISTILLATION LOSS

In our experiments (Sec. 4), the distillation phase adopts the SiD loss (Eq. 5) by default. Other distillation objectives are also applicable—e.g., KL-based variants such as SDS Poole et al. (2022),

Table 17: **Comparison of FID scores.** Lower is better.

| Method | FID |
|---|---|
| Teacher-Full | 50.29 |
| Teacher-Truncated ($\sigma = 0.25$) | 16.21 |
| Teacher-Truncated ($\sigma = 0.15$) | 11.80 |
| **RSD (ours)** | **5.67** |

Table 18: **Ablation of distillation losses on CIFAR-10 denoising** ($\sigma \in \{0.1, 0.2, 0.4\}$; FID)

| Method | $\sigma = 0.1$ | $\sigma = 0.2$ | $\sigma = 0.4$ |
|---|---|---|---|
| SDS | $> 200$ | $> 200$ | $> 200$ |
| DMD | $12.52 \pm 0.04$ | $7.48 \pm 0.06$ | $30.09 \pm 0.23$ |
| SiD | $\mathbf{3.98} \pm 0.04$ | $\mathbf{4.77} \pm 0.03$ | $\mathbf{21.63} \pm 0.03$ |

DMD Yin et al. (2024b) (also referred to as Diff-Instruct Luo et al. (2023) or VSD Wang et al. (2024b)), and SiD Zhou et al. (2024). For completeness, we report their generator-level results below and defer implementation details (e.g., time-scheduling and hyperparameters) to the original papers.

We ablate this design choice on the CIFAR-10 denoising task; results are summarized in Table 18.

**Notes.** All distillation variants use the *default* hyperparameters from their official repositories; we did not tune hyperparameters. The relatively weaker performance of D-SDS and D-DMD in Table 18 may therefore reflect suboptimal default settings for the corrupted-data regime rather than intrinsic limitations of the losses. Empirically, SiD is robust under defaults and already yields strong results, hence our choice to use SiD for all main experiments (Sec. 4). Importantly, the distillation stage in our framework is *modular* and can be replaced by more advanced objectives; future work may close the gap—or even surpass SiD—via principled hyperparameter tuning and improved losses.

# E   DISCUSSIONS AND LIMITATIONS

**Solving inverse problems.** We demonstrate how to use the distilled generator as a learned prior for inverse problems in Section 4.7 and Appendix D.3. Concretely, given measurements $y = \mathcal{A}(x) + \text{noise}$, we recover a plausible $x$ by optimizing over the latent $z$ so that the synthesized sample matches the observations, e.g.,

$$\min_z \ \left\| \mathcal{A}(G_\theta(z)) - y \right\|_2^2 \quad \text{(optionally with regularization or priors on } z\text{).}$$

A broader treatment—including principled conditioning, data-consistency guidance, and plug-and-play/score-based solvers—is a promising direction Chung et al. (2022); Zhang et al. (2024); Zhu et al. (2024); Zhang and Leong (2025); Chen et al. (2024b), with applications in scientific and engineering domains where reconstructing clean signals from measurements is critical.

**Applications in scientific discovery.** Our approach is particularly well-suited for scientific discovery, where clean observational data are often scarce or fundamentally unobtainable. Extending our method to datasets across diverse scientific domains is a promising avenue for future research. For instance, ground-truth black hole images are inherently unobservable, yet large collections of corrupted telescope measurements are available, as demonstrated by the Event Horizon Telescope (EHT) observations Akiyama et al. (2019). EHT relies on Very Long Baseline Interferometry (VLBI), where the measurement process is modeled as a 2D Fourier transform of the sky brightness distribution. Specifically, the forward model can be expressed as

$$V(u, v) = \iint I(x, y) e^{-2\pi i(ux + vy)} dx dy,$$

as in Eq. 2 of Akiyama et al. (2019), or in practice as

$$V_{(a,b)}^t = g_a^t g_b^t e^{-i(\phi_a^t - \phi_b^t)} \mathcal{I}_{(a,b)}^t(z) + \eta_{(a,b)}^t,$$

as described in Eq. 16 of Zheng et al. (2025).

**Unknown corruption operator.** We first note that in many scientific applications such as backhole imaging and multi-coil MRI, the corruption operator is **known**. In settings where the true corruption operator is unknown, off-the-shelf estimators can provide usable approximations of the corruption process, as illustrated in Section 4.5. Crucially, our method does not require an explicit closed-form specification of the operator: it only assumes access to its *forward propagation* (i.e., the ability to apply the corruption). This black-box requirement confers a key advantage—our approach can recover salient properties of the underlying data distribution without explicit knowledge of how the corruption is parameterized.

# F  THEORETICAL RESULTS AND PROOFS

We provide theoretical insights to support our empirical results. In Section F.1, we first focus on a simpler, linear Gaussian setting where we explicitly analyze the optimization landscape of score distillation given noisy samples, yielding quantitative error bounds and characterizations of global minimizers. Then, we offer several extensions of this theory in Section F.2, including handling linear corruption and multiple noise levels. Finally, in Section F.3, we provide a more general analysis asking when can the distilled student achieve a strictly smaller Fisher divergence to the clean distribution than the teacher. This regime will operate under distributional assumptions on the data and corruption along with capacity and optimization assumptions on the teacher and generator.

## F.1  QUANTITATIVE ANALYSIS IN A LINEAR SETTING

To begin, we first analyze the performance of score distillation in a stylized setting where the underlying data distribution is Gaussian. In particular, we will assume that our data follows a low-rank linear model for our analysis.

**Assumption 1** (Linear Low-Rank Data Distribution). *Suppose our underlying data distribution is given by a low-rank linear model $x = Ez \sim p_X$ and $z \sim \mathcal{N}(0, I_r)$, where $E \in \mathbb{R}^{d \times r}$ with $r < d$ and with orthonormal columns (i.e., $E^T E = I_r$).*

Assumption 1 is equivalent to $p_X := \mathcal{N}(0, EE^T)$. For a fixed corruption noise level $\sigma > 0$, consider the setting we only have access to the noisy distribution $y = x + \sigma\epsilon$, where $x \sim p_X$ and $\epsilon \sim \mathcal{N}(0, I_d)$. In other words, $p_{Y,\sigma} := \mathcal{N}(0, EE^T + \sigma^2 I_d)$. In our setting we assume that we have perfectly learned the noisy score:

**Assumption 2** (Perfect Score Estimation). *Suppose we can estimate the score function of corrupted data $y$ perfectly:*

$$\nabla \log p_{Y,\sigma}(x) = -\left(EE^T + \sigma^2 I_d\right)^{-1} x.$$

Our goal is to distill this distribution into a distribution $p_{G_\theta} := (G_\theta)_\sharp(\mathcal{N}(0, I_d))$ given by the push-forward of $\mathcal{N}(0, I_d)$ by a generative network $G_\theta : \mathbb{R}^d \to \mathbb{R}^d$. To model a U-Net Ronneberger et al. (2015) style architecture with bottleneck structure, we assume $G_\theta$ satisfies the following low-rank linear structure detailed in Assumption 3.

**Assumption 3** (Low-Rank Linear Generator). *Assume the generator is a low-rank linear mapping, where $G_\theta$ is parameterized by $\theta = (U, V)$ where $U, V \in \mathbb{R}^{d \times r}$ with $r < d$ and has the form:*

$$G_\theta(z) := UV^T z.$$

Note that $G_\theta$ induces a degenerate low-rank Gaussian distribution $p_{G_\theta} := \mathcal{N}(0, UV^T VU^T)$. Consider a bounded noise schedule $(\sigma_t) \subseteq [\sigma_{\min}, \sigma_{\max}]$ for some $0 < \sigma_{\min} < \sigma_{\max} < \infty$ and perturbed data points $x_t = x + \sigma_t \epsilon$ where $\epsilon \sim \mathcal{N}(0, I_d)$ and $x \sim p_{G_\theta}$. Then $x_t \sim p_{G_\theta}^{\sigma_t} := \mathcal{N}(0, UV^T VU^T + \sigma_t^2 I_d)$. To distill the noisy distribution, we minimize the score-based loss (or Fisher divergence) as in Zhou et al. (2024):

$$\mathcal{L}(\theta) := \mathbb{E}_{t \sim \text{Unif}(0,1)} \mathbb{E}_{x_t \sim p_{G_\theta}^{\sigma_t}} \left[ \left\| s_{\sigma,\sigma_t}(x_t) - \nabla \log p_{G_\theta}^{\sigma_t}(x_t) \right\|_2^2 \right]. \tag{7}$$

Here, $s_{\sigma,\sigma_t}(x) := -(EE^T + (\sigma^2 + \sigma_t^2)I_d)^{-1} x$. Note this objective is similar to Eq. 5, but with the real score in place of the fake score. This is also considered the idealized distillation loss (see Eq.

(8) in Zhou et al. (2024)). In Theorem 1, we show that minimizing Eq. 7 over a certain family of non-degenerate parameters finds a distilled distribution with **smaller** Wasserstein-2 distance to the underlying clean distribution.

**Theorem 1.** *Fix $\sigma > 0$. Under Assumptions 1, 2, and 3, consider the family of parameters $\theta = (U, V)$ such that*

$$\theta \in \Theta := \{(U, V) : U^T U = I_r, V^T V \succ 0\}.$$

*For any bounded noise schedule $(\sigma_t) \subseteq [\sigma_{\min}, \sigma_{\max}]$, the global minimizers of $\mathcal{L}$ (7) over $\Theta$, denoted by $\theta_\sigma^* := (U^*, V_\sigma^*)$, satisfy the following:*

$$U^* = EQ \text{ for some orthogonal matrix } Q \text{ and } (V_\sigma^*)^T V_\sigma^* = (1 + \sigma^2) I_r. \tag{8}$$

*For any such $\theta_\sigma^*$, the induced generator distribution $p_{G_{\theta_\sigma^*}} = \mathcal{N}(0, (1 + \sigma^2) EE^T)$ satisfies*

$$W_2^2(p_{G_{\theta_\sigma^*}}, p_X) = W_2^2(p_{Y,\sigma}, p_X) - (d - r)\sigma^2 < W_2^2(p_{Y,\sigma}, p_X). \tag{9}$$

**Discussion.** This result shows that global minimizers of the distillation loss over a family of "non-degenerate" parameters induces a distribution close to the ground truth. Moreover, we can precisely quantify the distance to the underlying distribution due to the fact that all distributions are now Gaussians and the Wasserstein-2 distance has a closed form. We further explore what the unconstrained minimizers are in Theorem 3 for the rank-one case.

Regarding our assumptions, we note that making either Gaussian or potentially more complex Gaussian mixture model assumptions is common in the generative modeling literature Chen et al. (2023); Cui et al. (2023); Wang and Vastola (2024). We further note that this result focuses on the setting where the underlying generator has low-rank structure. While it is common to make simplifying assumptions on the network architecture to understand score-based models Chen et al. (2023; 2024a), there is also recent work Wang et al. (2024a) that has shown when trained on data of low intrinsic dimensionality, score-based models can exhibit low-rank structures. Empirically, we find that neural-network-based distilled models can find such low-dimensional structures through noisy data. An interesting future direction of this work is to understand the influence of neural-network-based parameterizations of the score function along with analyzing the fake score setting.

Before we dive into the proof, we provide the following lemmas.

**Lemma 1.** *[Generalized Woodbury Matrix Identity Higham (2002)]*

*Given an invertible square matrix $A \in \mathbb{R}^{n \times n}$, along with matrices $U \in \mathbb{R}^{n \times k}$ and $V \in \mathbb{R}^{k \times n}$, define the perturbed matrix: $B = A + UV$. If $(I_k + VA^{-1}U)$ is invertible, then the inverse of $B$ is given by:*

$$B^{-1} = A^{-1} - A^{-1}U(I_k + VA^{-1}U)^{-1}VA^{-1}.$$

**Lemma 2.** *The Wasserstein-2 distance between two mean-zero Gaussians $\mathcal{N}(0, \Sigma_1)$ and $\mathcal{N}(0, \Sigma_2)$ whose covariance matrices commute, i.e., $\Sigma_1 \Sigma_2 = \Sigma_2 \Sigma_1$, is given by*

$$W_2^2(\mathcal{N}(0, \Sigma_1), \mathcal{N}(0, \Sigma_2)) = \sum_{i=1}^d \lambda_i(\Sigma_1) + \lambda_i(\Sigma_2) - 2\sqrt{\lambda_i(\Sigma_1)\lambda_i(\Sigma_2)}.$$

**Lemma 3** (Mirsky (1975)). *Suppose $A$ and $B$ are $d \times d$ complex matrices with singular values $\sigma_1(A) \geqslant \sigma_2(A) \geqslant \cdots \geqslant \sigma_d(A) \geqslant 0$ and $\sigma_1(B) \geqslant \sigma_2(B) \geqslant \cdots \geqslant \sigma_d(B) \geqslant 0$, respectively. Then*

$$|\text{tr}(AB)| \leqslant \sum_{i=1}^d \sigma_i(A)\sigma_i(B).$$

**Lemma 4.** *Let $E \in \mathbb{R}^{d \times r}$ with $r < d$ have orthonormal columns and $\Sigma \in \mathbb{R}^{r \times r}$ be symmetric positive definite. Then*

$$\underset{U^T U = I_r}{\arg\max} \, \text{tr}(EE^T U \Sigma U^T) = \{EQ : Q \text{ orthogonal}\}.$$

*Proof of Lemma 4.* Observe that by the von Neumann trace inequality (Lemma 3), we have that for any feasible $U$,

$$\text{tr}(EE^T U \Sigma U^T) = \text{tr}(U^T EE^T U \Sigma) \leqslant \sum_{i=1}^r \lambda_i(U^T EE^T U)\lambda_i(\Sigma) \leqslant \sum_{i=1}^r \lambda_i(\Sigma)$$

where the last line we used the fact that $\lambda_i(U^T E E^T U) \leqslant 1$ for each $i \in [r]$. Hence, to maximize $U \mapsto \mathrm{tr}(E E^T U \Sigma U^T)$ over $\{U : U^T U = I_r\}$, we want $U^*$ to satisfy $\mathrm{tr}(E E^T U^* \Sigma (U^*)^T) = \sum_{i=1}^{r} \lambda_i(\Sigma)$.

We claim that this occurs if and only if $U^* = EQ$ for some orthogonal $Q$. If $U^* = EQ$, then $(U^*)^T E E^T U^* = Q^T E^T E E^T E Q = I$ so

$$\mathrm{tr}(E E^T U^* \Sigma (U^*)^T) = \mathrm{tr}((U^*)^T E E^T U^* \Sigma) = \mathrm{tr}(\Sigma) = \sum_{i=1}^{r} \lambda_i(\Sigma).$$

For the other direction, suppose $U^*$ maximizes the objective. Then

$$\mathrm{tr}((U^*)^T E E^T U^* \Sigma) = \mathrm{tr}(\Sigma) \iff \mathrm{tr}\left(((U^*)^T E E^T U^* - I_r)\Sigma\right) = 0.$$

Set $Q := E^T U^*$. Note that the eigenvalues of $Q^T Q$ are bounded by 1 so $Q^T Q - I_r$ is negative semi-definite while $\Sigma$ is positive definite. But if $\mathrm{tr}((Q^T Q - I_r)\Sigma) = 0$, by positive definiteness of $\Sigma$, we must have $Q^T Q - I_r = 0$, i.e., $Q^T Q = I_r$. This means $Q$ is orthogonal. Since $Q$ is orthogonal and $Q = E^T U^* \implies U^* = EQ$, as desired. $\qquad\square$

**Lemma 5.** *Fix $\sigma > 0$ and consider a noise schedule $\sigma_t > 0$ for $t \in (0,1)$ such that $(\sigma_t) \subseteq [\sigma_{\min}, \sigma_{\max}]$ for some $0 < \sigma_{\min} < \sigma_{\max} < \infty$. Define the function $f_\sigma : (0,\infty) \to \mathbb{R}$ by*

$$f_\sigma(u) := \mathbb{E}_{t\sim\mathrm{Unif}(0,1)}\left[\frac{u}{(\sigma^2 + \sigma_t^2 + 1)^2} - \frac{u}{\sigma_t^2(u + \sigma_t^2)}\right].$$

*Then $f_\sigma$ is strictly convex and has a unique minimizer at $u^* = \sigma^2 + 1$ which is the unique solution to the equation*

$$\mathbb{E}_{t\sim\mathrm{Unif}(0,1)}\left[\frac{1}{(\sigma^2 + \sigma_t^2 + 1)^2}\right] = \mathbb{E}_{t\sim\mathrm{Unif}(0,1)}\left[\frac{1}{(u^* + \sigma_t^2)^2}\right].$$

*Proof of Lemma 5.* First, note that the conditions on $\sigma_t$ ensure that all of the following expectations are finite. By direct calculation, we have the derivatives of $f_\sigma$ are

$$f_\sigma'(u) = \mathbb{E}_t\left[\frac{1}{(\sigma^2 + \sigma_t^2 + 1)^2}\right] - \mathbb{E}_t\left[\frac{1}{(\sigma_t^2 + u)^2}\right] \text{ and } f_\sigma''(u) = \mathbb{E}_t\left[\frac{2}{(\sigma_t^2 + u)^3}\right].$$

Hence $f_\sigma''(u) > 0$ for all $u > 0$ so $f_\sigma$ is strictly convex. To find its minimizer $u^*$, setting the derivative equal to 0 yields $u^*$ must satisfy

$$\mathbb{E}_t\left[\frac{1}{(\sigma^2 + \sigma_t^2 + 1)^2}\right] = \mathbb{E}_t\left[\frac{1}{(\sigma_t^2 + u^*)^2}\right].$$

Note that the point $u^* = 1 + \sigma^2$ clearly satisfies the critical point equation. Uniqueness follows due to strict convexity.

$\qquad\square$

### F.1.1 PROOF OF THEOREM 1

We break down the proof of Theorem 1 into three key steps. First, we show that minimizing the objective (Eq. 7) is equivalent to minimizing a simpler objective. Then, we show that we can derive exact analytical expressions for the global minimizers of this simpler objective, which are then global minimizers of the original score-based loss. Finally, we will directly compute the Wasserstein distance between our learned distilled distribution to the clean distribution and compare this to the noisy distribution.

**Reduction of objective function:** For $\sigma_t > 0$, define $p_{G_\theta}^{\sigma_t} := \mathcal{N}(0, U V^T V U^T + \sigma_t^2 I_d)$ and $s_{\sigma,\sigma_t}(x) := -(E E^T + (\sigma^2 + \sigma_t^2)I_d)^{-1} x$. For the proof, we will assume our parameters $\theta = (U, V) \in \Theta$ so that $U^T U = I_r$ and $V^T V \succ 0$. We consider minimizing the loss

$$\mathcal{L}(\theta) := \mathbb{E}_{t\sim\mathrm{Unif}(0,1)} \mathbb{E}_{x_t \sim p_{G_\theta}^{\sigma_t}}\left[\left\|s_{\sigma,\sigma_t}(x_t) - \nabla \log p_{G_\theta}^{\sigma_t}(x_t)\right\|_2^2\right].$$

For $t \in (0, 1)$, consider the inner expectation of the loss

$$\tilde{\mathcal{L}}_t(\theta) := \mathbb{E}_{x_t \sim p_{G_\theta}^{\sigma_t}} \left[ \left\| s_{\sigma, \sigma_t}(x_t) - \nabla \log p_{G_\theta}^{\sigma_t}(x_t) \right\|_2^2 \right].$$

For notational convenience, set $\Sigma_{\sigma,t} := EE^T + (\sigma^2 + \sigma_t^2)I_d$ and $\Sigma_{\theta,t} := UV^TVU^T + \sigma_t^2 I_d$. Then $s_{\sigma,\sigma_t}(x) := -\Sigma_{\sigma,t}^{-1}x$ and $\nabla \log p_{G_\theta}^{\sigma_t}(x) := -\Sigma_{\theta,t}^{-1}x$. First, recall that for $x_t \sim p_{G_\theta}^{\sigma_t}$ and any matrix $\Sigma$, $\mathbb{E}_{x \sim p_{G_\theta}^{\sigma_t}}[\|\Sigma x_t\|_2^2] = \|\Sigma \Sigma_{\theta,t}^{1/2}\|_F^2$. Using this, we can compute the loss as follows:

$$\begin{aligned}
\tilde{\mathcal{L}}_t(\theta) &= \mathbb{E}_{x_t \sim p_{G_\theta}^{\sigma_t}} \left[ \left\| (\Sigma_{\sigma,t}^{-1} - \Sigma_{\theta,t}^{-1})x_t \right\|_2^2 \right] \\
&= \| (\Sigma_{\sigma,t}^{-1} - \Sigma_{\theta,t}^{-1})\Sigma_{\theta,t}^{1/2} \|_F^2 \\
&= \operatorname{tr} \left( \Sigma_{\theta,t}^{1/2}(\Sigma_{\sigma,t}^{-1} - \Sigma_{\theta,t}^{-1})(\Sigma_{\sigma,t}^{-1} - \Sigma_{\theta,t}^{-1})\Sigma_{\theta,t}^{1/2} \right) \\
&= \operatorname{tr} \left( \Sigma_{\theta,t}(\Sigma_{\sigma,t}^{-1} - \Sigma_{\theta,t}^{-1})(\Sigma_{\sigma,t}^{-1} - \Sigma_{\theta,t}^{-1}) \right) \\
&= \operatorname{tr} \left( (\Sigma_{\theta,t}\Sigma_{\sigma,t}^{-1} - I_d)(\Sigma_{\sigma,t}^{-1} - \Sigma_{\theta,t}^{-1}) \right) \\
&= \operatorname{tr} \left( \Sigma_{\theta,t}\Sigma_{\sigma,t}^{-2} - \Sigma_{\theta,t}\Sigma_{\sigma,t}^{-1}\Sigma_{\theta,t}^{-1} - \Sigma_{\sigma,t}^{-1} + \Sigma_{\theta,t}^{-1} \right) \\
&= \operatorname{tr} \left( \Sigma_{\theta,t}\Sigma_{\sigma,t}^{-2} \right) - \operatorname{tr} \left( \Sigma_{\theta,t}\Sigma_{\sigma,t}^{-1}\Sigma_{\theta,t}^{-1} \right) - \operatorname{tr} \left( \Sigma_{\sigma,t}^{-1} \right) + \operatorname{tr} \left( \Sigma_{\theta,t}^{-1} \right) \\
&= \operatorname{tr} \left( \Sigma_{\sigma,t}^{-2}\Sigma_{\theta,t} \right) - 2\operatorname{tr} \left( \Sigma_{\sigma,t}^{-1} \right) + \operatorname{tr} \left( \Sigma_{\theta,t}^{-1} \right) \\
&=: C_{\sigma,t} + \operatorname{tr} \left( \Sigma_{\sigma,t}^{-2}\Sigma_{\theta,t} \right) + \operatorname{tr} \left( \Sigma_{\theta,t}^{-1} \right).
\end{aligned}$$

Using Lemma 2, it is straightforward to see that

$$\begin{aligned}
\Sigma_{\sigma,t}^{-1} &= \frac{1}{\sigma^2 + \sigma_t^2}I_d - \frac{1}{(\sigma^2 + \sigma_t^2)^2(\sigma^2 + \sigma_t^2 + 1)}EE^T \quad \text{and} \\
\Sigma_{\theta,t}^{-1} &= \sigma_t^{-2}I_d - \sigma_t^{-4}U \left( (V^TV)^{-1} + \sigma_t^{-2}I_r \right)^{-1} U^T
\end{aligned}$$

Hence the third term in $\tilde{\mathcal{L}}_t$ is given by

$$\operatorname{tr}(\Sigma_{\theta,t}^{-1}) = \operatorname{tr} \left( \sigma_t^{-2}I_d - \sigma_t^{-4}U \left( (V^TV)^{-1} + \sigma_t^{-2}I_r \right)^{-1} U^T \right) =: C_{\sigma_t} - \sigma_t^{-4}\operatorname{tr} \left( \left( (V^TV)^{-1} + \sigma_t^{-2}I_r \right)^{-1} \right)$$

where we used the cyclic property of the trace and $U^TU = I_r$ in the last equality. For the second term, let $\beta_t^2 := \sigma^2 + \sigma_t^2$ and $\gamma_{\sigma,t} := \frac{1}{\beta_t^2(\beta_t^2+1)}$. Then we have by direct computation,

$$\begin{aligned}
\operatorname{tr} \left( \Sigma_{\sigma,t}^{-2}\Sigma_{\theta,t} \right) &= \operatorname{tr} \left( \left( \beta_t^{-2}I_d - \gamma_{\sigma,t}EE^T \right) \left( \beta_t^{-2}I_d - \gamma_{\sigma,t}EE^T \right) (UV^TVU^T + \sigma_t^2 I_d) \right) \\
&= \operatorname{tr} \left( \left( \beta_t^{-4}I_d - 2\beta_t^{-2}\gamma_{\sigma,t}EE^T + \gamma_{\sigma,t}^2 EE^T \right) (UV^TVU^T + \sigma_t^2 I_d) \right) \\
&= \operatorname{tr} \left( \beta_t^{-4}UV^TVU^T - \sigma_t^2\beta_t^{-4}I_d + \left( \gamma_{\sigma,t}^2 - 2\beta_t^{-2}\gamma_{\sigma,t} \right) EE^TUV^TVU^T \right) \\
&\quad - \operatorname{tr} \left( 2\beta_t^{-2}\sigma_t^2 EE^T + \gamma_{\sigma,t}^2\sigma_t^2 I_d \right) \\
&=: \tilde{C}_{\sigma,t} + \beta_t^{-4}\operatorname{tr}(UV^TVU^T) + \left( \gamma_{\sigma,t}^2 - 2\beta_t^{-2}\gamma_{\sigma,t} \right) \cdot \operatorname{tr}(EE^TUV^TVU^T) \\
&= \tilde{C}_{\sigma,t} + \beta_t^{-4}\operatorname{tr}(V^TV) + \left( \gamma_{\sigma,t}^2 - 2\beta_t^{-2}\gamma_{\sigma,t} \right) \cdot \operatorname{tr}(EE^TUV^TVU^T)
\end{aligned}$$

where we used the cyclic property of trace and orthogonality of $U$ in the final line. Combining the above displays, we get that there exists a constant $C_{\sigma,\sigma_t} := C_{\sigma,t} + C_{\sigma_t} + \tilde{C}_{\sigma,t}$ such that

$$\begin{aligned}
\tilde{\mathcal{L}}_t(\theta) &= C_{\sigma,\sigma_t} + \left( \frac{1}{\beta_t^4(\beta_t^2+1)^2} - \frac{2}{\beta_t^4(\beta_t^2+1)} \right) \cdot \operatorname{tr}(EE^TUV^TVU^T) \\
&\quad + \beta_t^{-4}\operatorname{tr}(V^TV) - \sigma_t^{-4}\operatorname{tr} \left( \left( (V^TV)^{-1} + \sigma_t^{-2}I_r \right)^{-1} \right) \\
&=: C_{\sigma,\sigma_t} + B_t(U,V) + R_t(V)
\end{aligned}$$

where we have defined the quantities

$$B_t(U, V) := \left( \frac{1}{\beta_t^4(\beta_t^2+1)^2} - \frac{2}{\beta_t^4(\beta_t^2+1)} \right) \cdot \text{tr}(EE^TUV^TVU^T) \text{ and}$$

$$R_t(V) := \beta_t^{-4}\text{tr}(V^TV) - \sigma_t^{-4}\text{tr}\left( \left( (V^TV)^{-1} + \sigma_t^{-2}I_r \right)^{-1} \right).$$

Recalling the definition of $\mathcal{L}(\cdot)$, we have that

$$\mathcal{L}(\theta) = \mathbb{E}_{t\sim\text{Unif}(0,1)}\left[ \tilde{\mathcal{L}}_t(\theta) \right] = \mathbb{E}_{t\sim\text{Unif}(0,1)}\left[ C_{\sigma,\sigma_t} + B_t(U, V) + R_t(U, V) \right].$$

Hence we have the equivalence

$$\underset{\theta\in\Theta}{\text{argmin}}\, \mathcal{L}(\theta) = \underset{\theta\in\Theta}{\text{argmin}}\, \mathbb{E}_{t\sim\text{Unif}(0,1)}\left[ B_t(U, V) \right] + \mathbb{E}_{t\sim\text{Unif}(0,1)}[R_t(V)].$$

**Form of minimizers:** We use the shorthand notation $\mathbb{E}_t[\cdot] := \mathbb{E}_{t\sim\text{Unif}(0,1)}[\cdot]$. First, note that we can first minimize $\mathbb{E}_t[B_t(U, V)]$ over feasible $U$. But note that

$$\mathbb{E}_t[B_t(U, V)] = \underbrace{\mathbb{E}_t\left[ \frac{1}{\beta_t^4(\beta_t^2+1)^2} - \frac{2}{\beta_t^4(\beta_t^2+1)} \right]}_{<0} \text{tr}(EE^TUV^TVU^T)$$

since for any $t$, $\frac{1}{(\beta_t^2+1)^2} < \frac{2}{(\beta_t^2+1)}$. Hence minimizing $\mathbb{E}_t[B_t(U, V)]$ is equivalent to maximizing $\text{tr}(EE^TUV^TVU^T)$. Taking $\Sigma = V^TV$ in Lemma 4, we have that the minimizer of $\mathbb{E}_t[B_t(U, V)]$ is given by

$$U^* = EQ \text{ for some orthogonal } Q.$$

Moreover, the proof of Lemma 4 shows that $\text{tr}(EE^TU^*V^TV(U^*)^T) = \text{tr}(V^TV)$. This gives

$$\mathbb{E}_t[B_t(U^*, V)] = \mathbb{E}_t\left( \frac{1}{\beta_t^4(\beta_t^2+1)^2} - \frac{2}{\beta_t^4(\beta_t^2+1)} \right) \text{tr}(V^TV).$$

In summary, we now must minimize the following with respect to invertible $V$:

$$\begin{aligned}
\mathbb{E}_t[B_t(U^*, V)] + \mathbb{E}_t[R_t(V)] &= \mathbb{E}_t\left( \frac{1}{\beta_t^4(\beta_t^2+1)^2} - \frac{2}{\beta_t^4(\beta_t^2+1)} + \frac{1}{\beta_t^4} \right) \text{tr}(V^TV) \\
&\quad - \mathbb{E}_t\left[ \sigma_t^{-4}\text{tr}\left( \left( (V^TV)^{-1} + \sigma_t^{-2}I_r \right)^{-1} \right) \right] \\
&= \mathbb{E}_t\left( \frac{1}{\beta_t^4}\left( \frac{1}{\beta_t^2+1} - 1 \right)^2 \right) \text{tr}(V^TV) - \mathbb{E}_t\left[ \sigma_t^{-4}\text{tr}\left( \left( (V^TV)^{-1} + \sigma_t^{-2}I_r \right)^{-1} \right) \right] \\
&= \mathbb{E}_t\left( \frac{1}{\beta_t^4}\left( \frac{\beta_t^2}{\beta_t^2+1} \right)^2 \right) \text{tr}(V^TV) - -\mathbb{E}_t\left[ \sigma_t^{-4}\text{tr}\left( \left( (V^TV)^{-1} + \sigma_t^{-2}I_r \right)^{-1} \right) \right] \\
&= \mathbb{E}_t\left( \frac{1}{(\beta_t^2+1)^2} \right) \text{tr}(V^TV) - \mathbb{E}_t\left[ \sigma_t^{-4}\text{tr}\left( \left( (V^TV)^{-1} + \sigma_t^{-2}I_r \right)^{-1} \right) \right]
\end{aligned}$$

where in the second equality, we completed the square.

We now claim that $\mathbb{E}_t[B_t(U^*, V)] + \mathbb{E}_t[R_t(V)]$ solely depends on the eigenvalues of $V^TV$. In particular, for invertible $V$, note that $V^TV \succ 0$ so it admits the decomposition $V^TV = Q\Lambda Q^T$ where $Q^TQ = QQ^T = I_r$ and $\Lambda$ is a diagonal matrix with positive entries $\Lambda_{ii} = \lambda_i(V^TV) > 0$. Hence $\text{tr}(V^TV) = \text{tr}(Q\Lambda Q^T) = \text{tr}(Q^TQ\Lambda) = \text{tr}(\Lambda) = \sum_{i=1}^r \lambda_i(V^TV)$. Likewise, we have using

the orthogonality of $Q$ that for any $\varepsilon > 0$,

$$
\begin{aligned}
\operatorname{tr}\left(\left((V^TV)^{-1} + \varepsilon^{-2}I_r\right)^{-1}\right) &= \operatorname{tr}\left(\left((Q\Lambda Q^T)^{-1} + \varepsilon^{-2}I_r\right)^{-1}\right) \\
&= \operatorname{tr}\left(\left(Q\Lambda^{-1}Q^T + \varepsilon^{-2}QQ^T\right)^{-1}\right) \\
&= \operatorname{tr}\left(\left(Q\left(\Lambda^{-1} + \varepsilon^{-2}I_r\right)Q^T\right)^{-1}\right) \\
&= \operatorname{tr}\left(Q\left(\Lambda^{-1} + \varepsilon^{-2}I_r\right)^{-1}Q^T\right) \\
&= \operatorname{tr}\left(\left(\Lambda^{-1} + \varepsilon^{-2}I_r\right)^{-1}\right) \\
&= \sum_{i=1}^{r} \frac{1}{\lambda_i(V^TV)^{-1} + \varepsilon^{-2}} \\
&= \sum_{i=1}^{r} \frac{\lambda_i(V^TV) \cdot \varepsilon^2}{\lambda_i(V^TV) + \varepsilon^2}.
\end{aligned}
$$

In sum, the final objective is a particular function of the eigenvalues of $V^TV$:

$$
\begin{aligned}
\mathbb{E}_t[B_t(U^*, V)] + \mathbb{E}_t[R_t(V)] &= \sum_{i=1}^{r} \mathbb{E}_t\left[\frac{\lambda_i(V^TV)}{(\beta_t^2 + 1)^2} - \frac{\lambda_i(V^TV)}{\sigma_t^2(\lambda_i(V^TV) + \sigma_t^2)}\right] \\
&= \sum_{i=1}^{r} \mathbb{E}_t\left[\frac{\lambda_i(V^TV)}{(\sigma^2 + \sigma_t^2 + 1)^2} - \frac{\lambda_i(V^TV)}{\sigma_t^2(\lambda_i(V^TV) + \sigma_t^2)}\right] \\
&=: \sum_{i=1}^{r} f_\sigma(\lambda_i(V^TV)).
\end{aligned}
$$

In Lemma 5, we show that the function $u \mapsto f_\sigma(u)$ is strictly convex on $(0, \infty)$ with a unique minimizer at $1 + \sigma^2$. Thus $V \mapsto B(U^*, V) + R(V)$ for invertible $V$ is minimized when the gram matrix of $V_\sigma^*$ has equal eigenvalues $\lambda_i((V_\sigma^*)^T V_\sigma^*) = 1 + \sigma^2$ for all $i \in [r]$. Since all of its eigenvalues are the same, by the Spectral Theorem, we must have that $(V_\sigma^*)^T V_\sigma^* = (1 + \sigma^2)I_r$.

**Wasserstein bound:** We now show the Wasserstein error bound. Note that $\theta_\sigma^* = (U^*, V_\sigma^*)$ induces the distribution $p_{G_{\theta_\sigma^*}}$ defined by

$$
x = G_{\theta_\sigma^*}(z), \ z \sim \mathcal{N}(0, I_d) \iff x \sim p_{G_{\theta_\sigma^*}} := \mathcal{N}(0, EQ(V_\sigma^*)^T V_\sigma^* Q^T E^T) = \mathcal{N}(0, (1+\sigma^2)EE^T).
$$

Then by Lemma 2, we have

$$
\begin{aligned}
W_2^2(p_{Y,\sigma}, p_X) &= r\left(1 + \sigma^2 + 1 - 2\sqrt{1 + \sigma^2}\right) + (d - r)\sigma^2, \\
W_2^2(p_{G_{\theta_\sigma^*}}, p_X) &= r\left(1 + \sigma^2 + 1 - 2\sqrt{1 + \sigma^2}\right).
\end{aligned}
$$

This gives

$$
W_2^2(p_{G_{\theta_\sigma^*}}, p_X) = W_2^2(p_{Y,\sigma}, p_X) - (d - r)\sigma^2 < W_2^2(p_{Y,\sigma}, p_X).
$$

## F.2 EXTENSIONS OF THE THEORY IN SECTION F.1

We now discuss three extensions of Theorem 1: 1) we allow for additional corruption in $y$, 2) characterize the full optimization landscape in the rank-one case, and 3) analyze the global minimizers when we may have varying noise levels in the training data.

### F.2.1 ADDITIONAL MEASUREMENT CORRUPTION

We will now consider the case when the data is not simply noisy, but also exhibits more general corruption. For a fixed corruption noise level $\sigma > 0$, consider the setting we only have access to the noisy distribution $y = \mathcal{A}(x) + \sigma\epsilon$, where $\epsilon \sim \mathcal{N}(0, I_m)$ and $\mathcal{A}(x) = Ax$ with $A \in \mathbb{R}^{m \times d}$ is a linear corruption operator. By our assumption on $p_X$ (see Assumption 1) and the noise, $p_{Y,\sigma} := \mathcal{N}(0, AEE^T A^T + \sigma^2 I_m)$. In order to get rid of error in estimating score, we assume that we have perfectly learned the noisy score:

**Assumption 4** (Perfect Score Estimation with $\mathcal{A}$). *Suppose $A \in \mathbb{R}^{m \times d}$ with $\mathrm{rank}(A) = m$ and we can estimate the score function of corrupted data $y$ perfectly:*

$$s_{A,\sigma^2}(x) := \nabla \log p_{Y,\sigma^2}(x) = -\left(AEE^T A^T + \sigma^2 I_m\right)^{-1} x.$$

Our goal is to match the corrupted noise distribution's score with the score of $p_{G_\theta}$ under Assumption 3 corrupted by $A$ over a series of noise schedules $(\sigma_t)$, which is given by

$$\tilde{p}_{G_\theta}^{\sigma_t}(y) = \mathcal{N}(0, AUV^TVU^T A^T + \sigma_t^2 I_m).$$

To distill the noisy distribution, we minimize the score-based loss (or Fisher divergence) as in Zhou et al. (2024):

$$\mathcal{L}(\theta) := \mathbb{E}_{t \sim \mathrm{Unif}[0,1]} \mathbb{E}_{y_t \sim \tilde{p}_{G_\theta}^{\sigma_t}} \left[ \| s_{A,\sigma^2 + \sigma_t^2}(y_t) - \nabla \log \tilde{p}_{G_\theta}^{\sigma_t}(y_t) \|_2^2 \right]. \tag{10}$$

We show that we can characterize the global minimizers of this loss, which correspond to a noise-dependent scaling of the *true* eigenspace of $p_X$ plus perturbations in the kernel of $A$. If we penalize the norm of our solution, we can nearly recover the true data distribution in Wasserstein-2 distance up to the noise in our measurements. We consider the rank-$r = 1$ case for simplicity.

**Theorem 2.** *Fix $\sigma > 0$ and consider $e \in \mathbb{R}^d$ with unit norm and $Ae \neq 0$. Under Assumptions 1, 4, and 3 and any bounded noise schedule $(\sigma_t) \subseteq [\sigma_{\min}, \sigma_{\max}]$, the set of global minimizers of the loss 10 under the parameterization $u \mapsto \theta(u) = (u, u/\|u\|)$ is given by*

$$\Theta_* := \left\{ \pm\sqrt{1 + \frac{\sigma^2}{\|Ae\|^2}} \cdot e \right\} + \ker(A).$$

*If $e \in \mathrm{Im}(A^T)$, we have that $\theta_* = \theta_*(u_*)$ with the minimum norm solution $u_* \in \mathrm{argmin}_{u \in \Theta_*} \|u\|$ satisfies*

$$W_2^2(p_X, p_{G_{\theta^*}}) = \left( \sqrt{1 + \frac{\sigma^2}{\|Ae\|^2}} - 1 \right)^2.$$

**Discussion.** Theorem 2 aims to provide a quantitative bound of the Wasserstein distance between the distribution learned via distillation and the target clean distribution. To do this, we characterize the global minimizers of the loss, which correspond to scalings of the true principal component $e$ plus perturbations in the kernel of $A$. The scaling depends on an effective signal-to-noise ratio $SNR := \|Ae\|^2/\sigma^2$ Note that terms involving perturbations in the kernel of $A$ are expected since the corruption compresses the data, leaving many plausible images that could give rise to the same measurements (a fundamental part of the ill-posedness in inverse problems). Furthermore, we show that if we penalize the norm of our parameter, the Wasserstein distance simplifies into a more interpretable quantity. A more general bound for all elements in the set of global minimizers is shown at the end of the proof. The intuition for the Theorem is that if we encourage finding a "simple" model (i.e., one with low-norm), the Wasserstein distance decreases and is effectively inversely proportional to the $SNR$. The distance between our learned distilled distribution and the true distribution goes to zero whenever 1) the noise goes to zero or 2) the signal strength increases. This result recovers the rank-1 version of Theorem 2 when $A = I$, showing that the learned distilled distribution's Wasserstein distance to the true distribution is less than the noisy distribution's Wasserstein distance to the true distribution subtracted by a factor of $(d-1)\sigma^2$, showing a clear improvement in distribution learning. Finally, we note that the condition $e \in \mathrm{Im}(A^T)$ we use is akin to assumptions in the compressed sensing literature on stable recovery via the construction of dual certificates Foucart and Rauhut (2013).

To prove the Theorem, we first show that the objective under the parameterization $\theta = (u, u/\|u\|)$ simplifies into a form that we directly analyze.

**Lemma 6.** *Consider the setting of Theorem 2. For $t \in [0,1]$, let $a_t := \sigma_t^2$, $c_t := a_t + \sigma^2$, and $\eta_t := \frac{2c_t + \|Ae\|^2}{c_t^2(c_t + \|Ae\|^2)^2}$. Then the objective $\mathcal{L}(\theta)$ with $\theta = (u, u/\|u\|)$ satisfies the following: there exists a constant $C$ independent of $u \in \mathbb{R}^d$ such that*

$$\mathcal{L}(\theta) = C + L(u)$$

*where*

$$L(u) := \mathbb{E}_{t \sim \text{Unif}[0,1]} \left[ \frac{1}{c_t^2} \|Au\|^2 - \eta_t (e^T A^T Au)^2 - \frac{\|Au\|^2}{a_t(a_t + \|Au\|^2)} \right].$$

*Proof of Lemma 6.* Consider the loss function $\mathcal{L}(\theta)$ under the parameterization $\theta = (u, u/\|u\|)$ for $u \neq 0$:

$$\mathcal{L}(\theta) := \mathbb{E}_{t \sim \text{Unif}[0,1]} \mathbb{E}_{y_t \sim \tilde{p}_{G_\theta}^{\sigma_t}} \left[ \left\| \left( \Sigma_{t,e,A}^{-1} - \Sigma_{t,\theta,A} \right)^{-1} y_t \right\|_2^2 \right]$$

where

$$\Sigma_{t,e,A} := Aee^T A^T + c_t I_m \text{ and } \Sigma_{t,\theta,A} := Auu^T A^T + a_t I_m.$$

Using a similar reduction in the proof of Theorem 1, we get that the above loss equals

$$\mathcal{L}(\theta) = \mathbb{E}_{t \sim \text{Unif}[0,1]} \left[ C_{\sigma,t} \right] + \mathbb{E}_{t \sim \text{Unif}[0,1]} \left[ \text{tr} \left( \Sigma_{t,e,A}^{-2} \Sigma_{t,\theta,A} \right) \right] + \mathbb{E}_{t \sim \text{Unif}[0,1]} \left[ \text{tr} \left( \Sigma_{t,\theta,A}^{-1} \right) \right]$$

where $C_{\sigma,t}$ is a constant that depends only on time and $\sigma$ and not on $\theta$. Note that using the Woodbury matrix identity in Lemma 1, we get

$$\Sigma_{t,e,A}^{-2} = (Aee^T A^T + c_t I_m)^{-2} = c_t^{-2} I_m - \frac{2c_t + \|Ae\|^2}{c_t^2 (c_t + \|Ae\|^2)^2} Aee^T A^T$$

which implies

$$\begin{aligned} \Sigma_{t,e,A}^{-2} \Sigma_{t,\theta,A} &= \left( c_t^{-2} I_m - \frac{2c_t + \|Ae\|^2}{c_t^2 (c_t + \|Ae\|^2)^2} Aee^T A^T \right) Auu^T A^T \\ &+ c_t^{-2} a_t I_m - a_t \frac{2c_t + \|Ae\|^2}{c_t^2 (c_t + \|Ae\|^2)^2} Aee^T A^T \\ &= c_t^{-2} Auu^T A^T - \eta_t Aee^T A^T Auu^T A^T + c_t^{-2} a_t I_m - a_t \eta_t Aee^T A^T. \end{aligned}$$

Also, we have that

$$\Sigma_{t,\theta,A}^{-1} = (Auu^T A^T + a_t I_m)^{-1} = a_t^{-1} I_m - \frac{Auu^T A^T}{a_t(a_t + \|Au\|^2)}.$$

Thus computing the trace, taking an expectation, and collecting terms that only involve $u$, we see that there exists a constant $C$ depending on $(\sigma_t)$, $\sigma$, $A$, and $e$ (independent of $u$) such that

$$\begin{aligned} \mathcal{L}(\theta) &= C + \mathbb{E}_t[c_t^{-2}] \text{tr}(Auu^T A^T) - \mathbb{E}_t[\eta_t] \text{tr}(Aee^T A^T Auu^T A^T) - \mathbb{E}_t \left[ \frac{\text{tr}(Auu^T A^T)}{a_t(a_t + \|Au\|^2)} \right] \\ &= C + \mathbb{E}_t[c_t^{-2}] \|Au\|^2 - \mathbb{E}_t[\eta_t] (e^T A^T Au)^2 - \mathbb{E}_t \left[ \frac{\|Au\|^2}{a_t(a_t + \|Au\|^2)} \right] \\ &=: C + L(u). \end{aligned}$$

$\square$

*Proof of Theorem 2.* Note that Lemma 6 shows that we have the simpler formula

$$\mathcal{L}(\theta) = C + L(u)$$

where

$$L(u) := \mathbb{E}_t \left[ \frac{1}{c_t^2} \|Au\|^2 - \eta_t (e^T A^T Au)^2 - \frac{\|Au\|^2}{a_t(a_t + \|Au\|^2)} \right]$$

For notational simplicity, let $G = A^T A$, $g(u) = u^T G u$, $h(u) = e^T G u$, $\phi_t(g) = g/(a_t(a_t + g))$, and $\eta_t := (2c_t + \|Ae\|^2)/(c_t^2(c_t + \|Ae\|^2)^2)$. Then our objective results in

$$L(u) := \mathbb{E}_t[c_t^{-2}] g(u) - \mathbb{E}_t[\eta_t] h(u)^2 - \mathbb{E}_t[\phi_t(g(u))] =: \tilde{c} g(u) - \tilde{\eta} h(u)^2 - \mathbb{E}_t[\phi_t(g(u))].$$

We claim that any global minimizer of $L$ is of the form $u_* = \pm\lambda_* e + q$ for some constant $\lambda_*$ to be defined and $q \in \ker(A)$. First, note that we can lower bound $L(u)$ by another function $\Psi(g(u))$ as follows: since $G$ is PSD, note that we have the generalized Cauchy Schwarz inequality:

$$h(u)^2 = (e^T G u)^2 \leqslant (e^T G e)(u^T G u) = g(e)g(u).$$

Moreover, this holds with equality if and only if $Au$ and $Ae$ are collinear, i.e., $Au = \lambda Ae$ or equivalently $u = \lambda e + v$ for $v \in \ker(A)$. Then we get the lower bound

$$L(u) = \tilde{c}g(u) - \tilde{\eta}h(u)^2 - \mathbb{E}_t[\phi_t(g(u))] \geqslant (\tilde{c} - \tilde{\eta}\|Ae\|^2)g(u) - \mathbb{E}_t[\phi_t(g(u))] =: \Psi(g(u))$$

where we have defined the new function $\Psi(g) := (\tilde{c} - \tilde{\eta}\|Ae\|^2)g - \mathbb{E}_t[\phi_t(g)]$ for $g \geqslant 0$. Since $g(u) \geqslant 0$ for every $u$, we have that

$$L(u) \geqslant \min_{g \geqslant 0} \Psi(g).$$

We first analyze the minimizers of $\Psi$ and then construct $u_*$ such that $L(u_*) = \Psi(g_*)$. First let $\Delta = (\tilde{c} - \tilde{\eta}\|Ae\|^2)$ which is given by

$$\Delta = \mathbb{E}_t\left[c_t^{-2} - \eta_t\|Ae\|^2\right] = \mathbb{E}_t\left[\frac{1}{c_t^2} - \frac{(2c_t + \|Ae\|^2)\|Ae\|^2}{c_t^2(c_t + \|Ae\|^2)^2}\right].$$

We claim that $\Delta > 0$. Indeed, note that for any $g > 0$,

$$\frac{g(2c_t + g)}{(c_t + g)^2} = \frac{g(2c_t + g)g}{c_t^2 + (2c_t + g)g} = \frac{g(2c_t + g) + c_t^2 - c_t^2}{c_t^2 + (2c_t + g)g} = 1 - \frac{c_t^2}{(c_t + g)^2}.$$

Applying this to $\Delta$ with $g = \|Ae\|^2$ yields

$$\begin{aligned}
\Delta &= \mathbb{E}_t\left[\frac{1}{c_t^2} - \frac{(2c_t + \|Ae\|^2)\|Ae\|^2}{c_t^2(c_t + \|Ae\|^2)^2}\right] \\
&= \mathbb{E}_t\left[\frac{1}{c_t^2}\left(1 - \frac{(2c_t + \|Ae\|^2)\|Ae\|^2}{(c_t + \|Ae\|^2)^2}\right)\right] \\
&= \mathbb{E}_t\left[\frac{1}{c_t^2}\left(1 - 1 + \frac{c_t^2}{(c_t + \|Ae\|^2)^2}\right)\right] \\
&= \mathbb{E}_t\left[\frac{1}{(c_t + \|Ae\|^2)^2}\right] > 0
\end{aligned}$$

as desired. This gives

$$\Psi(g) = \Delta g - \mathbb{E}_t[\phi_t(g)].$$

Observe that

$$\Psi'(g) = \Delta - \frac{\partial}{\partial g}\int_0^1 \frac{g}{a_t^2 + a_t g}dt = \Delta - \int_0^1 \frac{a_t^2}{(a_t^2 + a_t g)^2}dt = \Delta - \mathbb{E}_t\left[\frac{1}{(a_t + g)^2}\right]$$

and

$$\Psi''(g) = -\partial/\partial g\,\mathbb{E}_t[(a_t + g)^{-2}] = 2\mathbb{E}_t[(a_t + g)^{-3}] > 0\;\forall g.$$

Hence the function $\Psi$ is strictly convex. Moreover, a sign analysis reveals that

$$\Psi'(0) = \Delta - \mathbb{E}_t[1/a_t^2] = \mathbb{E}_t\left[\frac{1}{(c_t + \|Ae\|^2)^2} - \frac{1}{a_t^2}\right] = \mathbb{E}_t\left[\frac{1}{(\sigma^2 + a_t + \|Ae\|^2)^2} - \frac{1}{a_t^2}\right] < 0$$

while for $g > \sigma^2 + \|Ae\|^2$, we have

$$\Psi'(g) = \mathbb{E}_t\left[\frac{1}{(\sigma^2 + a_t + \|Ae\|^2)^2} - \frac{1}{(a_t + g)^2}\right] > 0$$

so there must exist a unique root $g_*$ such that $\Psi'(g_*) = 0$. In fact, $g_* = \sigma^2 + \|Ae\|^2$ achieves $\Psi'(g_*) = 0$. Finally, we have that

$$L(u) \geqslant \min_{g \geqslant 0} \Psi(g) = \Psi(g_*) = \Delta(\sigma^2 + \|Ae\|^2) - \mathbb{E}_t[\phi_t(\sigma^2 + \|Ae\|^2)].$$

We claim that $u_* = \pm\sqrt{g_*/\|Ae\|^2}e$ achieves $L(u_*) = \Psi(g_*)$. Indeed, note that $g(u_*) = (u_*)^T G u_* = g_*/\|Ae\|^2 e^T G e = g_*/\|Ae\|^2 \cdot \|Ae\|^2 = g_*$ and

$$h(u_*)^2 = (e^T G u_*)^2 = \frac{g_*}{\|Ae\|^2}(e^T G e) = g_*\|Ae\|^2.$$

Hence

$$
\begin{aligned}
L(u_*) &= \tilde{c}g(u_*) - \tilde{\eta}h(u_*) - \mathbb{E}_t[\phi_t(g(u_*))] \\
&= \tilde{c}g_* - \tilde{\eta}g_*\|Ae\|^2 - \mathbb{E}_t[\phi_t(g_*)] \\
&= (\tilde{c} - \tilde{\eta}\|Ae\|^2)g_* - \mathbb{E}_t[\phi_t(g_*)] \\
&= \Delta g_* - \mathbb{E}_t[\phi_t(g_*)] \\
&= \Psi(g_*).
\end{aligned}
$$

The only other cases include $h_* = u_* + v$ for $v \in \ker(A)$ since these are precisely the equality cases of the generalized Cauchy-Schwarz inequality. Note that for such $h_*$,

$$g(h_*) = (h_*)^T G h_* = (u_* + v)^T G(u_* + v) = g(u_*) + 2v^T G u_* + v^T G v = g_*$$

and $h(u_* + v) = e^T G(u_* + v) = h(u_*)$. These cases are also global minimizers, which gives us the total set of globally optimal solutions:

$$\Theta_* := \left\{ \pm\sqrt{\frac{\sigma^2 + \|Ae\|^2}{\|Ae\|^2}}e \right\} + \ker(A).$$

Finally, we consider the Wasserstein distance bound. Note that for any $u \neq 0$, we have that

$$
\begin{aligned}
W_2^2(\mathcal{N}(0, ee^T), \mathcal{N}(0, uu^T)) &= \mathrm{tr}\left( ee^T + uu^T - 2\left( (uu^T)^{1/2}ee^T(uu^T)^{1/2} \right)^{1/2} \right) \\
&= 1 + \|u\|^2 - 2\mathrm{tr}\left[ \left( \frac{uu^T}{\|u\|}ee^T\frac{uu^T}{\|u\|} \right)^{1/2} \right] \\
&= 1 + \|u\|^2 - 2\mathrm{tr}\left[ \left( \frac{(e^T u)^2}{\|u\|^2}uu^T \right)^{1/2} \right] \\
&= 1 + \|u\|^2 - 2\mathrm{tr}\left[ \frac{|e^T u|}{\|u\|^2}uu^T \right] \\
&= 1 + \|u\|^2 - 2|e^T u|.
\end{aligned}
$$

In our case, note that $u_* = q \pm \lambda_* e$ where $q \in \ker(A)$ and $\lambda_* = \sqrt{1 + \sigma^2/\|Ae\|^2} > 1$. Since $e \in \mathrm{Im}(A^T)$, $e^T q = 0$. Hence $|e^T u_*| = \lambda_*$ and $\|u_*\|^2 = \|q\|^2 + \lambda_*^2$ where we used $\|e\| = 1$ in both equalities. Using the previous result and these properties of $u_*$, we see that for any $u_* \in \Theta_*$,

$$W_2^2(\mathcal{N}(0, ee^T), \mathcal{N}(0, u_* u_*^T)) = 1 + \|q\|^2 + \lambda_*^2 - 2\lambda_* = \|q\|^2 + (\lambda_* - 1)^2.$$

The minimum norm element of $\Theta_*$ is precisely given by either $\lambda_* e$ or $-\lambda_* e$. This is because $e$ and $q$ are orthogonal for any $q \in \ker(A)$, so for any $u_* \in \Theta_*$, $\|u_*\|^2 = \lambda_*^2\|e\|^2 + \|q\|^2 \geqslant \lambda_*^2 = \|\lambda_* e\|^2$. Taking $q = 0$ minimizes the norm of $u_* \in \Theta_*$. $\qquad\square$

### F.2.2 STRICT SADDLE PROPERTY

A natural question is whether the objective landscape is well-behaved for first-order optimization algorithms. We now analyze the rank-one case and show that the objective landscape has a strict saddle property, namely that all critical points are either global minimizers or strict saddles, i.e., points for which the gradient is zero, but the Hessian exhibits a negative eigenvalue, hence a descent direction.

**Theorem 3.** *Consider the setting of Theorem 1 with $r = 1$ and $E = e \in \mathbb{R}^d$ with unit norm. Then the objective $\mathcal{L}$ under the parameterization $\theta(u) = (u, u/\|u\|)$ satisfies the following: the set of critical points for $u \neq 0$ is precisely given by*

$$\Omega := \{\sqrt{1 + \sigma^2}e, -\sqrt{1 + \sigma^2}e\} \cup \Omega^\perp \text{ where } \Omega^\perp := \left\{ u \in \mathbb{R}^d : \langle u, e \rangle = 0, \|u\| = \sigma \right\}.$$

*Each point in $\{\sqrt{1+\sigma^2}e, -\sqrt{1+\sigma^2}e\}$ is a global minimizer. Moreover, each point in $\Omega^\perp$ is a strict saddle, meaning that for any $u \in \Omega^\perp$, the Hessian $\nabla^2 \mathcal{L}(u)$ is a strictly negative eigenvalue. Hence all local minima are global minima.*

*Proof of Theorem 3.* By Lemma 6, we have the simpler formula

$$\mathcal{L}(\theta) = C + L(u)$$

where

$$L(u) := \mathbb{E}_t \left[ \frac{1}{c_t^2} \|u\|^2 - \eta_t (e^T u)^2 - \frac{\|u\|^2}{a_t(a_t + \|u\|^2)} \right]$$

We will use the notation $a_l = \sigma_l^2$, $c_l = a_l + \sigma^2$, and $\eta_l = (2c_l + \|e\|^2)/(c_l^2(c_l + \|e\|^2)^2)$ for $l \sim \mathrm{Unif}[0, 1]$. Furthermore, let $\tilde{c} := \mathbb{E}_l[c_l^{-2}]$, $\tilde{\eta} := \mathbb{E}_l[\eta_l]$ and $\phi_l(g) := \frac{g}{a_l(a_l+g)^2}$ for $g > 0$. Further setting $g(u) = u^T u$, and $h(u) = e^T u$ our objective can be written as

$$L(u) := \tilde{c} u^T u - \tilde{\eta}(e^T u)^2 - \mathbb{E}_l[\phi_l(u^T u)].$$

An elementary calculation shows that $\phi_l'(g) = \frac{1}{(a_t+g)^2}$ and $\phi_l''(g) = -2(a_t + g)^{-3}$ for $g > 0$.

Note that

$$\nabla L(u) = 2 \left( (\tilde{c} - \mathbb{E}_t[\phi_t'(\|u\|^2)])u - \tilde{\eta} e^T u e \right) = 0$$

if and only if

$$(\tilde{c} - \mathbb{E}_t[\phi_t'(\|u\|^2)])u = \tilde{\eta} e^T u e.$$

Note that $u \in \{\pm\sqrt{1+\sigma^2}e\}$ satisfies this condition. On the other hand, consider a decomposition $u = \beta e + s$ where $s \perp e$. Set $F(r) = \tilde{c} - \mathbb{E}_t[\phi_t'(r)]$. Then $u = \beta e + s$ is a critical point if and only if

$$\beta(F(\beta^2 + \|s\|^2) - \tilde{\eta})e + F(\beta^2 + \|s\|^2)s = 0.$$

Suppose $\beta = 0$. Then this above equation requires $F(\|s\|^2)s = 0$. If $s \neq 0$, then we need $\|s\| = \sigma$ since $F(\sigma^2) = 0$. If $\beta \neq 0$, then we require $F(\beta^2 + \|s\|^2) = \tilde{\eta} > 0$ so $s = 0$ which means $F(\beta^2) = \tilde{\eta}$. This indeed satisfied by $\beta^2 = 1 + \sigma^2$, i.e., $\beta = \pm\sqrt{1+\sigma^2}$. Hence the critical points are

$$\left\{ \pm\sqrt{1+\sigma^2}e \right\} \bigcup \{u : u \perp e, \|u\| = \sigma\}.$$

Note that the Hessian is

$$\nabla^2 L(u) = 4F'(\|u\|^2)uu^T + 2F(\|u\|^2)I_d - 2\tilde{\eta} ee^T.$$

Note that for critical points $u$ such that $u \perp e$ and $\|u\| = \sigma$, we have

$$\nabla^2 L(u)e = 4F'(\|u\|^2)uu^T e + 2F(\|u\|^2)e - 2\tilde{\eta}e = 2(F(\sigma^2) - \tilde{\eta}) = -2\tilde{\eta}e$$

so $e$ is an eigenvector of $\nabla^2 L(u)$ with eigenvalue $-2\tilde{\eta} < 0$, which is strictly negative. Hence $e$ is a descent direction of the objective $L(u)$ so $u$ is a strict saddle point. If $u = \pm\sqrt{1+\sigma^2}e$, then

$$\nabla^2 L(u) = 4(1+\sigma^2)F'(1+\sigma^2)ee^T + 2F(1+\sigma^2)I_d - 2\tilde{\eta}ee^T = \left( 4(1+\sigma^2)F'(1+\sigma^2) - 2\tilde{\eta} \right) ee^T + 2\tilde{\eta}I_d.$$

Note that for directions orthogonal to $e$, $v \perp e$,

$$\nabla^2 L(u)v = 2\tilde{\eta}v \text{ with } 2\tilde{\eta} > 0.$$

Along $e$, we have

$$\nabla^2 L(u)e = (2\tilde{\eta} + 4(1+\sigma^2)F'(1+\sigma^2) - 2\tilde{\eta})e = 4(1+\sigma^2)F'(1+\sigma^2)e$$

where $4(1+\sigma^2)F'(1+\sigma^2) > 0$ so $\nabla^2 L(u)$ is in fact positive definite at $u = \pm\sqrt{1+\sigma^2}e$. Hence such points are global minimizers while other points are strict saddles. $\qquad\square$

### F.2.3 MULTIPLE NOISE SCALES

It is possible to extend Theorem 1 to the setting in which one has access to a dataset of varying noise levels $\sigma \sim p(\sigma)$ (as in the experiments discussed in Appendix D.4). An illustrative example is the case when we have noisy images $y = x + \sigma z$ where $\sigma$ comes from a finite set of noise levels $\{\sigma_1, \ldots, \sigma_K\}$. This can be modeled as $\sigma \sim p(\sigma) = \sum_{k=1}^{K} \pi_k \delta_{\sigma_k}$ where $\pi_k \geqslant 0$, $\sum_{k=1}^{K} \pi_k = 1$ and $\sigma_k > 0$ for each $k \in [K]$. We will minimize the following objective that also considers noise at different scales:

$$\tilde{\mathcal{L}}(\theta) := \mathbb{E}_{\sigma \sim p(\sigma)} \mathcal{L}(\theta) := \mathbb{E}_{\sigma \sim p(\sigma), t \sim \mathrm{Unif}(0,1)} \mathbb{E}_{x_t \sim p_{G_\theta}^{\sigma_t}} \left[ \left\| s_{\sigma, \sigma_t}(x_t) - \nabla \log p_{G_\theta}^{\sigma_t}(x_t) \right\|_2^2 \right]$$

We prove that, when the noise levels follow a general distribution $\sigma \sim p(\sigma)$, we can also characterize minimizers to the above loss. In particular, the scaling of the covariance now depends on the $p(\sigma)$ and $(\sigma_t)$. We will show through an example for a finite set of noise levels how the distilled generator can outperform any of the noisy teachers.

**Theorem 4.** *Consider the same setting as Theorem 1. Then for any distribution $p(\sigma)$ supported on $(0, \infty)$, there is exists a unique $\lambda^* > 0$ that depends on $(\sigma_t)$ and $p(\sigma)$ such that the global minimizers of $\tilde{\mathcal{L}}$ over $\Theta$, denoted by $\theta_\sigma^* := (U^*, V_\sigma^*)$, satisfy the following:*

$$U^* = EQ \text{ for some orthogonal matrix } Q \text{ and } (V_\sigma^*)^T V_\sigma^* = \lambda^* I_r. \tag{11}$$

*In particular, $\lambda^*$ is the unique solution to the equation*

$$\mathbb{E}_{p(\sigma), t} \left[ (\sigma^2 + \sigma_t^2 + 1)^{-2} \right] = \mathbb{E}_t \left[ (\lambda^* + \sigma_t^2)^{-2} \right].$$

*Hence the learned distilled distribution is given by $p_{G_{\theta_\sigma^*}} = \mathcal{N}(0, \lambda^* E E^T)$.*

*Proof.* The proof of Theorem turns out to be very similar to the proof of Theorem 1, with a particular modification of Lemma 5 needed to find the precise form of the minimizer. In particular, one can show using similar arguments that the optimal $U$ is given by $U^* = EQ$ for orthogonal $Q$. To find the form of $V$, one needs to minimize the following function over feasible $V$, which only depends on the eigenvalues of $V^T V$:

$$\sum_{i=1}^{r} F(\lambda_i(V^T V)) := \sum_{i=1}^{r} \mathbb{E}_{p(\sigma), t} \left[ \frac{\lambda_i(V^T V)}{(\sigma^2 + \sigma_t^2 + 1)^2} - \frac{\lambda_i(V^T V)}{\sigma_t^2(\lambda_i(V^T V) + \sigma_t^2)} \right]$$

where $F(u) := \mathbb{E}_{p(\sigma)}[f_\sigma(u)]$ and $f_\sigma$ is the function defined in Lemma 5. We claim that $F$ is strictly convex and has a unique minimizer $\lambda^*$. Note that for any $u > 0$, its derivatives are

$$F'(u) = \mathbb{E}_{p(\sigma), t} \left[ \frac{1}{(\sigma^2 + \sigma_t^2 + 1)^2} \right] - \mathbb{E}_t \left[ \frac{1}{(\sigma_t^2 + u)^2} \right] \text{ and}$$

$$F''(u) = \mathbb{E}_{p(\sigma), t} \left[ \frac{2}{(\sigma_t^2 + u)^3} \right] > 0.$$

Moreover, note that $u \mapsto \psi(u) := \mathbb{E}_{p(\sigma), t} \left[ (\sigma_t^2 + u)^{-2} \right]$ is strictly decreasing with $\lim_{u \to \infty} \psi(u) = 0$ and $\lim_{u \to 0^+} \psi(u) = \mathbb{E}_t[\sigma_t^{-4}] > \mathbb{E}_{p(\sigma), t} \left[ (\sigma^2 + \sigma_t^2 + 1)^{-2} \right]$. Note the last strict inequality follows by strict monotonicity of $\psi$. Hence there must exist a unique $\lambda^* \in (0, \infty)$ that minimizes $F$ satisfying $F'(\lambda^*) = 0$, i.e., $\mathbb{E}_{p(\sigma), t} \left[ (\sigma^2 + \sigma_t^2 + 1)^{-2} \right] = \mathbb{E}_t \left[ (\lambda^* + \sigma_t^2)^{-2} \right]$. Thus the above function for invertible $V^T V$ is minimized when the gram matrix of $V_\sigma^*$ has equal eigenvalues $\lambda_i((V_\sigma^*)^T V_\sigma^*) = \lambda^*$ for all $i \in [r]$. Since all of its eigenvalues are the same, by the Spectral Theorem, we must have that $(V_\sigma^*)^T V_\sigma^* = \lambda^* I_r$. □

**Example with a finite number of noise levels:** Consider the case when $p(\sigma) = \sum_{k=1}^{K} \pi_k \delta_{\sigma_k}$ for $\sigma_k > 0$ with $\pi_k \geqslant 0$ and $\sum_{k=1}^{K} \pi_k = 1$. Then $\lambda^*$ is the unique solution to

$$\sum_{k=1}^{K} \pi_k \mathbb{E}_t \left[ (\sigma_k^2 + \sigma_t^2 + 1)^{-2} \right] = \mathbb{E}_t \left[ (\lambda^* + \sigma_t^2)^{-2} \right].$$

Since the right-hand side is strictly decreasing with respect to $\lambda^*$, one can show that we always have

$$1 + \sigma_{\min}^2 \leqslant \lambda^* \leqslant 1 + \sigma_{\max}^2, \ \sigma_{\min} := \min_k \sigma_k, \ \sigma_{\max} := \max_k \sigma_k.$$

The precise value of $\lambda^*$ interestingly depends now on the noise schedule $(\sigma_t)$. In particular, when the noise schedule $\sigma_t$ has much smaller values, the right-hand side increases, requiring $\lambda^*$ to decrease to satisfy the equation. Likewise, when $\sigma_t$ focuses on larger noise levels, the right-hand side goes down, requiring a larger $\lambda^*$ for the equation to be satisfied.

One can also give a mathematical condition on when the distilled distribution is closer in Wasserstein distance to each of the noisy distributions to the ground-truth. In particular, consider the low-rank regime $r \ll d$. Then if $\lambda^*$ satisfies

$$\lambda^* < \min_{k \in [K]} \left[ 1 + \sqrt{\left( \sqrt{1 + \sigma_k^2} - 1 \right)^2 + \frac{d - r}{r} \sigma_k^2} \right]^2 \approx 1 + \frac{d}{r} \sigma_{\min}^2$$

then

$$W_2^2(p_{G_{\theta_\sigma^*}}, p_X) < \min_{k \in [K]} W_2^2(p_{Y, \sigma_k}, p_X),$$

i.e., the distilled generator is closer to the ground-truth distribution than every noisy teacher. Note that if the difference between $\sigma_{\max}$ and $\sigma_{\min}$ is not too large and the data is low-rank enough $d \gg r$, then this can be satisfied. For example, if the largest noise level is less than a multiple of the smallest noise level $\sigma_{\max} < \sqrt{d/r} \cdot \sigma_{\min}$, then this would ensure $\lambda^* \leqslant 1 + \sigma_{\max}^2 < 1 + \frac{d}{r}\sigma_{\min}^2$. As an example, suppose $d/r = 10$. Then as long as $\sigma_{\max} < 3\sigma_{\min}$, the above condition would be satisfied.

## F.3 GENERAL GUARANTEES FOR DISTILLATION

Following up on our linear analysis, we aim to derive more general guarantees showing that, under capacity and optimization assumptions on the teacher and generator, we can give bounds on the distilled generator's performance. This will be for more general corruptions and then we will state a result more tailored to denoising after where we can give conditions on when the distilled model improves upon the noisy distribution.

To state our results, we set the notation. Let $\mathcal{F}(p||q) := \mathbb{E}_p[\|\nabla \log p(x) - \nabla \log q(x)\|_2^2]$ denote the Fisher divergence between $p$ and $q$. For a density $p$, we let $dp = p(x)dx = pdx$. Let $\chi^2(p||q) = \int (p/q - 1)^2 dq$ denote the chi-square divergence between $p$ and $q$. For notational simplicity, we let $\lesssim$ denote an inequality up to absolute constants so that $a \lesssim b$ if there exists an absolute constant $C > 0$ such that $a \leqslant Cb$. We let $y = Ax + \sigma\epsilon$ with $\sigma > 0$ and define $p_Y := \mathcal{T}[p_X] := A_\sharp p_X * \mathcal{N}(0, \sigma^2 I_m)$. We will use the notation

$$x_t = x + \sigma_t \epsilon, y_t = y + \sigma_t \epsilon'$$

with marginals $p_{X,t}, p_{Y,t}$. For the generator $G_\theta$ with clean law $p_\theta$ and parameters $\theta \in \Theta$, its measurement law is $p_{\theta,Y} := \mathcal{T}[p_\theta]$ with noise-convolved marginals $p_{\theta,Y,t}$. We consider distilling by minimizing

$$\mathcal{L}_{\mathrm{distill}}(\theta) := \mathbb{E}_{t, \tilde{y}_t \sim p_{\theta,Y,t}} \|f_\phi(\tilde{y}_t, t) - f_\psi(\tilde{y}_t, t)\|_2^2$$

where $f_\phi$ is the teacher and $f_\psi$ is the fake diffusion model. For each $t$, let $s_{Y,t}$ and $s_{\theta,Y,t}$ denote the scores of $p_{Y,t}$ and $p_{\theta,Y,t}$, respectively. Throughout, we will assume sufficient regularity of the densities so that all scores $\nabla \log p$ and gradients $\nabla p$ are well-defined, including those induced by $p_\theta$ for $\theta \in \Theta$. Moreover, put

$$\delta_\phi(y, t) := f_\phi(y, t) - s_{Y,t}(y) \qquad \delta_\psi(\tilde{y}, t) := f_\psi(\tilde{y}, t) - s_{\theta,Y,t}(\tilde{y})$$

$$\varepsilon_{\phi,2}^2 := \sup_t \mathbb{E}_{p_{Y,t}} \|\delta_\phi(y_t, t)\|_2^2, \ \varepsilon_{\phi,4}^2 := \sup_t \left( \mathbb{E}_{p_{Y,t}} \|\delta_\phi(y_t, t)\|_2^4 \right)^{1/2}.$$

We will assume we have optimized our parameter $\theta$ to a point $\hat{\theta}$. Define the local fake-diffusion error at $\hat{\theta}$:

$$\varepsilon_\psi^2(\hat{\theta}) := \sup_t \mathbb{E}_{p_{\hat{\theta},Y,t}} \|\delta_\psi(\tilde{y}_t, t)\|_2^2.$$

Finally, let $\Delta(\hat{\theta})$ denote the local density ratio

$$\Delta(\hat{\theta}) := \sup_t \sup_y w_t(y), \ w_t(y) := p_{\hat{\theta},Y,t}(y)/p_{Y,t}(y).$$

We will make the following assumptions on the data and learned parameters:

**Assumption 5** (Data distribution). *Suppose that the corrupted data distribution $p_{Y,t}$ satisfies a uniform Poincaré-like inequality in the sense that there exists a $\lambda_0 > 0$ such that*

$$\chi^2(p||p_{Y,t}) \leqslant \lambda_0^{-1} \int \|\nabla(p/p_{Y,t})\|^2 dp_{Y,t}, \ \forall t, p \in \{p_{\theta,Y,t}\}_{\theta \in \Theta}.$$

**Assumption 6** (Capacity and local near-optimality). *There exists a $\theta^*$ such that $p_{\theta^*,Y} = p_Y$ and $\hat{\theta}$ satisfies the following for some $\varepsilon_{\mathrm{opt}} \geqslant 0$:*

$$\mathcal{L}_{\mathrm{distill}}(\hat{\theta}) \leqslant \mathcal{L}_{\mathrm{distill}}(\theta^*) + \varepsilon_{\mathrm{opt}}.$$

*Moreover, the fake diffusion network parameters $\psi$ satisfies $\mathbb{E}_{t,p_{Y,t}}\|f_\phi(y_t,t) - f_\psi(y_t,t)\|_2^2 \leqslant \mathbb{E}_{t,p_{Y,t}}\|f_\phi(y_t,t) - s_{Y,t}\|_2^2$. We note that this can be relaxed to an upper bound up to a constant.*

We will aim to prove the following:

**Theorem 5.** *Under Assumptions 5 and 6, we have that the following holds:*

1. *(**general bound**) the learned distilled distribution in measurement space satisfies*

$$\mathbb{E}_t \mathcal{F}(p_{\hat{\theta},Y,t}||p_{Y,t}) \lesssim \left( \varepsilon_{\phi,4}^2 \sqrt{\frac{\Delta(\hat{\theta})}{\lambda_0}} + \sqrt{\varepsilon_{\phi,4}^4 \frac{\Delta(\hat{\theta})}{\lambda_0} + \varepsilon_{\phi,2}^2 + \varepsilon_\psi^2(\hat{\theta}) + \varepsilon_{\mathrm{opt}}} \right)^2;$$

2. *(**measurement injectivity**) if the measurement operator $\mathcal{T} : p \mapsto A_\sharp p * \mathcal{N}(0, \sigma^2 I_m)$ satisfies $\mathbb{E}_t \mathcal{F}(p_{\hat{\theta},t}||p_{X,t}) \leqslant \hat{\kappa} \cdot \mathbb{E}_t \mathcal{F}(p_{\hat{\theta},Y,t}||p_{Y,t})$ for some $\hat{\kappa} > 0$, then we have that the learned distilled distribution in image space satisfies*

$$\mathbb{E}_t \mathcal{F}(p_{\hat{\theta},t}||p_{X,t}) \lesssim \hat{\kappa} \left( \varepsilon_{\phi,4}^2 \sqrt{\frac{\Delta(\hat{\theta})}{\lambda_0}} + \sqrt{4\varepsilon_{\phi,4}^4 \frac{\Delta(\hat{\theta})}{\lambda_0} + \varepsilon_{\phi,2}^2 + \varepsilon_\psi^2(\hat{\theta}) + \varepsilon_{\mathrm{opt}}} \right)^2;$$

**Discussion.** This result guarantees a bound on the Fisher divergence between the distilled generator's measurement distribution and the true measurement distribution. The key idea is that minimizing the distillation loss encourages the generator to learn an image distribution whose *induced* measurements are close the true measurements. Lemma 8 more explicitly connects the distillation loss to the reverse Fisher divergence between the measurement distributions. The second key component is the second bound, which shows that if the corruption operator satisfies an injectivity property over the data, then we can transfer this bound to the distilled distribution in image space. Hence distillation has the potential to succeed when 1) the distilled generator learns to create images such that, when corrupted further, look like the measurements and 2) the corruption operator is stable or injective over our distributions. We show in the following Corollary that in the instructive case of denoising, the corruption operator is stable and we can give a condition on when distillation can improve over the noisy distribution.

**Corollary 1** (Improvement in denoising). *Under the setting of Theorem 5, when $A = I$, we have that the measurement injectivity condition holds for some $\hat{\kappa} > 0$ that depends on the noise schedule and for $\varepsilon_{\phi,2}, \varepsilon_{\phi,4}, \varepsilon_\psi(\hat{\theta}), \varepsilon_{\mathrm{opt}}$ sufficiently small, we have that the distilled distribution improves upon the noisy distribution*

$$\mathbb{E}_t \mathcal{F}(p_{\hat{\theta},t}||p_{X,t}) < \mathbb{E}_t \mathcal{F}(p_{Y,t}||p_{X,t}).$$

To prove these results, we require a number of technical lemmas. The first is a change of measure result that will be useful in transferring expectations.

**Lemma 7.** *For fixed $t$, let $e_\phi(y,t) := \|\delta_\phi(y,t)\|_2^2$. Then we have that*

$$\mathbb{E}_{p_{\theta,Y,t}} e_\phi(y,t) \leqslant \varepsilon_{\phi,2}^2 + \varepsilon_{\phi,4}^2 \sqrt{\chi^2(p_{\theta,Y,t}||p_{Y,t})}.$$

*Proof of Lemma 7.* Suppose we set $q = p_{Y,t}$, $p = p_{\theta,Y,t}$, and $w = dp/dq$. Recall that for a density $q$, the induced $L^2(q)$ norm is given by $\|f\|_{L^2(q)}^2 = \int f^2 dq$. Then we have via an application of Cauchy-Schwarz that

$$
\begin{aligned}
|\mathbb{E}_p e_\phi(y,t) - \mathbb{E}_q e_\phi(y,t)| &= \left| \int (w(y) - 1) e_\phi(y,t) dq \right| \\
&\leqslant \|w - 1\|_{L^2(q)} \|e_\phi\|_{L^2(q)} \\
&= \sqrt{\chi^2(p\|q)} \cdot \left( \mathbb{E}_q e_\phi^2(y,t) \right)^{1/2}.
\end{aligned}
$$

The result follows by using the definitions of $\varepsilon_{\phi,2}$ and $\varepsilon_{\phi,4}$. $\qquad \square$

The next Lemma is crucial, in that it shows how the distillation loss from SiD Zhou et al. (2024) encourages the generator $G_\theta$ to produce images whose measurements match the distribution of the true measurements.

**Lemma 8.** *For $\hat{\theta}$, define $\Gamma(\hat{\theta}) := \mathbb{E}_t \left[ \varepsilon_{\phi,4}^2 \sqrt{\chi^2(p_{\hat{\theta},Y,t}\|p_{Y,t})} \right]$. Then we have that the distillation loss satisfies*

$$
\frac{1}{2} \mathbb{E}_t \mathcal{F}(p_{\hat{\theta},Y,t}\|p_{Y,t}) - 2\varepsilon_\psi^2(\hat{\theta}) - 2\varepsilon_{\phi,2}^2 - 2\Gamma(\hat{\theta}) \leqslant \mathcal{L}_{\text{distill}}(\hat{\theta})
$$

*Proof of Lemma 8.* We first consider the decomposition

$$
f_\phi - f_\psi = (s_{Y,t} - s_{\hat{\theta},Y,t}) + \delta_\phi - \delta_\psi.
$$

Using $\|x + y - w\|_2^2 \leqslant 3(\|x\|_2^2 + \|y\|_2^2 + \|w\|_2^2)$ for $x = s_{Y,t} - s_{\hat{\theta},Y,t}$, $y = \delta_\phi$ and $w = \delta_\psi$ and taking expectations, we have that for fixed $t$,

$$
\begin{aligned}
\mathbb{E}_{p_{\hat{\theta},Y,t}} \|f_\phi(\tilde{y}_t,t) - f_\psi(\tilde{y}_t,t)\|_2^2 &\leqslant 3\mathbb{E}_{p_{\hat{\theta},Y,t}} \|s_{Y,t}(\tilde{y}_t,t) - s_{\hat{\theta},Y,t}(\tilde{y}_t,t)\|_2^2 + 3\mathbb{E}_{p_{\hat{\theta},Y,t}} e_\phi(\tilde{y}_t,t) + 3\mathbb{E}_{p_{\hat{\theta},Y,t}} \|\delta_\psi(\tilde{y}_t,t)\|_2^2 \\
&\leqslant 3\mathbb{E}_{p_{\hat{\theta},Y,t}} \|s_{Y,t}(\tilde{y}_t,t) - s_{\hat{\theta},Y,t}(\tilde{y}_t,t)\|_2^2 + 3\left( \varepsilon_{\phi,2}^2 + \varepsilon_{\phi,4}^2 \sqrt{\chi^2(p_{\hat{\theta},Y,t}\|p_{Y,t})} \right) + 3\varepsilon_\psi^2(\hat{\theta})
\end{aligned}
$$

where the last line follows by Lemma 7 and the definition of $\varepsilon_\psi^2(\hat{\theta})$. Taking an expectation over $t$ yields

$$
\mathcal{L}_{\text{distill}}(\hat{\theta}) \leqslant 3\mathbb{E}_t \mathcal{F}(p_{\hat{\theta},Y,t}\|p_{Y,t}) + 3\varepsilon_{\phi,2}^2 + 3\Gamma(\hat{\theta}) + 3\varepsilon_\psi^2(\hat{\theta}).
$$

The lower bound holds by using the bound $\|x + y - w\|_2^2 \geqslant \frac{1}{2}\|x\|_2^2 - 2(\|y\|_2^2 + \|w\|_2^2)$, applying the same bounds, and taking expectations. Note that this bound holds because for any $z$,

$$
\|x + z\|_2^2 = \|x\|_2^2 + 2\langle x, z \rangle + \|z\|_2^2.
$$

Recall Young's inequality: $|\langle x, z \rangle| \leqslant \frac{\alpha^2}{2}\|x\|_2^2 + \frac{1}{2\alpha^2}\|z\|_2^2$ for $\alpha > 0$. Hence we have the lower bound

$$
\|x + z\|_2^2 \geqslant \|x\|_2^2 - \alpha^2\|x\|_2^2 - \alpha^{-2}\|z\|_2^2 + \|z\|_2^2.
$$

Choosing $\alpha^2 = 1/2$, setting $z = y - w$, and using $\|y - w\|_2^2 \leqslant 2(\|y\|_2^2 + \|w\|_2^2)$ yields the desired inequality. $\qquad \square$

An additional ingredient we need is control over the $\chi^2$ distance and relating it to the Fisher divergence. For that, we need the following Lemma.

**Lemma 9.** *Under Assumption 5, we have that*

$$
\Gamma(\hat{\theta}) \leqslant \varepsilon_{\phi,4}^2 \sqrt{\frac{\Delta(\hat{\theta})}{\lambda_0}} \cdot \left( \mathbb{E}_t \mathcal{F}(p_{\hat{\theta},Y,t}\|p_{Y,t}) \right)^{1/2}.
$$

*Proof of Lemma 9.* We first fix $t$ and set $q = p_{Y,t}$, $p = p_{\hat{\theta},Y,t}$ and $w = p/q$. Then Assumption 5 yields

$$
\chi^2(p\|q) \leqslant \lambda_0^{-1} \int \|\nabla w\|_2^2 dq = \lambda_0^{-1} \int w \frac{\|\nabla w\|_2^2}{w} dq \leqslant \frac{\Delta(\hat{\theta})}{\lambda_0} \int \frac{\|\nabla w\|_2^2}{w} dq = \frac{\Delta(\hat{\theta})}{\lambda_0} \mathcal{F}(p\|q)
$$

where the first line follows by assumption, the second inequality follows by definition of $\Delta(\hat{\theta})$ and the last equality follows by definition of the Fisher divergence. Ineed, for the last equality, note that if $w = p/q$, then $\nabla w = \nabla(p/q) = \frac{p}{q}(\nabla \log p - \nabla \log q) = w(s_p - s_q)$ where $s_p$ and $s_q$ are the scores of $p$ and $q$, respectively. This ensures that

$$\int \frac{\|\nabla w\|_2^2}{w} dq = \int w\|s_p - s_q\|^2 dq = \int \|s_p - s_q\|^2 p dx = \mathcal{F}(p\|q).$$

Taking square roots and applying an expectation over $t$ along with Jensen's inequality for the concave map $v \mapsto \sqrt{v}$ yields the desired bound. □

Armed with these technical results, we now prove Theorem 5.

*Proof of Theorem 5.* First, recall that by Lemma 8, we have the lower bound

$$\mathcal{L}_{\text{distill}}(\hat{\theta}) \geqslant \frac{1}{2}\mathbb{E}_t \mathcal{F}(p_{\hat{\theta},Y,t}\|p_{Y,t}) - 2\varepsilon_\psi^2(\hat{\theta}) - 2\varepsilon_{\phi,2}^2 - 2\Gamma(\hat{\theta}). \tag{12}$$

Moreover, by definition of $\theta^*$, we have that

$$\mathcal{L}_{\text{distill}}(\theta^*) = \mathbb{E}_{t,p_{\theta^*,Y,t}}\|f_\phi - f_\psi\|_2^2 = \mathbb{E}_{t,p_{Y,t}}\|f_\phi - f_\psi\|_2^2 \leqslant \mathbb{E}_{t,p_{Y,t}}\|f_\phi - s_{Y,t}\|_2^2 \leqslant \varepsilon_{\phi,2}^2$$

where we used $p_{\theta^*,Y} = p_Y$ in the second equality and the assumption on $\psi$ in the second-to-last inequality. Then note that by Assumption 6, we have that

$$\mathcal{L}_{\text{distill}}(\hat{\theta}) \leqslant \mathcal{L}_{\text{distill}}(\theta^*) + \varepsilon_{\text{opt}} \leqslant \varepsilon_{\phi,2}^2 + \varepsilon_{\text{opt}}. \tag{13}$$

Combining 12 and 13 along with Lemma 9 yields the quadratic inequality $\frac{1}{2}X \leqslant A + B\sqrt{X}$ where we have set $X := \mathbb{E}_t\mathcal{F}(p_{\hat{\theta},Y,t}\|p_{Y,t})$, $A := 3\varepsilon_{\phi,2}^2 + \varepsilon_{\text{opt}} + 2\varepsilon_\psi^2(\hat{\theta})$ and $B := 2\varepsilon_{\phi,4}^2\sqrt{\frac{\Delta(\hat{\theta})}{\lambda_0}}$. Solving this inequality, we have that $X \leqslant \left(B + \sqrt{B^2 + 2A}\right)^2$. Substituting the values of $X$, $A$, and $B$ yields the desired result.

□

Finally, we show the denoising Corollary.

*Proof of Corollary 1.* The Corollary is a consequence of the fact that when $A = I$, we have that $p_{Y,t} = \mathcal{T}[p_X] * \mathcal{N}(0, \sigma_t^2 I) = p_X * \mathcal{N}(0, \sigma^2 I) * \mathcal{N}(0, \sigma_t^2 I) = p_X * \mathcal{N}(0, (\sigma^2 + \sigma_t^2)I)$. Hence if we define the time-shift map $\tau(t)$ by $\sigma_{\tau(t)}^2 = \sigma_t^2 + \sigma^2$, we have that for any $\theta$ and $t$, $\mathcal{F}(p_{\theta,Y,t}\|p_{Y,t}) = \mathcal{F}(p_{\theta,\tau(t)}\|p_{X,\tau(t)})$. Denote the distribution of $t \sim \rho$ where $\rho$ has support in $\tau([t_{\min}, t_{\max}])$ and set

$$\hat{\kappa} := \sup_{t \in [t_{\min}, t_{\max}]} \frac{\rho(\tau(t))\tau'(t)}{\rho(t)} \in (0, \infty).$$

Then we have that

$$\mathbb{E}_t\mathcal{F}(p_{\hat{\theta},t}\|p_{X,t}) \leqslant \hat{\kappa} \cdot \mathbb{E}_t\mathcal{F}(p_{\hat{\theta},\tau(t)}\|p_{X,\tau(t)}) = \hat{\kappa} \cdot \mathbb{E}_t\mathcal{F}(p_{\hat{\theta},Y,t}\|p_{Y,t}).$$

Define $\Delta_\sigma := \mathbb{E}_t\mathcal{F}(p_{Y,t}\|p_{X,t})$. Then using the measurement injectivity bound in Theorem 5, there exists a universal constant $C$ such that if

$$\hat{\kappa} \cdot \left(\varepsilon_{\phi,4}^2\sqrt{\frac{\Delta(\hat{\theta})}{\lambda_0}} + \sqrt{\varepsilon_{\phi,4}^4\frac{\Delta(\hat{\theta})}{\lambda_0} + \varepsilon_{\phi,2}^2 + \varepsilon_\psi^2(\hat{\theta}) + \varepsilon_{\text{opt}}}\right)^2 \leqslant \Delta_\sigma/C$$

then we have

$$\mathbb{E}_t\mathcal{F}(p_{\hat{\theta},t}\|p_{X,t}) < \mathbb{E}_t\mathcal{F}(p_{Y,t}\|p_{X,t}).$$

□

**Connection to mode-seeking.** The previous theory connects to mode-seeking behavior in the following way. In particular, under the setting of our theory, obtaining a small Fisher divergence encourages small reverse KL-divergence. To see this, recall that the KL-divergence and $\chi^2$-divergence satisfy $D_{\mathrm{KL}}(p\|q) \leqslant \chi^2(p\|q)$ when $p \ll q$ (which follows by the inequality $\log t \leqslant t - 1$ for $t > 0$). Following the proof of Lemma 9, this shows that we have the inequality

$$D_{\mathrm{KL}}(p_{\theta,Y,t}\|p_{Y,t}) \leqslant \chi^2(p_{\theta,Y,t}\|p_{Y,t}) \leqslant \frac{\Delta}{\lambda_0}\mathcal{F}(p_{\theta,Y,t}\|p_{Y,t})$$

where $\Delta = \Delta(\theta)$ is the density ratio of $p_{\theta,Y,t}$ and $p_{Y,t}$ and $\lambda_0$ is the constant in Assumption 5. Hence if one can minimize the Fisher divergence and the ratio $\Delta$, this will also encourage smaller reverse KL-divergence (mode-seeking behavior). A further theoretical investigation of this connection is an important direction for future work.

## G    VISUALIZATION FOR OVERESTIMATED DATA NOISE LEVEL

In Section 4.5, we discussed handling an *unknown* data-noise level $\sigma$ and showed that *slight overestimation* preserves strong performance, consistent with blind inverse-problem solvers Zhang et al. (2017; 2018). Here, we provide an intuitive 2D toy example demonstrating that modest overestimation yields clean generations, whereas *underestimation* produces noticeably noisier samples.

We construct a noisy training set with ground-truth $\sigma = 0.05$. During both pretraining and distillation, we vary the assumed noise $\hat{\sigma}$ to represent *underestimation* ($\hat{\sigma} < \sigma$), *accurate* estimation ($\hat{\sigma} = \sigma$), and *overestimation* ($\hat{\sigma} > \sigma$). As shown in Fig. 14, a slight overestimation increases effective regularization, helping generated samples better adhere to the data manifold.

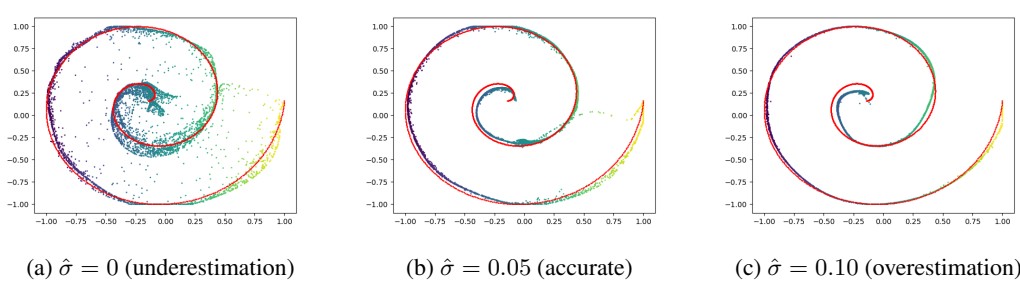

(a) $\hat{\sigma} = 0$ (underestimation)      (b) $\hat{\sigma} = 0.05$ (accurate)      (c) $\hat{\sigma} = 0.10$ (overestimation)

Figure 14: **Effect of noise-level misspecification in a 2D toy example.** We train with a noisy dataset at $\sigma = 0.05$ and vary the assumed noise $\hat{\sigma}$ during pretraining and distillation. Underestimation ($\hat{\sigma} < \sigma$) yields noisy generations; accurate estimation recovers the target structure; and slight overestimation ($\hat{\sigma} > \sigma$) acts as additional regularization, improving adherence to the data manifold. See also Section 4.5 for ablations.

## H    PHASE I: VARIANTS OF PRETRAINING DETAILS

In Section 3.1, we introduced four variants of diffusion model pretraining methods. Here, we provide the training details for each of these variants.

### H.1    STANDARD DIFFUSION

Standard Diffusion aims to learn the distribution of $y^{(i)} = \mathcal{A}(x^{(i)}) + \sigma\varepsilon$ directly. To do this, we train the diffusion model on the corrupted dataset $\{y^{(i)}\}_{i=1}^{N}$. The training objective is given by:

$$\mathcal{L}_{\mathrm{SD}} = \mathbb{E}_{\sigma_t,y,\varepsilon}\left[\lambda(t)\|f_\phi(y + \sigma_t\varepsilon, t) - y\|_2^2\right], \tag{14}$$

where $\lambda(t)$ is a time-dependent weighting function and $\sigma_t$ is sampled from a predefined noise schedule. We are using the EDM schedule for $\lambda(t)$ and $\sigma_t$ same as in Karras et al. (2022).

The detailed training procedure is summarized in Algorithm 2.

---

**Algorithm 2** Standard Diffusion Training

---

1: **procedure** STANDARD-DIFFUSION($\{y^{(i)}\}_{i=1}^N, \sigma, p(\sigma_t), K$)
2:     **for** $k = 1$ to $K$ **do**
3:         Sample a batch $y \sim \{y^{(i)}\}_{i=1}^N$
4:         Sample noise level $\sigma_t \sim p(\sigma_t)$
5:         Sample noise $\varepsilon \sim \mathcal{N}(0, I_d)$
6:         Construct noisy input: $y_t = y + \sigma_t \cdot \varepsilon$
7:         Update parameters $\phi$ via gradient descent on $\mathcal{L}_{\text{SD}}$ (Eq. 14)
8:     **end for**
9:     **return** Trained diffusion model $f_\phi$
10: **end procedure**

---

### H.2 DIFFUSION FOR DENOISING

Diffusion for denoising aims to learn the distribution of $\mathcal{A}(x^{(i)})$ from noisy observations $y^{(i)} = \mathcal{A}(x^{(i)}) + \sigma\varepsilon$, where $\sigma$ is a known (constant) noise level. The objective is to mitigate the impact of additive noise during training. Related denoising strategies have been explored in Daras et al. (2025).

The Tweedie adjustment is compatible with more advanced diffusion training techniques (Appendix H.3 and Appendix H.4), tailored for random inpainting and Fourier-space inpainting, respectively.

**Training loss.** We use the following objective:

$$\mathcal{L}_{\text{D}} = \mathbb{E}_{\sigma_t, y, \varepsilon} \left[ \left\| \frac{\sigma_t^2 - \sigma^2}{\sigma_t^2} f_\phi(y_t, t) + \frac{\sigma^2}{\sigma_t^2} y_t - y \right\|_2^2 \right], \tag{15}$$

where $\sigma_t$ is sampled from the diffusion noise schedule and clipped so that $\sigma_t \geq \sigma$, and $y_t := y + \sqrt{\sigma_t^2 - \sigma^2}\, \varepsilon$.

This formulation induces differences in both training and inference. Full details are given in Algorithm 3 and Algorithm 4. For sampling, we denote by **Teacher-Full** the standard EDM sampling from $\sigma_{\max}$ down to 0, and by **Teacher-Truncated** the EDM sampling truncated at the data noise level $\sigma$. See Algorithm 4 for details.

---

**Algorithm 3** Diffusion for Denoising (Training)

---

1: **procedure** DIFFUSION-FOR-DENOISING-TRAINING($\{y^{(i)}\}_{i=1}^N, \sigma, p(\sigma_t), K$)
2:     **for** $k = 1$ to $K$ **do**
3:         Sample a minibatch $y \sim \{y^{(i)}\}_{i=1}^N, \sigma_t \sim p(\sigma_t), \varepsilon \sim \mathcal{N}(0, I_d)$
4:         $\sigma_t \leftarrow \max\{\sigma, \sigma_t\}$                                    ▷ Clip noise level
5:         $y_t \leftarrow y + \sqrt{\sigma_t^2 - \sigma^2}\, \varepsilon$
6:         Update $f_\phi$ by descending $\nabla_\phi \mathcal{L}_{\text{D}}$ in Eq. 15
7:     **end for**
8:     **return** trained diffusion model $f_\phi$
9: **end procedure**

---

---

**Algorithm 4** Diffusion for Denoising (Sampling)

---

1: **procedure** DIFFUSION-FOR-DENOISING-SAMPLING($f_\phi, \sigma, \{\sigma_t\}_{t=0}^T$)
2:     Sample $x_T \sim \mathcal{N}(0, \sigma_T^2 I_d)$
3:     **for** $t = T, T-1, \ldots, 1$ **do**
4:         $\hat{x}_0 \leftarrow f_\phi(x_t, t)$
5:         **if** truncation enabled $\wedge \sigma_{t-1} < \sigma$ **then**
6:             **return** $\hat{x}_0$                    $\triangleright$ Teacher-Truncated
7:         **end if**
8:         $x_{t-1} \leftarrow x_t - \frac{\sigma_t - \sigma_{t-1}}{\sigma_t}\left(x_t - \hat{x}_0\right)$          $\triangleright$ EDM-style update
9:     **end for**
10:     **return** $\hat{x}_0$                              $\triangleright$ Teacher-Full
11: **end procedure**

---

### H.3 DIFFUSION FOR RANDOM INPAINTING TASK

Daras et al. Daras et al. (2023a) introduced a diffusion training objective specifically designed for the random inpainting task. The goal is to learn the underlying clean data distribution $p_X$ from partial observations of the form $y^{(i)} = Mx^{(i)}$, where $M$ is a binary inpainting mask applied to the image. To introduce further stochasticity, a secondary corruption mask $\tilde{M}$ is applied during training.

The training loss for random inpainting is given by:

$$\mathcal{L}_{\text{RI}} = \mathbb{E}_{\sigma_t, y, \varepsilon}\left[\left\|M\left(f_\phi(\tilde{M}, \tilde{M}y_t, t) - y\right)\right\|_2^2\right], \tag{16}$$

where $y_t = y + \sigma_t \varepsilon$, and $f_\phi$ is conditioned on both the corrupted observation and the masking pattern. All training schedules and hyperparameters follow the same configuration as in the original paper Daras et al. (2023b).

The full training procedure is outlined in Algorithm 5.

---

**Algorithm 5** Diffusion Training for Random Inpainting

---

1: **procedure** DIFFUSION-INPAINTING-TRAINING($\{y^{(i)}\}_{i=1}^N, M, p(\sigma_t), K$)
2:     **for** $k = 1$ to $K$ **do**
3:         Sample batch $y \sim \{y^{(i)}\}_{i=1}^N$
4:         Sample noise level $\sigma_t \sim p(\sigma_t)$
5:         Sample noise $\varepsilon \sim \mathcal{N}(0, I_d)$
6:         Sample a further corruption mask $\tilde{M}$ conditioned on $M$
7:         Compute $y_t \leftarrow y + \sigma_t \cdot \varepsilon$
8:         Update $f_\phi$ using gradient descent on $\mathcal{L}_{\text{RI}}$ in Eq. 16
9:     **end for**
10:     **return** Trained diffusion model $f_\phi$
11: **end procedure**

---

### H.4 DIFFUSION TRAINING FOR FOURIER SPACE INPAINTING

Aali et al. (2025) introduced a diffusion training objective specifically designed for the multi-coil MRI on Fourier Space. The goal is to learn the underlying clean data distribution $p_X$ from

$$y = (\underbrace{\sum_{i=1}^{N_c} S_i^H \mathcal{F}^{-1} M \mathcal{F} S_i}_{\mathcal{A}})\, x,$$

where $S_i$ denotes the coil sensitivity profile of the $i$-th coil, $\mathcal{F}$ is the Fourier transform, and $M$ is the masking operator in Fourier space. The further corrupted observation would be

$$\tilde{y} = \underbrace{\left(\sum_{i=1}^{N_c} S_i^H \mathcal{F}^{-1} \tilde{M} \mathcal{F} S_i\right)}_{\tilde{\mathcal{A}}} x.$$

The training loss for multi-coil MRI is given by:

$$\mathcal{L}_{\text{FS}} = \mathbb{E}_{\sigma_t, y, \varepsilon}\left[\left\|\mathcal{A}(f_\phi(\tilde{y}_t, \tilde{M}, t) - y)\right\|_2^2\right], \tag{17}$$

where $y_t = y + \sigma_t \varepsilon$, and $f_\phi$ is conditioned on both the corrupted observation and the masking pattern.

The full training procedure is outlined in Algorithm 6. All training schedules and hyperparameters follow the same configuration as in the original paper Aali et al. (2025).

---

**Algorithm 6** Diffusion Training for Fourier Space Inpainting

---

1: **procedure** DIFFUSION-FSINPAINTING-TRAINING($\{y^{(i)}\}_{i=1}^N$, $M$, $p(\sigma_t)$, $K$, $S$)
2:     **for** $k = 1$ to $K$ **do**
3:         Sample batch $y \sim \{y^{(i)}\}_{i=1}^N$
4:         Sample noise level $\sigma_t \sim p(\sigma_t)$
5:         Sample noise $\varepsilon \sim \mathcal{N}(0, I_d)$
6:         Sample a further corruption mask $\tilde{M}$ conditioned on $M$
7:         Compute $y_t \leftarrow y + \sigma_t \cdot \varepsilon$
8:         Update $f_\phi$ using gradient descent on $\mathcal{L}_{\text{RI}}$ in Eq. 17
9:     **end for**
10:     **return** Trained diffusion model $f_\phi$
11: **end procedure**

---

## I   PHASE II: DISTILLATION

In Section 2.3, we introduced the SiD generator loss (Eq. 5). The SiD objective admits additional design choices, as discussed in the original paper Zhou et al. (2024). For completeness, we present the exact generator-loss formulation used in our implementation and defer details such as time-step scheduling and hyperparameter settings to the cited work.

**SiD generator loss.** Let $x_g = G_\theta(z)$ and $x_t = x_g + \sigma_t \varepsilon$ with $z, \varepsilon \sim \mathcal{N}(0, I_d)$. The loss is

$$\mathcal{L}_{\text{SiD}}(w_g) = \mathbb{E}_{z, t, \varepsilon}\Big[(1 - \alpha)\,\lambda(t)\left\|f_\psi(x_t, t) - f_\phi(x_t, t)\right\|_2^2$$
$$+ \lambda(t)\left(f_\phi(x_t, t) - f_\psi(x_t, t)\right)^\top\left(f_\psi(x_t, t) - x_g\right)\Big], \tag{18}$$

where $f_\phi$ is the teacher (pretrained diffusion model), $f_\psi$ is the fake diffusion model, $\lambda(t)$ is a time-dependent weight, and $\alpha$ balances the two terms. Unless otherwise noted, we use $\alpha = 1.2$, following Zhou et al. (2024), which reports strong performance across tasks.

Note that, following common practice, we initialize both the auxiliary diffusion model $f_\psi$ and the generator $G_\theta$ from the teacher diffusion model $f_\phi$ Zhou et al. (2024); Yin et al. (2024a;b), which is crucial for facilitating and stabilizing training. Importantly, all three networks share the same architecture and capacity. Nonetheless, the generator has been shown to outperform the teacher Zhou et al. (2024), as it avoids the accumulation error inherent in multi-step sampling.

## J   IMPLEMENTATION DETAILS

**Hardware and measurement.** Unless otherwise noted, all pretraining and distillation runs use $8\times$NVIDIA A6000 GPUs. Inference wall-clock time is measured on $4\times$ NVIDIA A6000 GPUs with a batch size of 1024. Images are normalized to $[-1, 1]$ prior to adding Gaussian noise.

**Teacher pretraining: denoising task.**    For the denoising task, we follow the EDM setup Karras et al. (2022). On CIFAR-10, we train for **200** M image iterations to match the EDM computational budget; on FFHQ, CelebA-HQ, and AFHQ-v2 we train for **100** M iterations (half of EDM's budget). We adopt EDM hyperparameters verbatim Karras et al. (2022). All images are corrupted with additive Gaussian noise at the prescribed level.

**Distillation: denoising task.**    For the distillation phase, we train the one-step generator on CIFAR-10 for **100** M image iterations and on FFHQ, CelebA-HQ, and AFHQ-v2 for **15** M iterations; this budget suffices to reach competitive FID. Unless stated otherwise, hyperparameters mirror those of SiD Zhou et al. (2024). For CelebA-HQ, we use the same configuration as FFHQ/AFHQ-v2 except for a dropout rate of $0.15$.

**Teacher-consistency.**    For experiments in Tab 2 involving Teacher-Consistency Daras et al. (2024), we use 8 reverse steps and 32 Monte Carlo samples to approximate expectations. The consistency-loss weight is selected from $\{0.1, 1.0, 10.0\}$ as a fixed coefficient to maximize performance.

**General corruption tasks: pretraining.**    We again follow the EDM training protocol Karras et al. (2022) and pretrain for **100** M image iterations: (i) For Gaussian deblurring and super-resolution with $\sigma = 0$, we use the Standard Diffusion objective (Eq. 14). (ii) For the same tasks with $\sigma = 0.2$, we adopt the Diffusion for denoising loss (Eq. 15). (iii) For random inpainting with $\sigma = 0$, we use the publicly available Diffusion for random inpainting checkpoint at `https://github.com/giannisdaras/ambient-diffusion/tree/main`. (iv) For random inpainting with $\sigma = 0.2$, we train with the Diffusion for denoising loss (Eq. 15); we avoid the dedicated inpainting loss (Eq. 16) in this noisy regime due to instability, consistent with Daras et al. (2023b). (v) For Fourier-space random inpainting on MRI, we initialize from the checkpoint at `https://github.com/utcsilab/ambient-diffusion-mri.git`. Unless noted, pretrained diffusion models use EDM hyperparameters Karras et al. (2022).

**General corruption tasks: distillation.**    During distillation we train the one-step generator for **50–100** M image iterations, with early stopping if the validation FID begins to diverge.

**Reproducibility.**    Upon acceptance, we will release code and checkpoints to facilitate reproduction and further research.

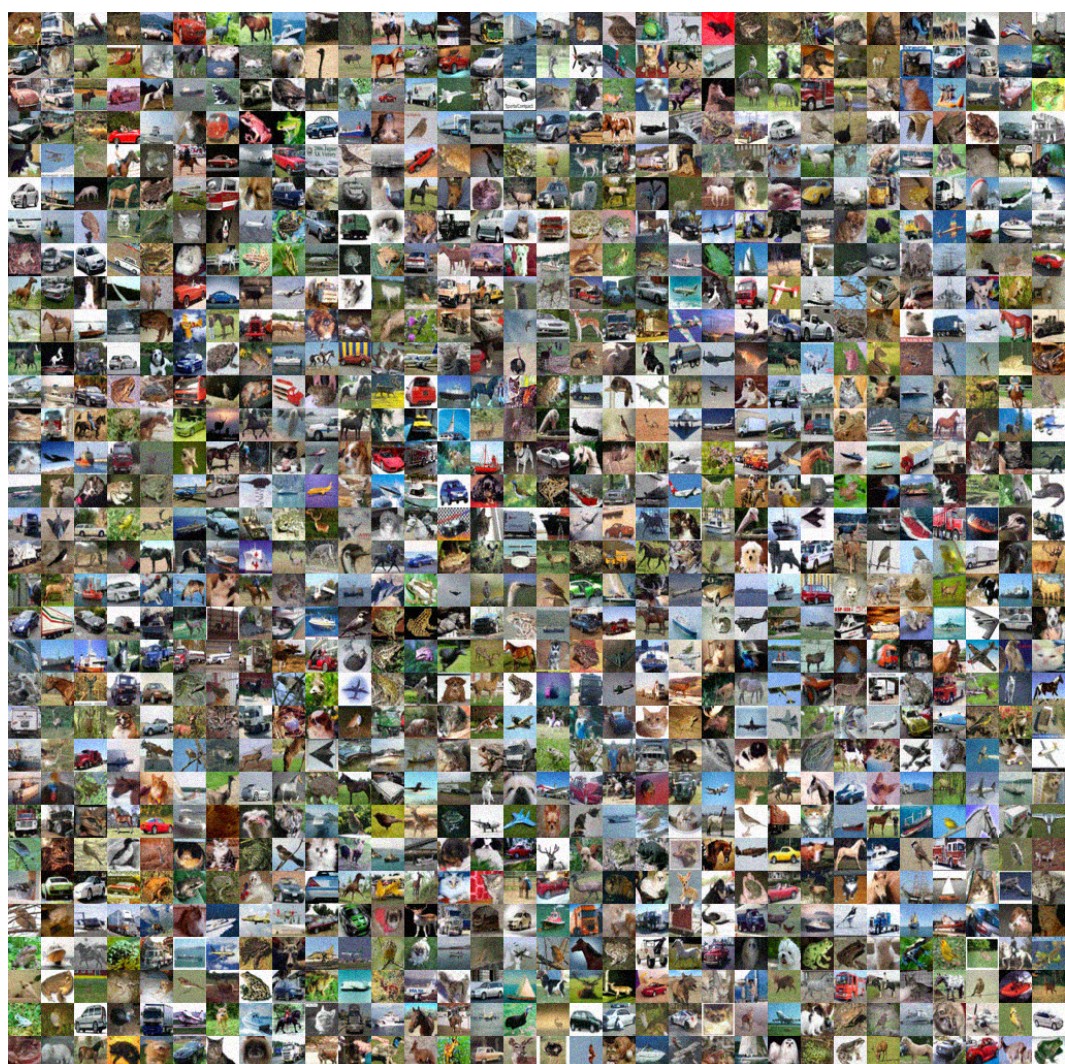

Figure 15: CIFAR-10 32x32 noisy dataset with $\sigma = 0.1$ (FID: 73.74).

## K  ADDITIONAL QUALITATIVE RESULTS

In this section, we present additional qualitative results. A quick view is in Appendix A.

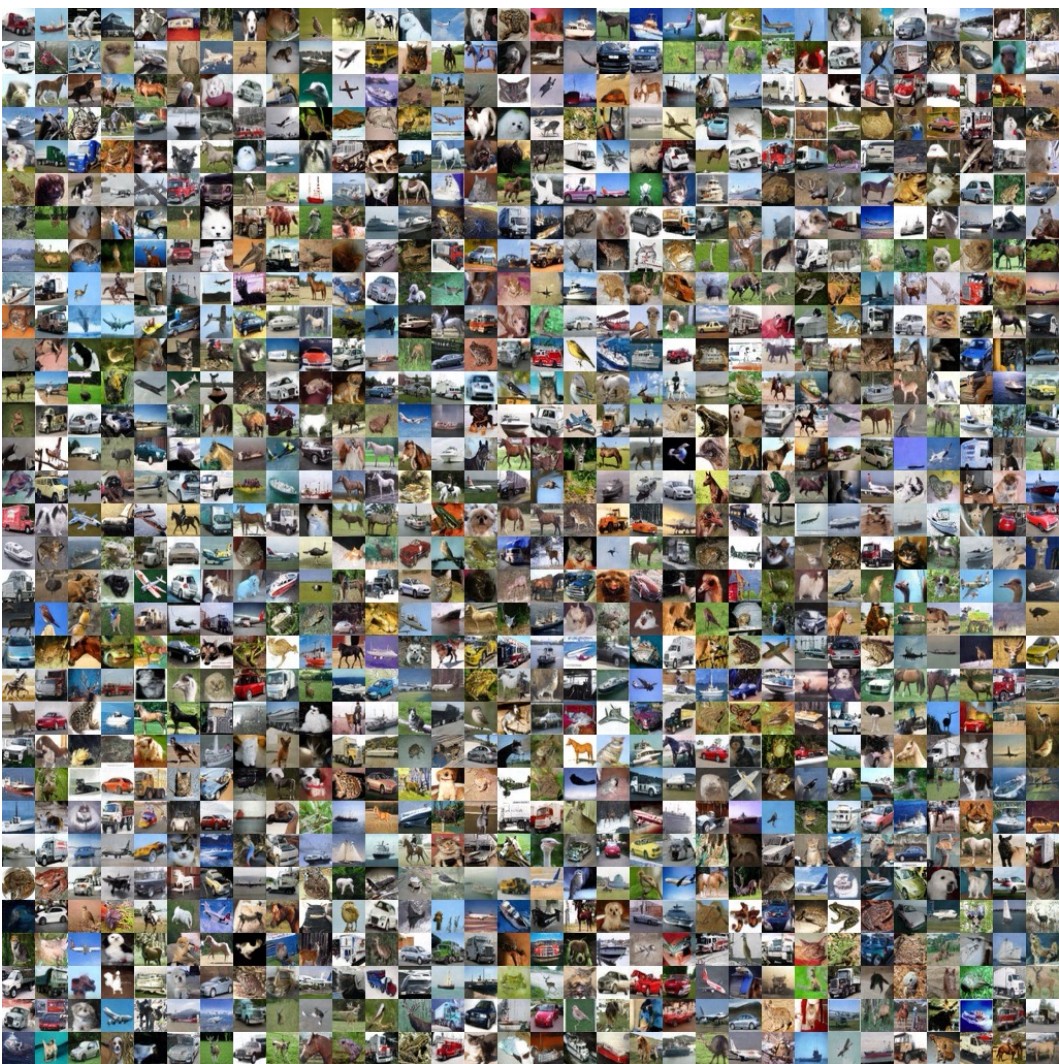

Figure 16: Unconditional CIFAR-10 32x32 random images generated with RSD training with noisy dataset with $\sigma = 0.1$ (FID: 3.98).

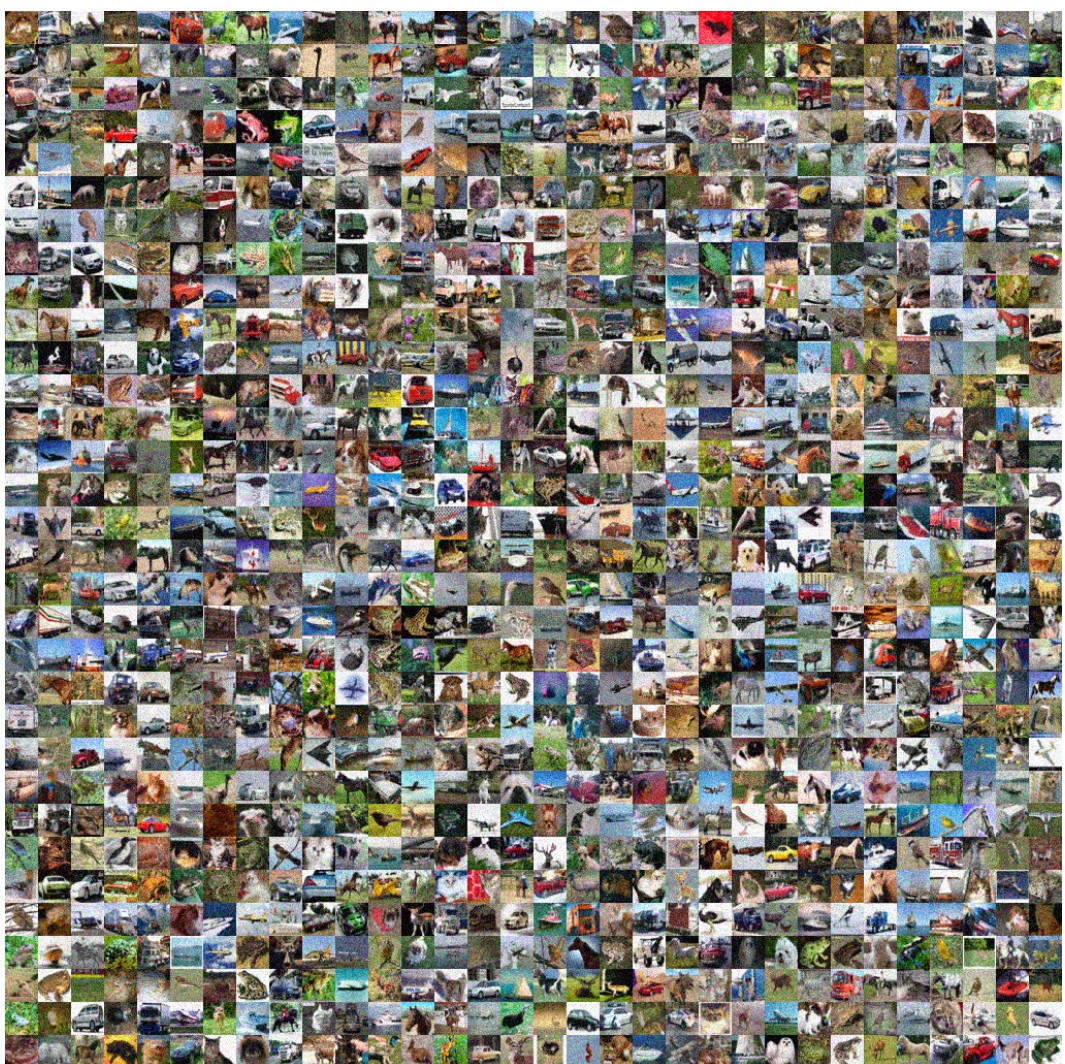

Figure 17: CIFAR-10 32x32 noisy dataset with $\sigma = 0.2$ (FID: 127.22).

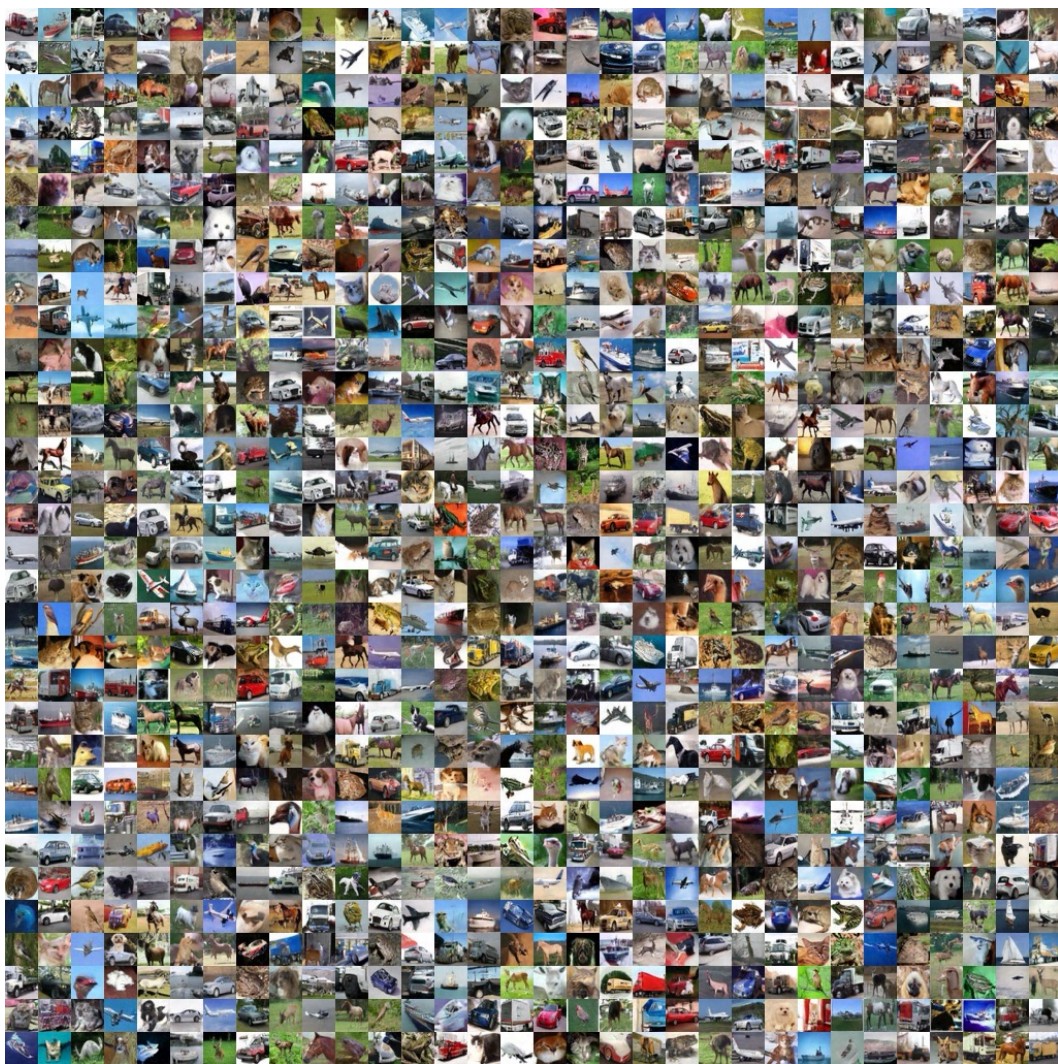

Figure 18: Unconditional CIFAR-10 32x32 random images generated with RSD training with noisy dataset with $\sigma = 0.2$ (FID: 4.77).

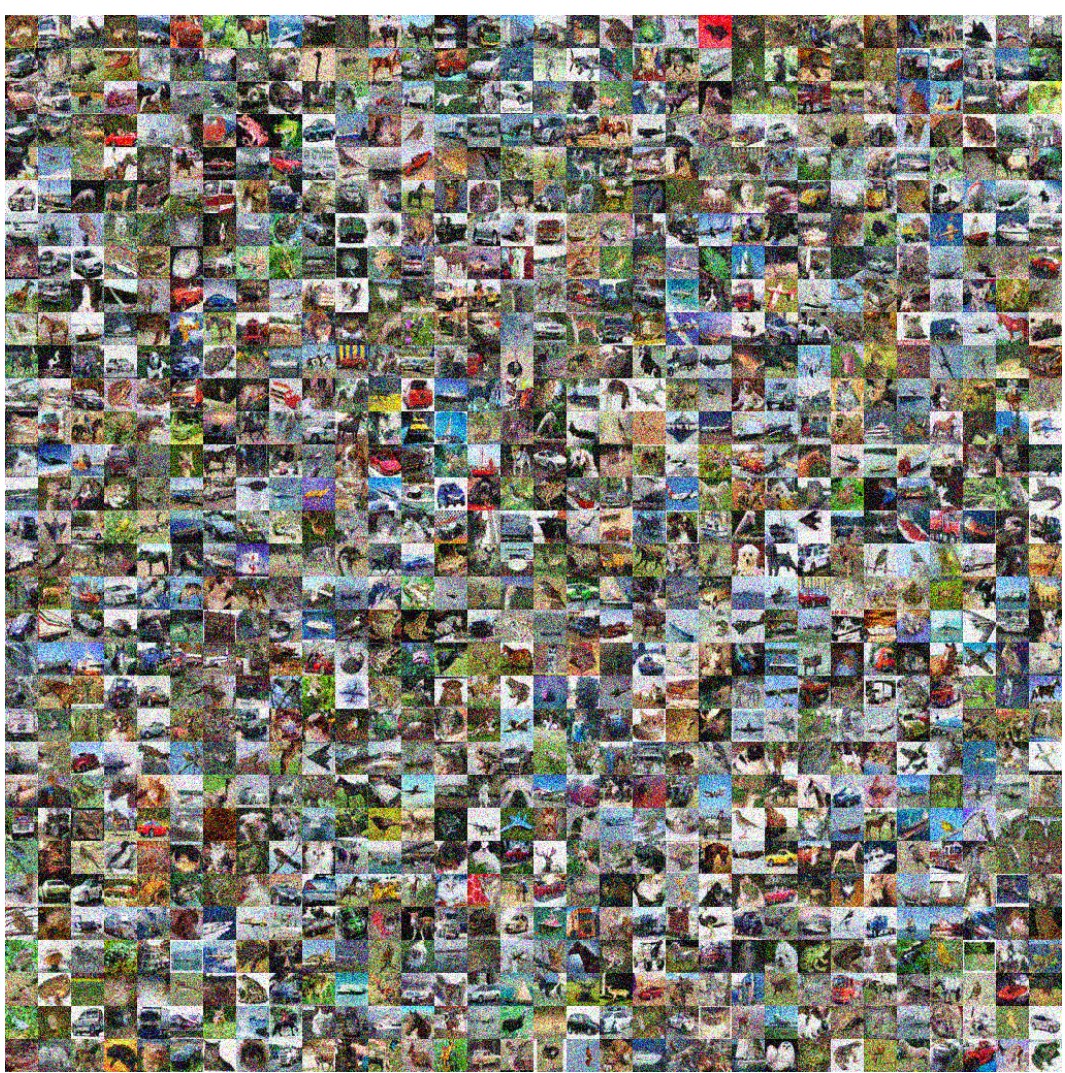

Figure 19: CIFAR-10 32x32 noisy dataset with $\sigma = 0.4$ (FID: 205.52).

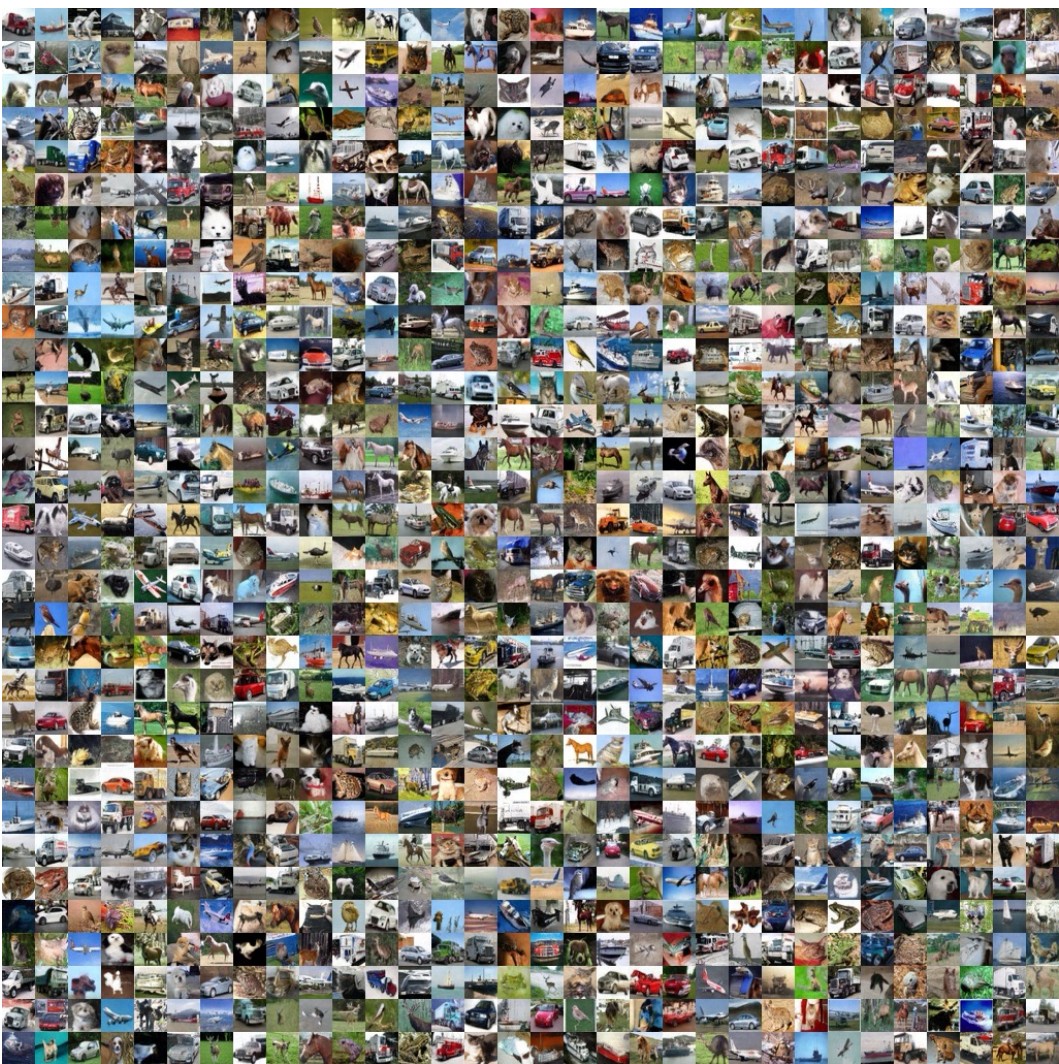

Figure 20: Unconditional CIFAR-10 32x32 random images generated with RSD training with noisy dataset with $\sigma = 0.4$ (FID: 21.63).

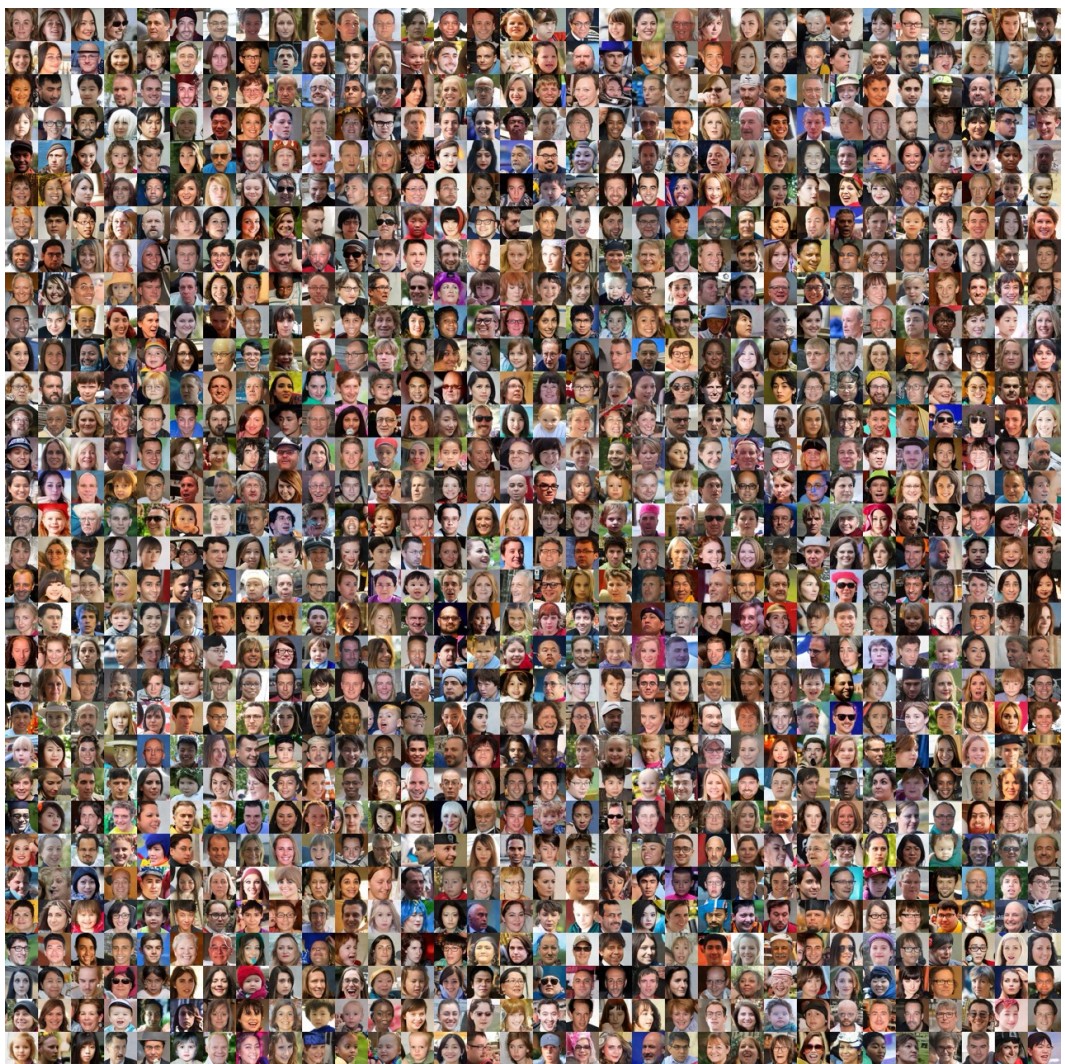

Figure 21: Unconditional FFHQ 64x64 random images generated with RSD training on noisy dataset with $\sigma = 0.2$ (FID: 6.29).

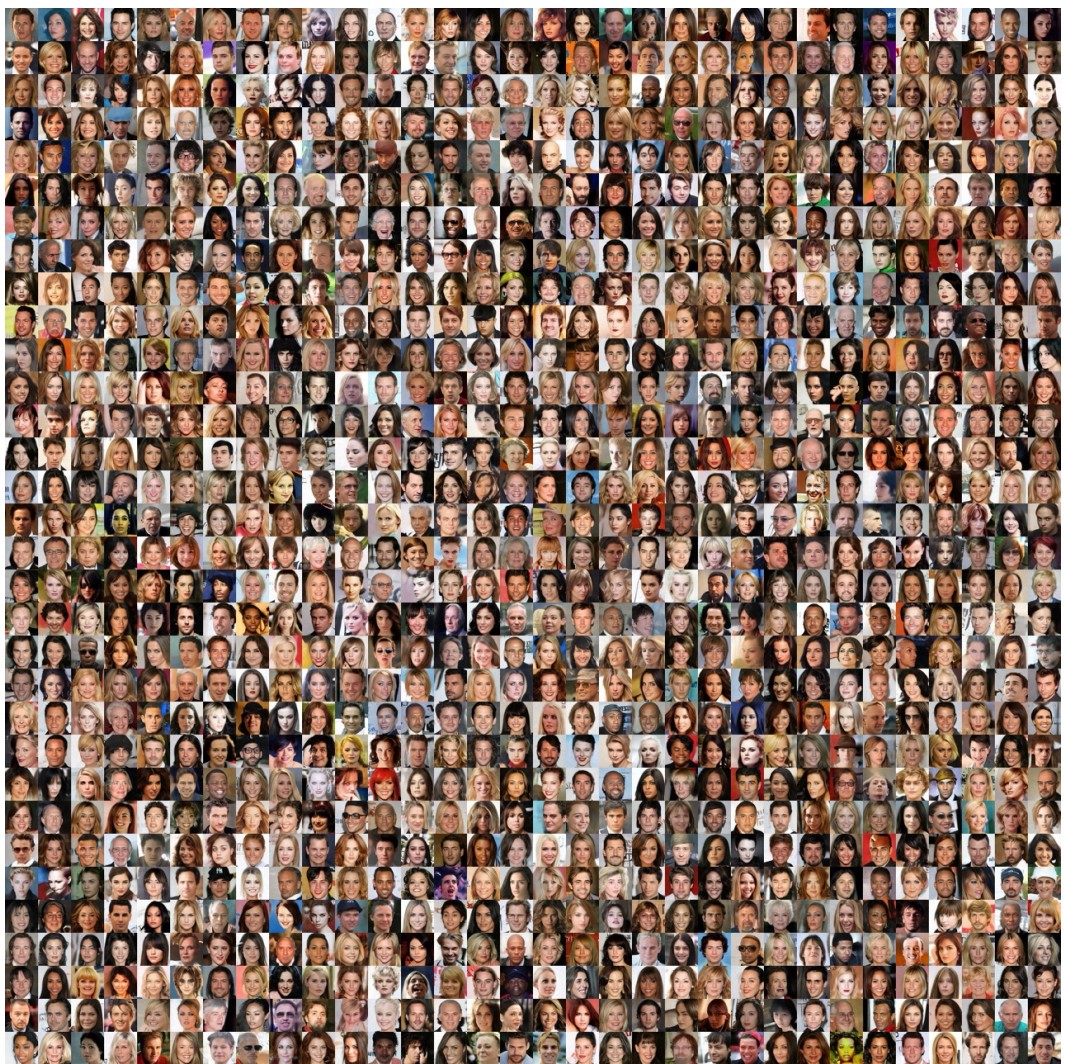

Figure 22: Unconditional CelebA-HQ 64x64 random images generated with RSD training on noisy dataset with $\sigma = 0.2$ (FID: 6.48).

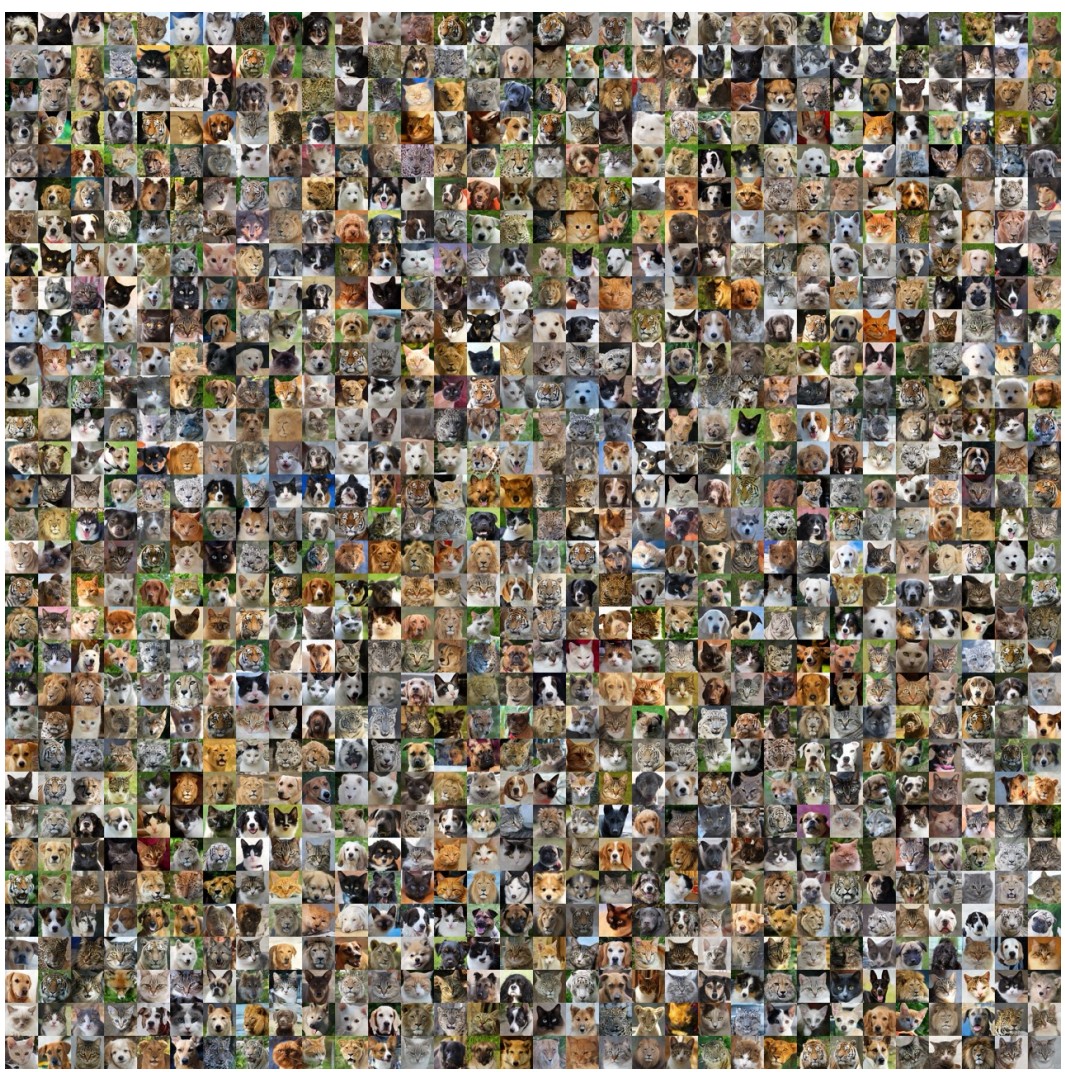

Figure 23: Unconditional AFHQ-v2 64x64 random images generated with RSD training on noisy dataset with $\sigma = 0.2$ (FID: 5.42).

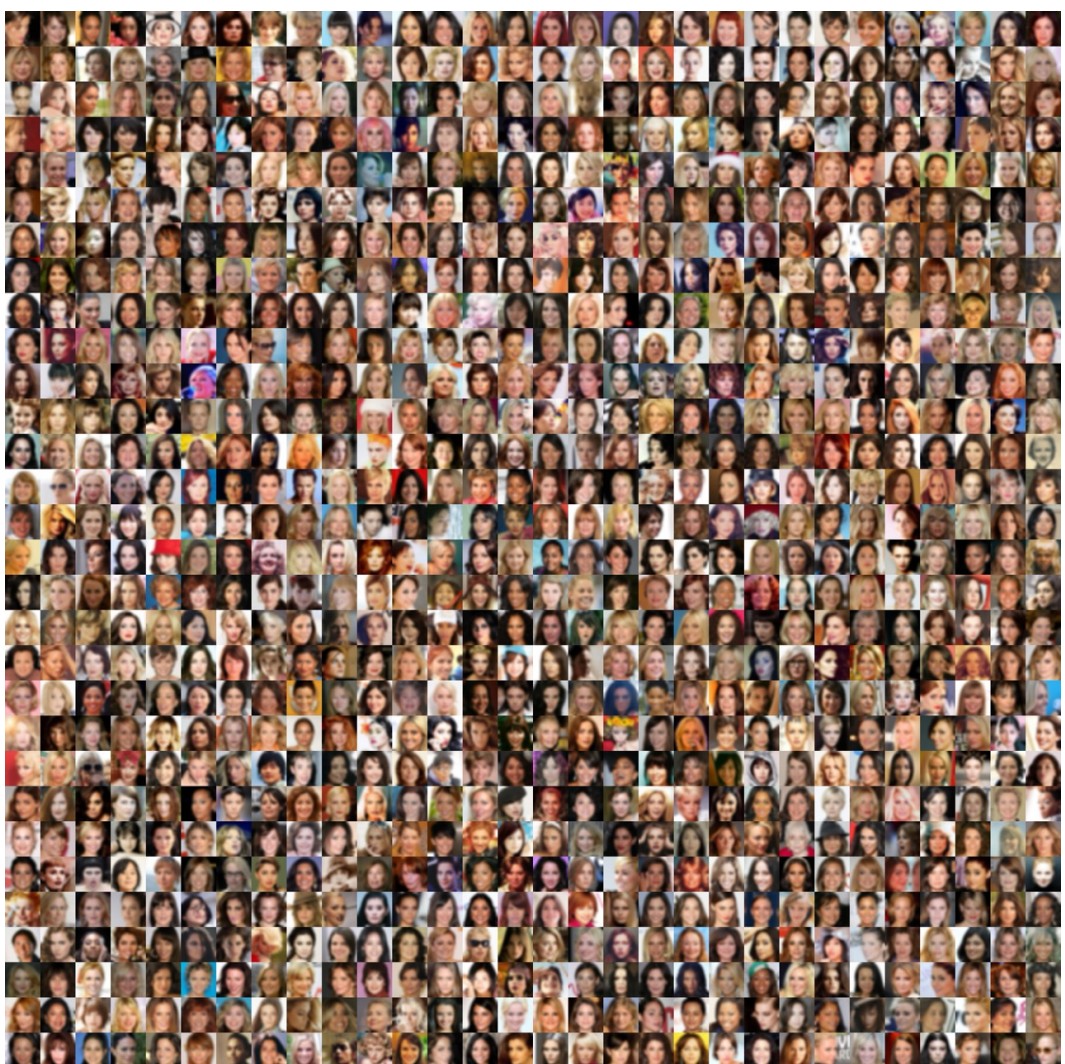

Figure 24: Examples from the training dataset used for the Gaussian blur task with $\sigma = 0$.

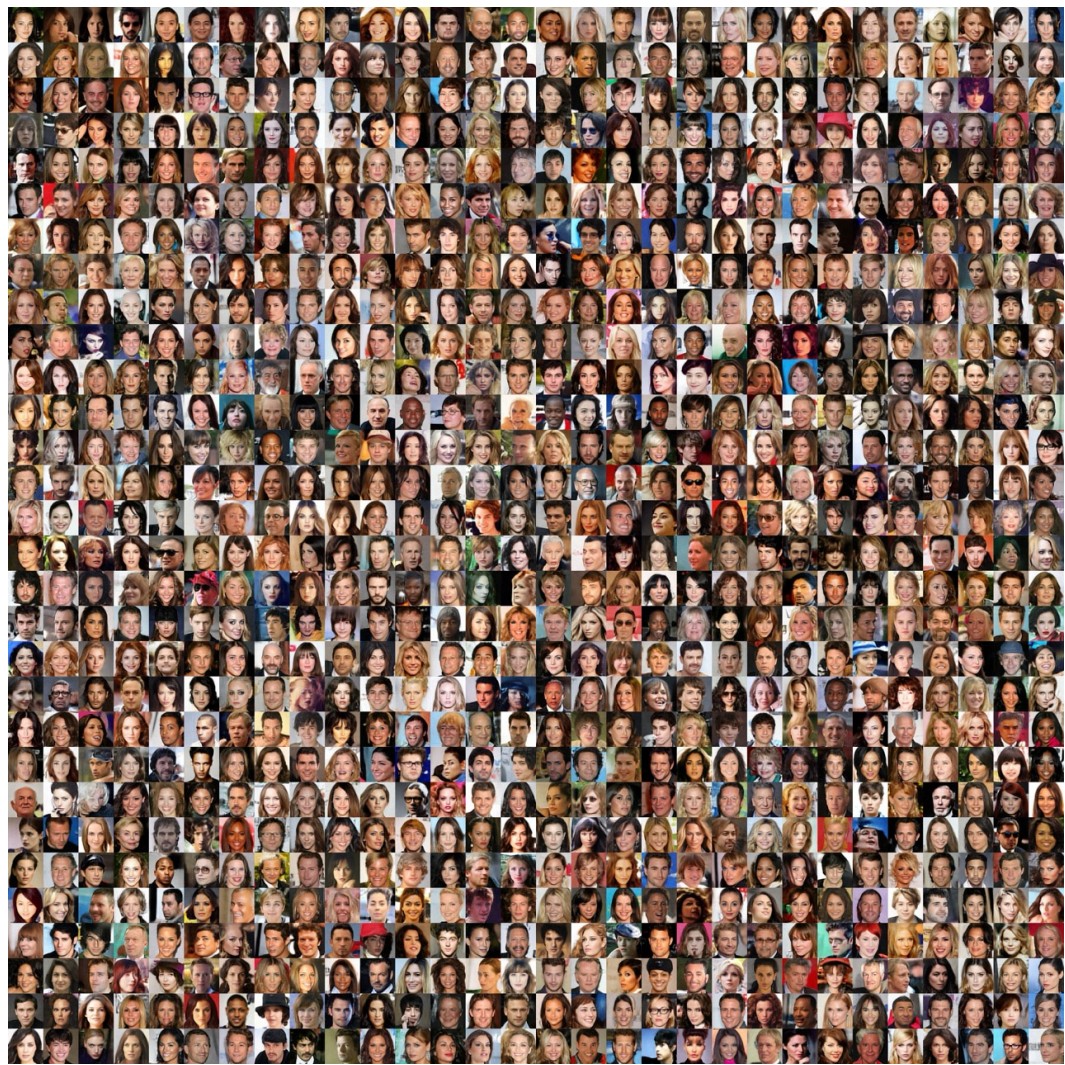

Figure 25: Qualitative results of RSD generation for the Gaussian blur task with $\sigma = 0$.

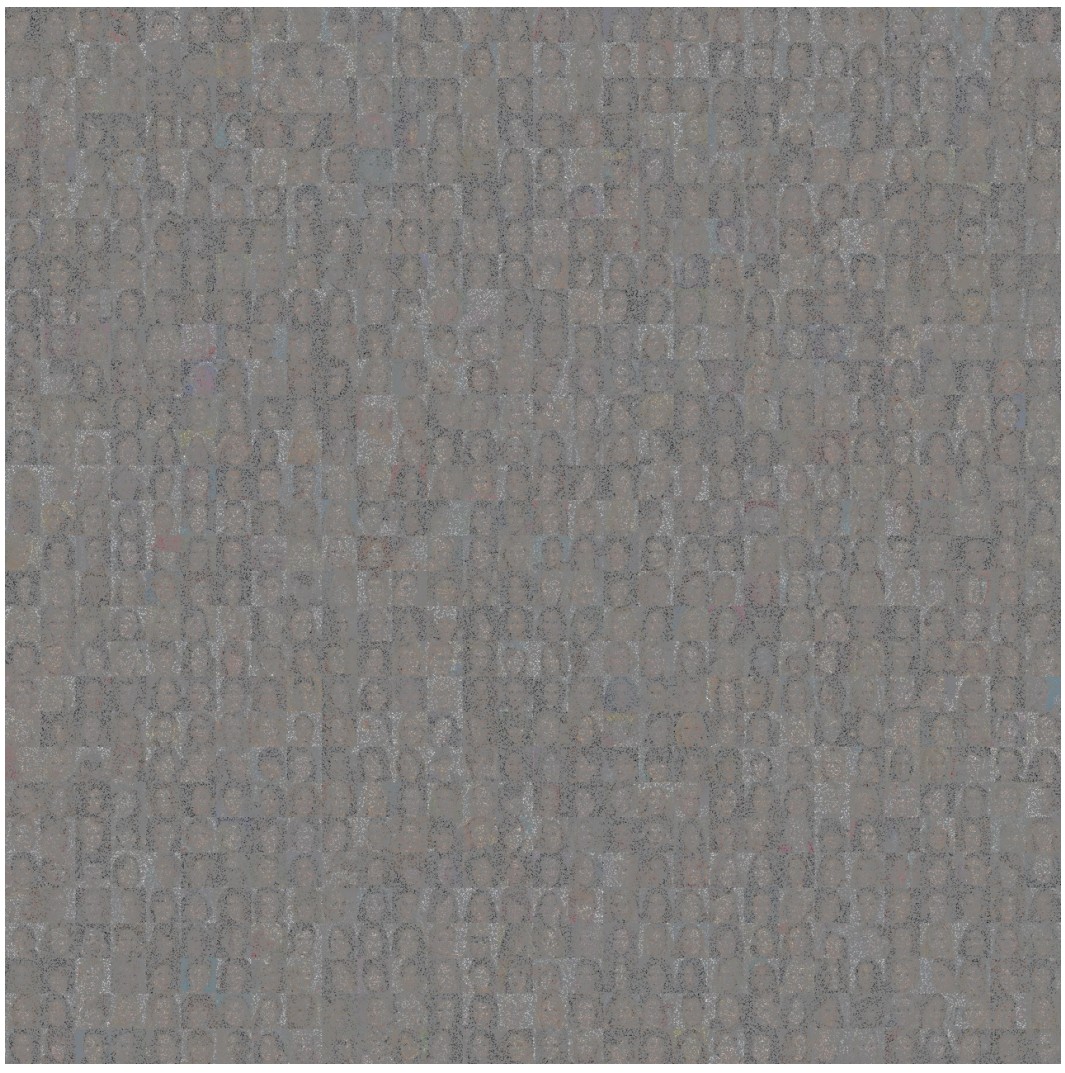

Figure 26: Dataset samples for the random inpainting task with $p = 0.9$ and $\sigma = 0$.

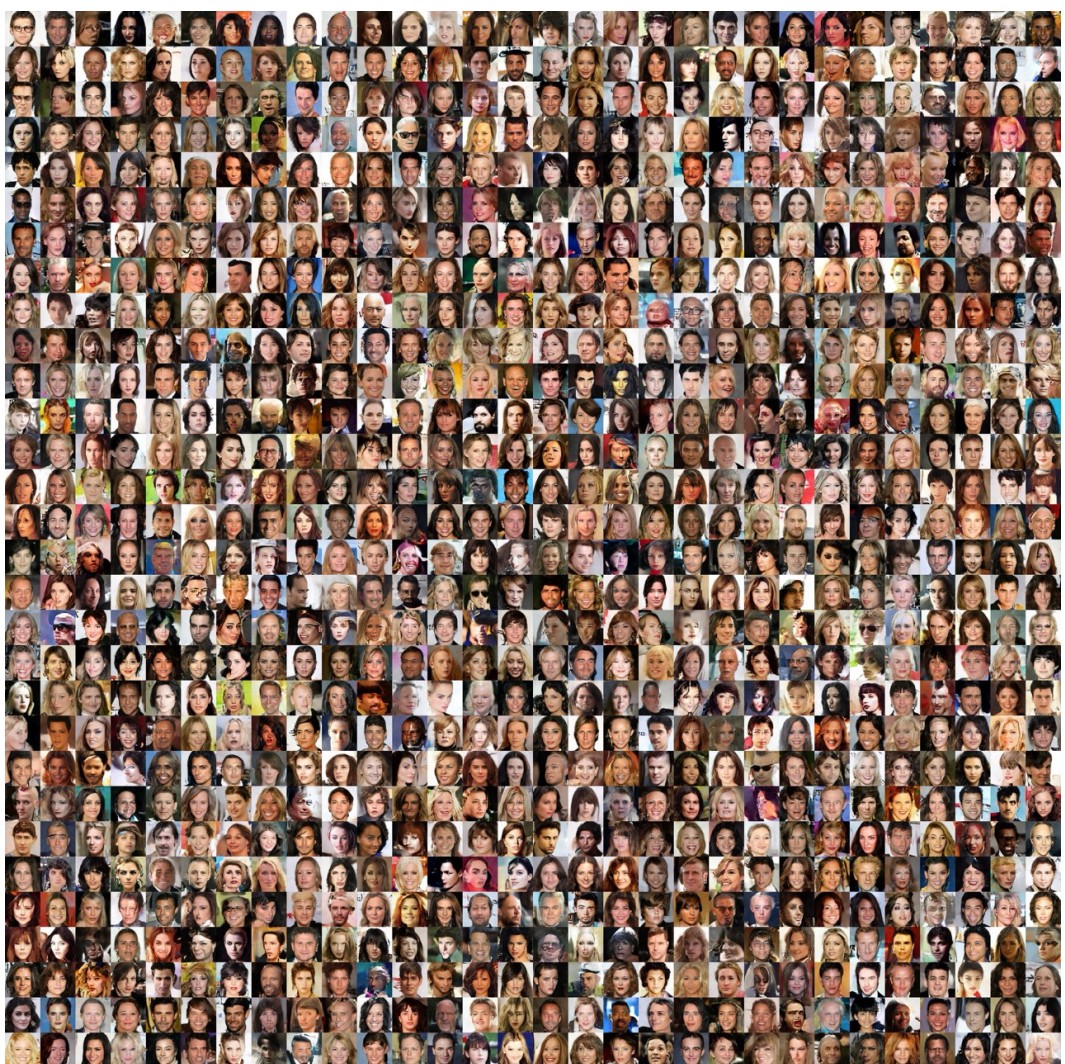

Figure 27: RSD generation results for the random inpainting task with $p = 0.9$ and $\sigma = 0$.

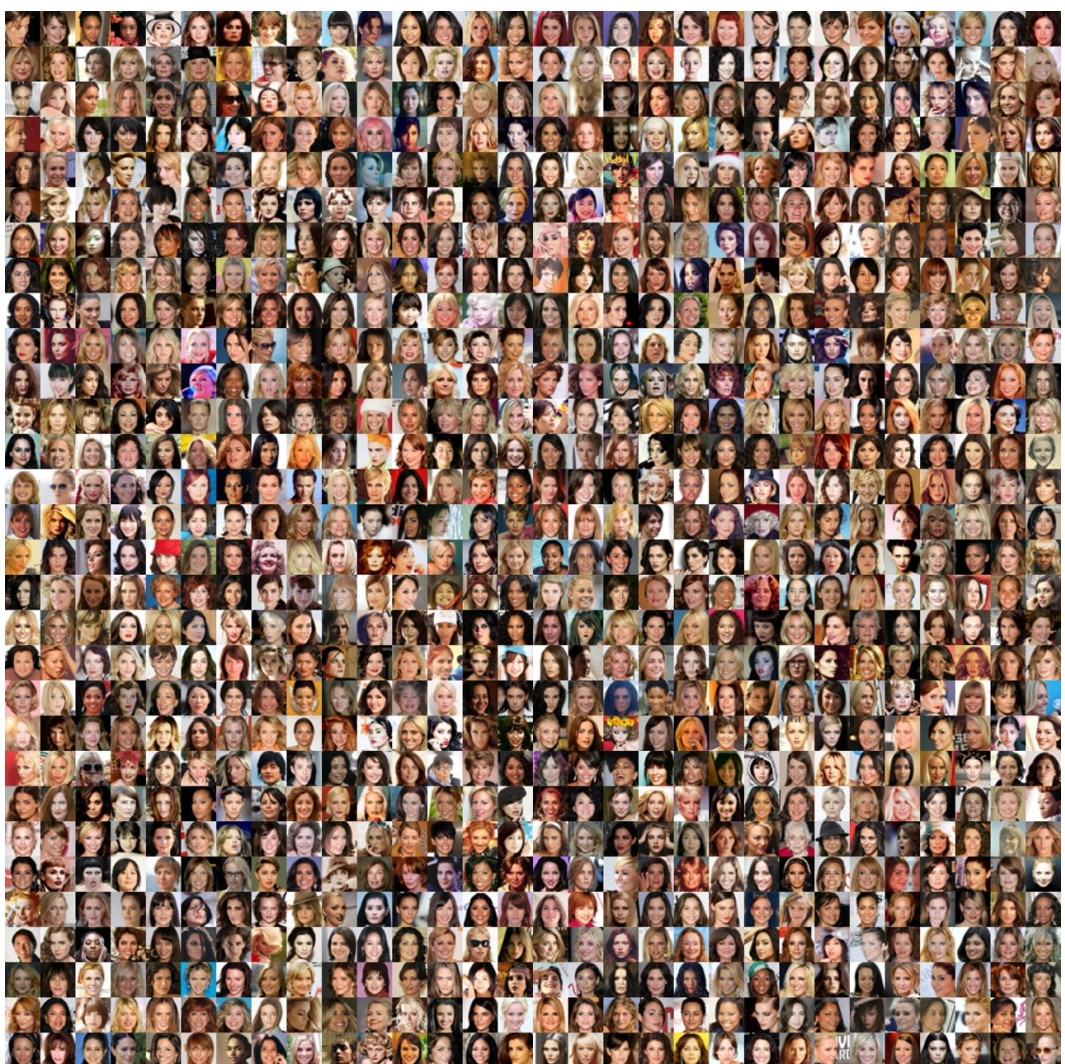

Figure 28: Dataset samples for Super Resolution $\times 2$ Task with $\sigma = 0$

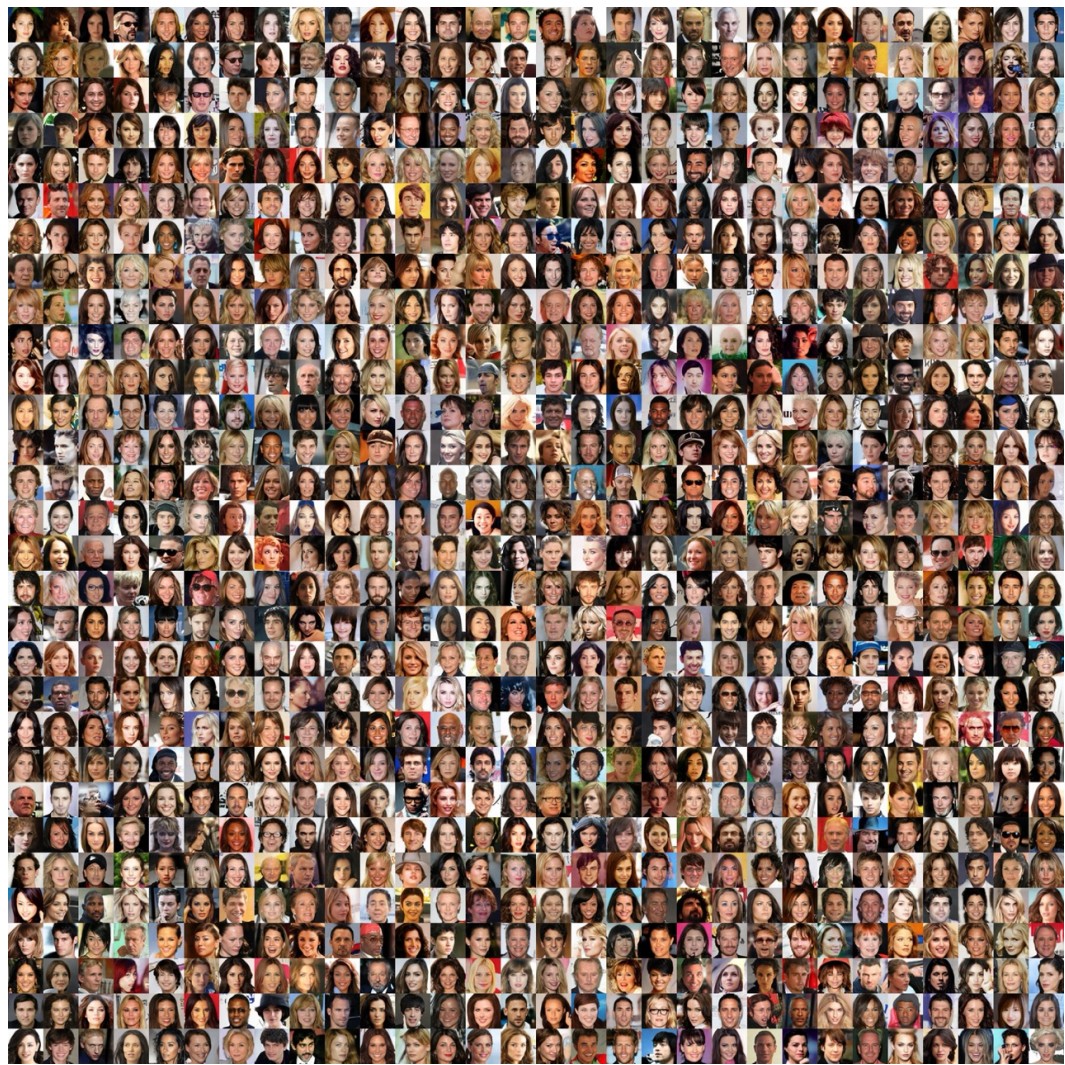

Figure 29: RSD generation results for Super Resolution $\times 2$ Task with $\sigma = 0$

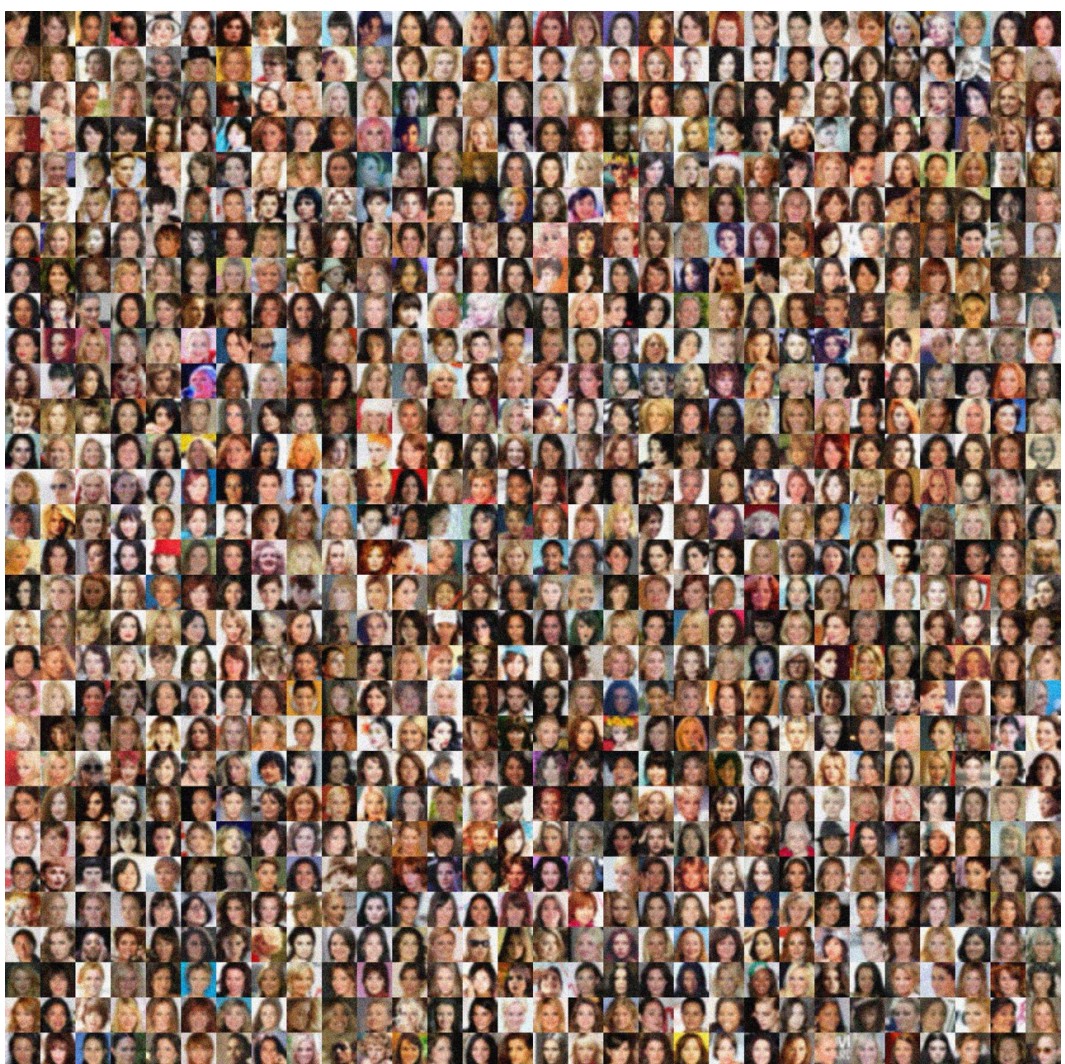

Figure 30: Dataset samples for Gaussian Blur with $\sigma = 0.2$

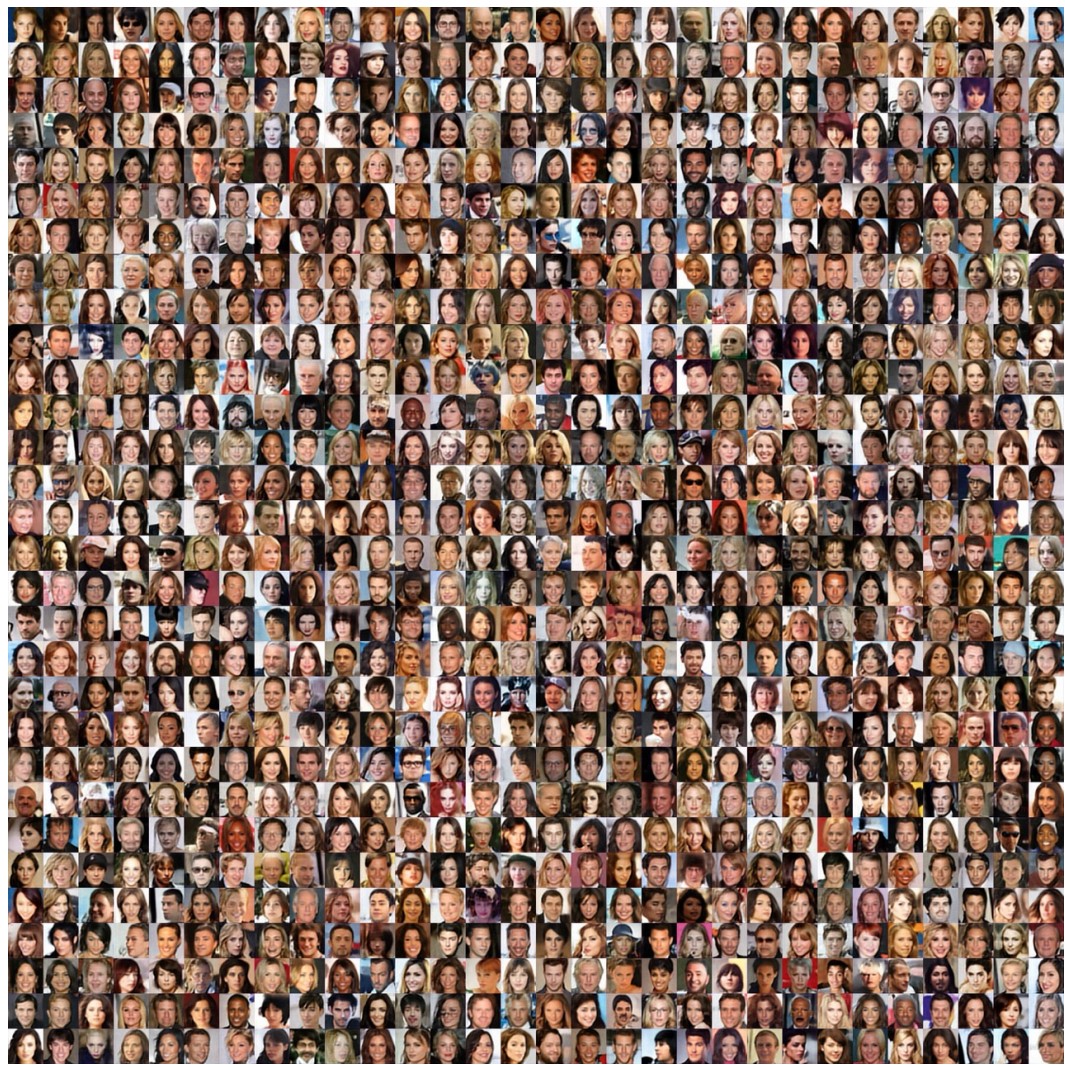

Figure 31: RSD generation results for Gaussian Blur with $\sigma = 0.2$

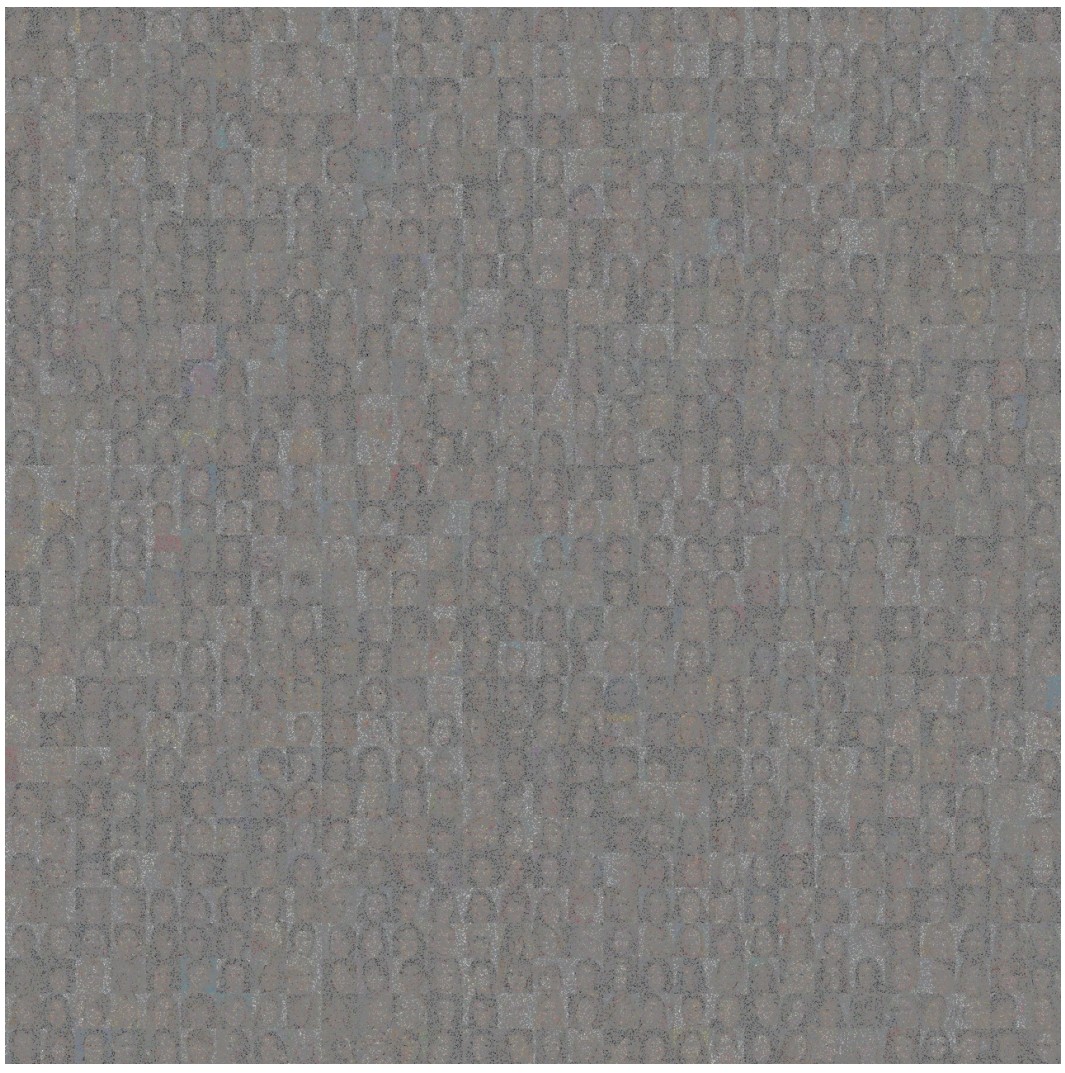

Figure 32: Dataset samples for Random Inpainting with $p = 0.9$ and $\sigma = 0.2$

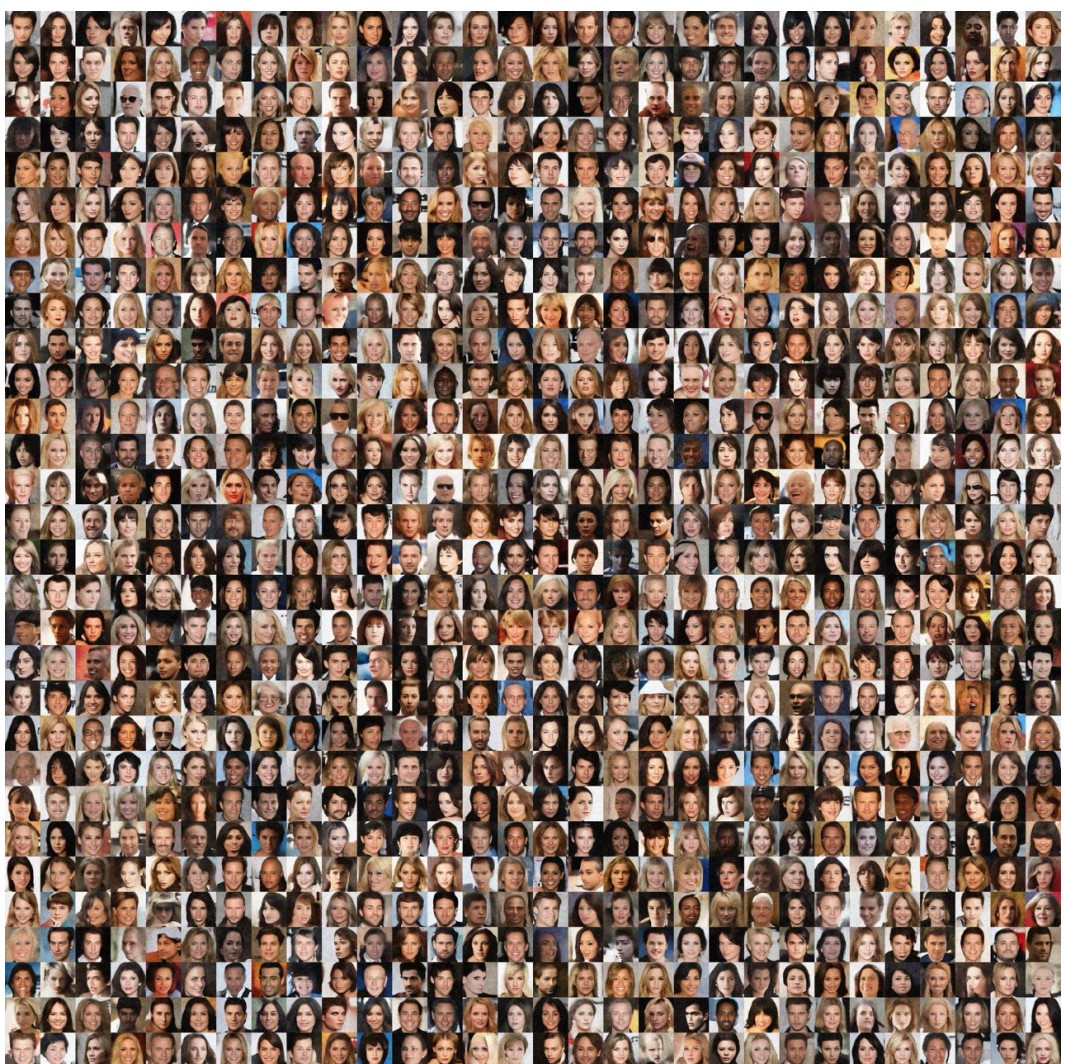

Figure 33: RSD generation results for Random Inpainting with $p = 0.9$ and $\sigma = 0.2$

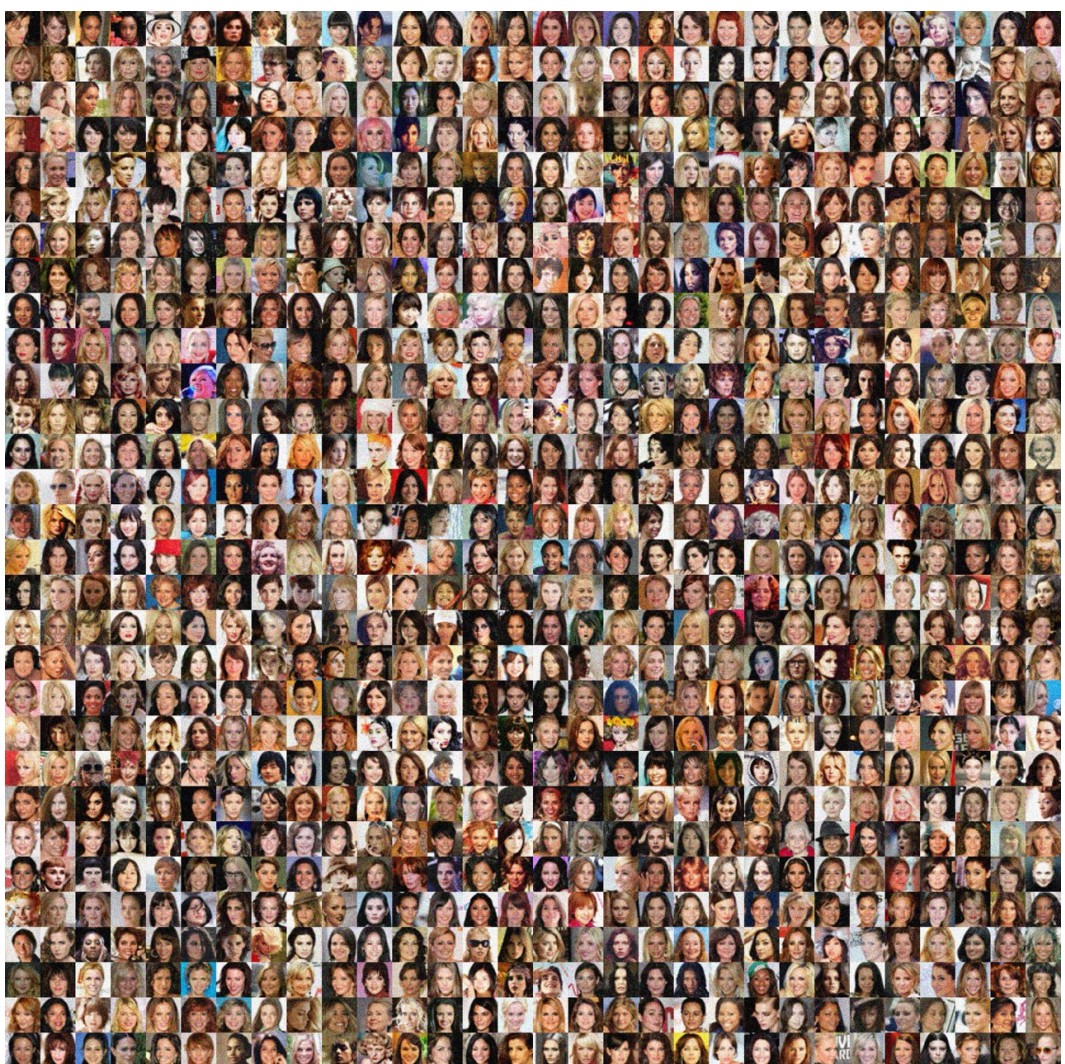

Figure 34: Dataset samples for Super Resolution $\times 2$ Task with $\sigma = 0.2$

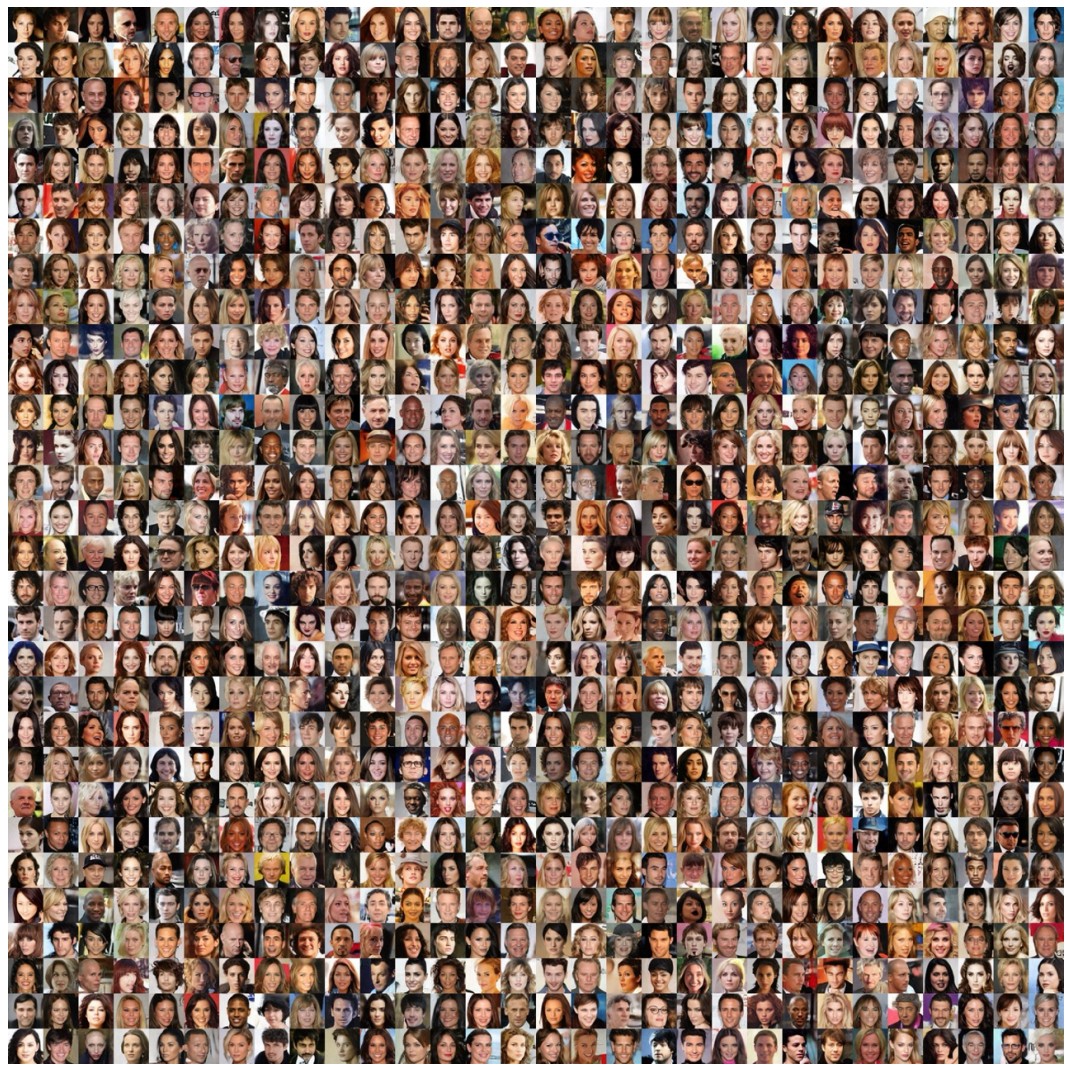

Figure 35: RSD generation results for Super Resolution $\times 2$ Task with $\sigma = 0.2$

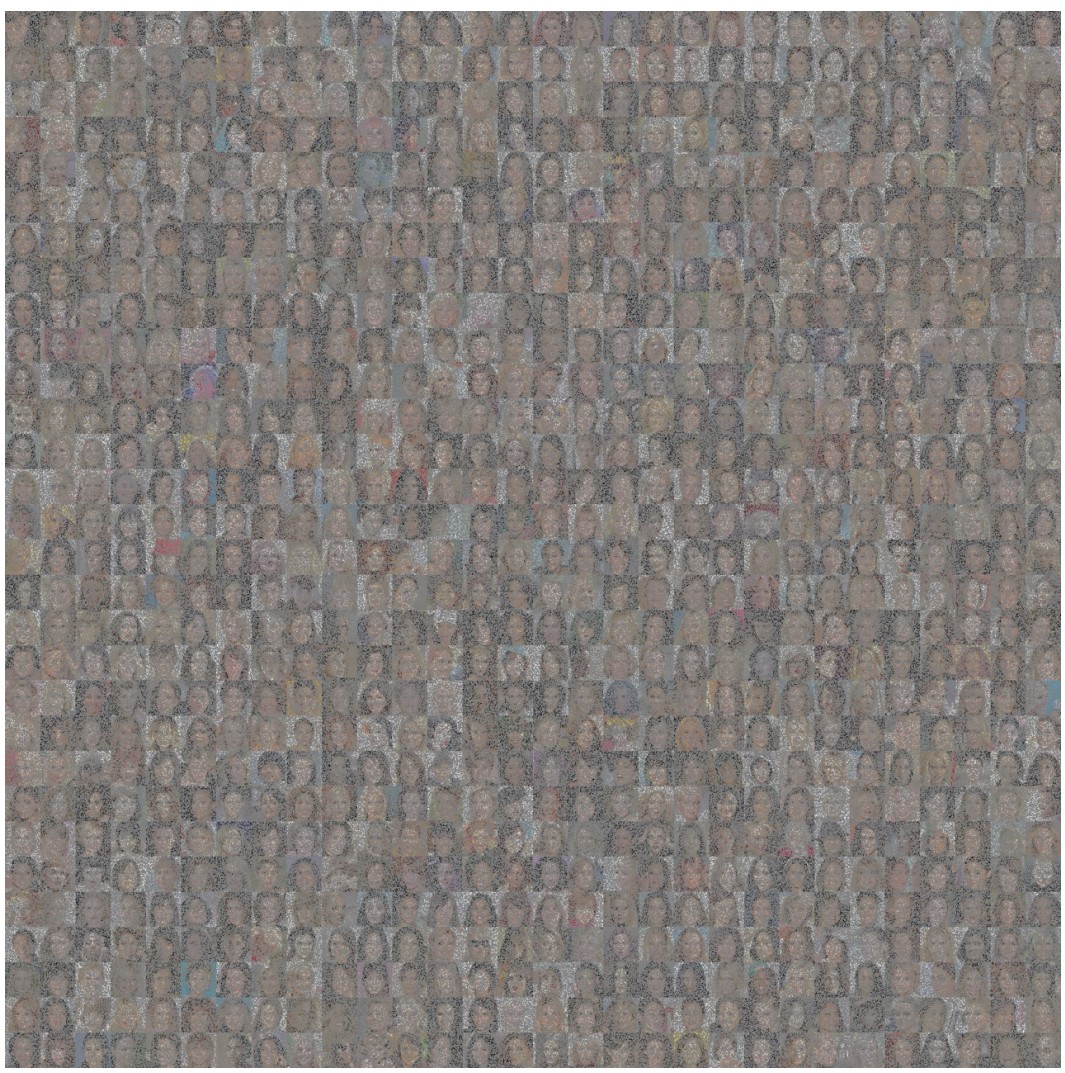

Figure 36: Dataset samples for Random Inpainting with $p = 0.8$ and $\sigma = 0$

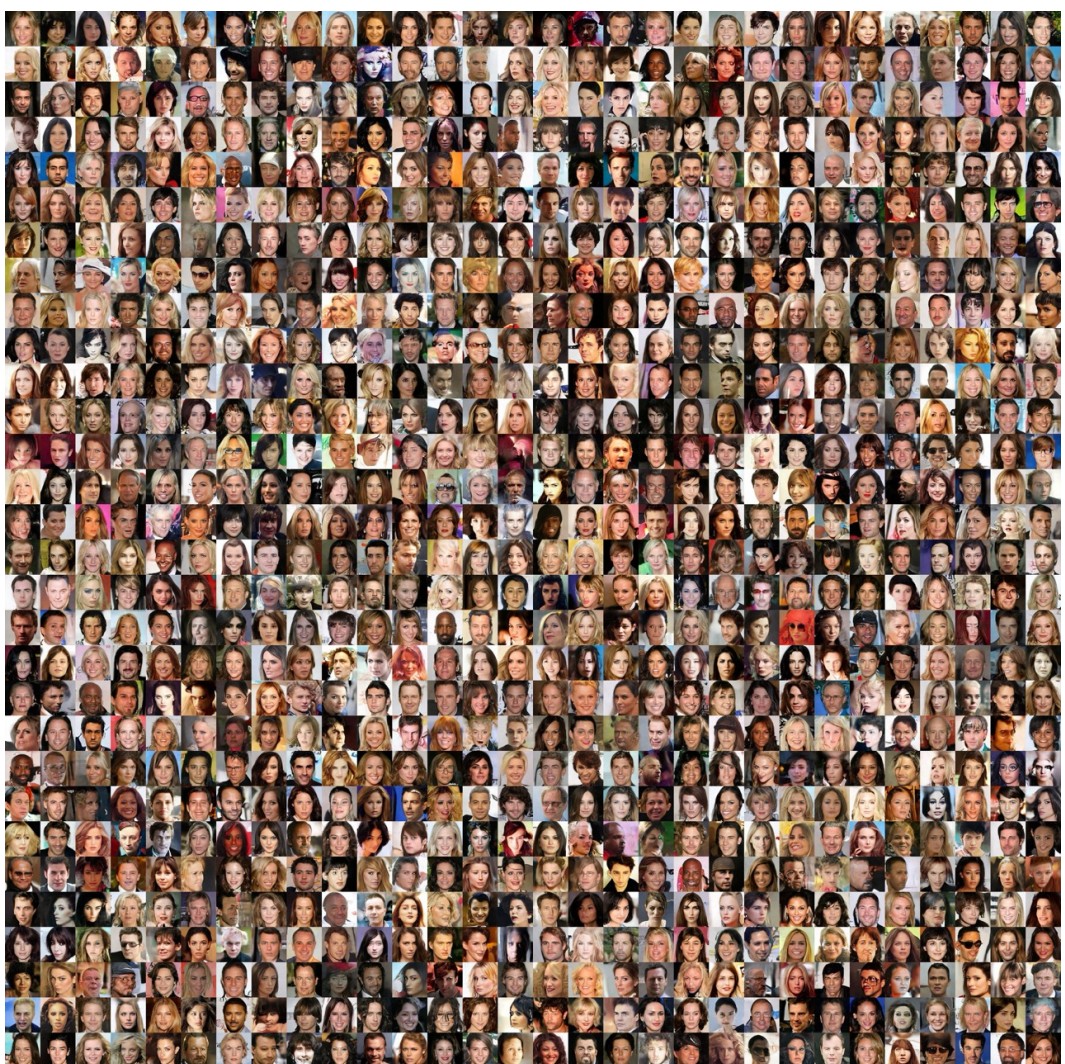

Figure 37: RSD generation results for Random Inpainting with $p = 0.8$ and $\sigma = 0$

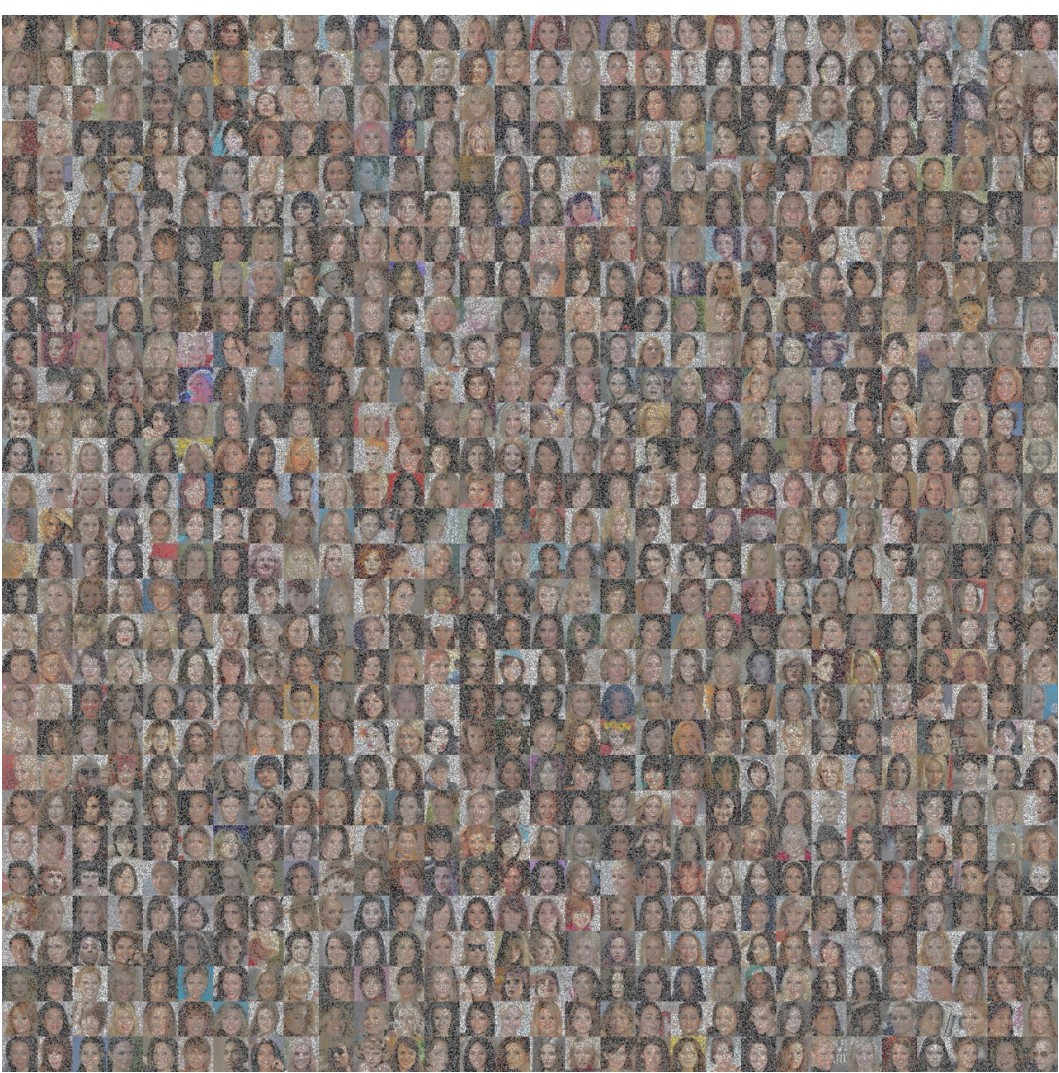

Figure 38: Dataset samples for Random Inpainting with $p = 0.6$ and $\sigma = 0$

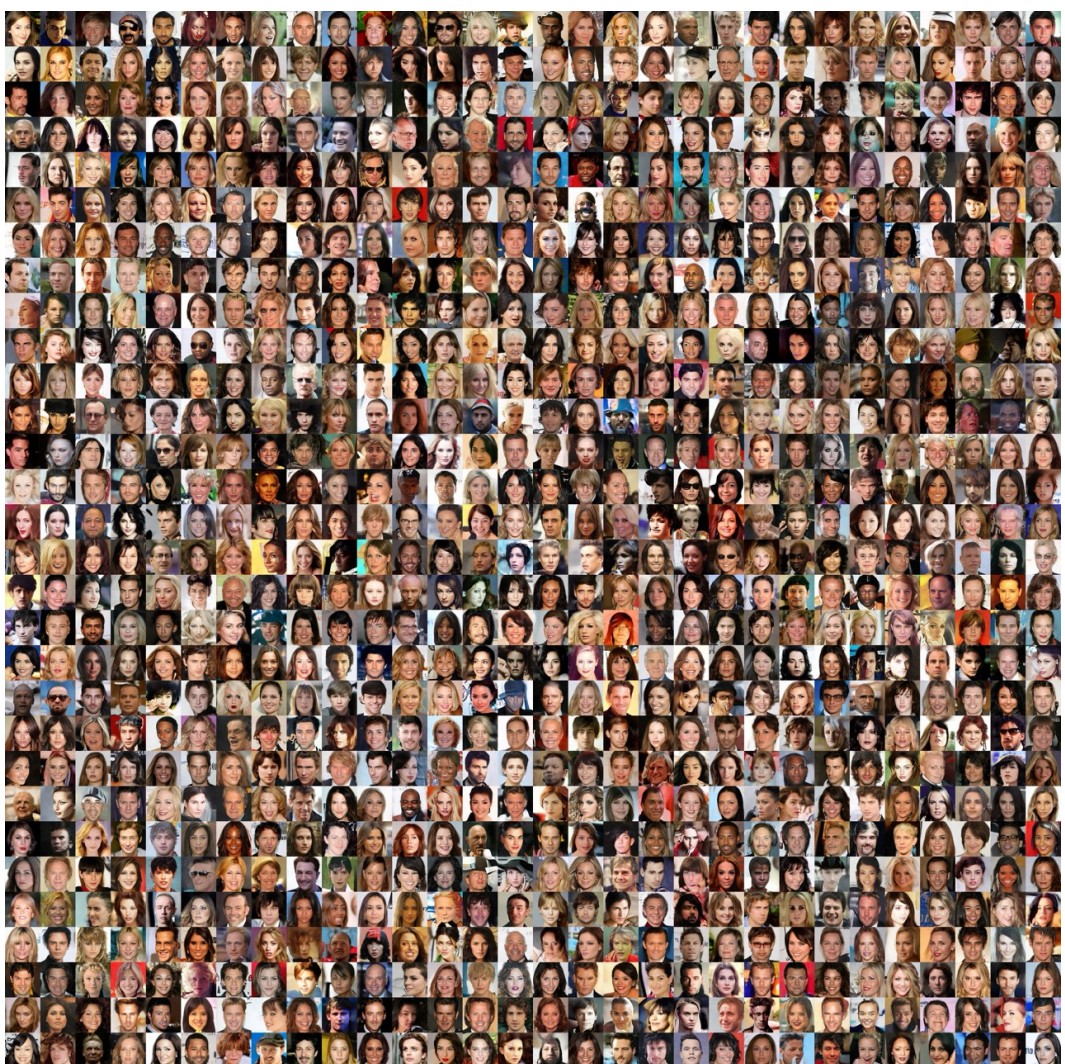

Figure 39: RSD generation results for Random Inpainting with $p = 0.6$ and $\sigma = 0$

