# OpenReview forum: "Score Distillation Beyond Acceleration: Generative Modeling from Corrupted Data"
_ICLR.cc/2026/Conference — ICLR 2026 Poster_

### Official Review · Reviewer_5ZSz · 2025-10-31

**Soundness:** 3
**Presentation:** 3
**Contribution:** 3
**Rating:** 6
**Confidence:** 4

**Summary:**

This paper introduces a unified and powerful framework, Distillation from Corrupted Data (DCD), for a long-standing and critical challenge: training high-fidelity generative models using only corrupted or degraded data. The core idea is a two-phase approach: first, a "corruption-aware" diffusion model (teacher) is trained directly on the imperfect observations; second, this slow, multi-step teacher is distilled into a highly efficient one-step generator (student). The central and most significant finding is that this distillation process does not merely accelerate sampling but consistently and substantially *improves* the generative quality, often surpassing the teacher by a large margin. The authors provide an exceptionally thorough empirical validation across a wide array of tasks—including denoising, deblurring, inpainting, super-resolution, and real-world multi-coil MRI reconstruction—demonstrating state-of-the-art performance in the challenging "zero-shot" setting where no clean data is ever used.

**Strengths:**

1. The paper tackles a problem of immense practical importance. The discovery that score distillation can serve as a powerful tool for *improving* generation quality from corrupted data, rather than just for acceleration, is a significant conceptual contribution that could shift how we approach learning from imperfect sources.

2. The experimental results are comprehensive and consistently impressive. DCD achieves remarkable FID improvements over its teachers across all tested corruption types, datasets, and severity levels. Its ability to outperform specialized "few-shot" methods without access to any clean data is a testament to the framework's power and robustness.

3. The two-phase design is clean, intuitive, and highly flexible. By decoupling the corruption-aware pretraining from the distillation, the framework can readily incorporate new and future advances in diffusion modeling for inverse problems. This modularity makes DCD a highly practical and future-proof solution.

4. The successful application to the multi-coil MRI problem, a challenging scientific domain with complex-valued data and real acquisition constraints, strongly underscores the real-world viability and potential of this work.

**Weaknesses:**

1. While the paper mentions theoretical support in the appendix, the main text could benefit from a more intuitive explanation of *why* distillation provides such a significant quality boost in the corrupted data regime. Is it a form of regularization? Does it find a more stable generator mode? A brief, high-level discussion would further strengthen the paper's narrative.

**Questions:**

N/A

---

> ### Author Response · Authors · 2025-11-22
> **rebuttal**
>
> We sincerely thank the reviewer for your thoughtful and generous review, which highlights our work’s significant conceptual contribution towards distillation, strong and comprehensive experimental results, clean and modular design, and real-world effectivenes.
>
> We updated our paper with a new section 3.3 for a deeper and more intuitive analysis of the underlying mechanism of our method, titled “Why Score Distillation Beyond Acceleration?”. We provide the same text for your convenience below:
>
> ### Why Score Distillation Beyond Acceleration?
>
> Score distillation is traditionally viewed as a means of *accelerating* sampling by compressing a multi-step diffusion process into a single forward pass. However, our paper reveals a more fundamental **conceptual shift**:  in corrupted or low-quality data regimes, score distillation acts as a *principled mechanism* for improving the sample quality of teacher diffusion models.
>
>
> ---
>
> #### **A Toy Example**
> In Fig. 2, the observation is a noisy spiral dataset with $\sigma$ = 0.05.  Teacher diffusion models on noisy datasets—(b) Teacher-Full and (c) Teacher-Truncated—must spread their probability mass across all regions supported by the corrupted data, making the learned density overly diffuse. In contrast, (d) DCD excels at denoising the original dataset, producing a narrow, concentrated, and sharp approximation.  This improvement occurs *without* ever providing the clean data during training.
>
>
>
> #### **Intuitive Understanding**
> Why does this sharpening occur? Diffusion models are trained using the ELBO objective for maximum likelihood, which minimizes the **forward KL divergence**—a *mode-covering* objective. Thus, teacher diffusions allocate probability mass across *all* plausible regions indicated by the noisy observations, even those that do not correspond to the true clean data distribution.
>
> In contrast, the **reverse distillation objective** (Eq. 5) is taken over the *student generator’s* distribution, naturally inducing **mode-seeking** behavior: the generator only needs to place mass where it actually samples. Importantly, **encouraging mode seeking does not imply mode collapse.**  (FID and Recall in Tab. 11 as well as multiple qualitative examples in the Appendix support this.) This shift allows the distilled generator to concentrate its probability mass on the **high-density, well-supported regions** implied by the teacher’s score field, while discarding overly diffuse or low-density regions that teacher diffusion models include.
>
>
>
> #### **Theoretical Support**
> We provide detailed analysis in the appendix:
>
> - Section C.1 derives bounds on distilled generator performance under standard capacity and optimization assumptions.
> - Section C.2 specializes to a linear Gaussian setting, yielding explicit optimization landscapes, quantitative error bounds, and closed-form expressions for global minimizers (Eq. 10).
> - Section C.3 extends the theory to general linear corruption settings.
>
> ---
>
>
> We hope our explanation could address your concerns and increase your confidence in our work, and we are willing to offer more clarification if needed.

---

### Official Review · Reviewer_AX6j · 2025-10-31

**Soundness:** 4
**Presentation:** 4
**Contribution:** 4
**Rating:** 8
**Confidence:** 4

**Summary:**

This paper introduces **Distillation from Corrupted Data (DCD)**, a two-stage framework for training one-step generative models directly from corrupted observations without access to clean data. The method first trains a corruption-aware diffusion teacher on corrupted data (e.g., noisy, blurred, masked, or undersampled measurements), then distills it into a one-step generator using a score-matching objective that respects the corruption operator. The authors demonstrate state-of-the-art performance across a variety of corruption types (denoising, deblurring, inpainting, super-resolution, MRI reconstruction) on multiple datasets (CIFAR-10, CelebA-HQ, FFHQ, AFHQ-v2, FastMRI), consistently improving FID over teacher models while achieving up to 30× speedup in inference.

**Strengths:**

- **Originality:** Combines corruption-aware diffusion training with score distillation in a unified framework, with a focus on real-world scenarios where clean data is inaccessible.
- **Quality:** Extensive experiments across multiple tasks, datasets, and noise levels demonstrate robust and consistent improvements.
- **Clarity:** The paper is well-written, with clear explanations of both methodology and experimental design.
- **Significance:** Addresses a critical challenge in generative modeling for scientific and medical imaging, with practical speedups and performance gains.

**Weaknesses:**

- **Limited Comparison to GANs:** While diffusion-based methods are the focus, a comparison with GAN-based approaches trained on corrupted data could provide a broader perspective.
- **Complexity of Theoretical Analysis:** The theoretical section is dense and may be challenging for readers without a strong background in divergence metrics and Gaussian analysis.
- **Hyperparameter Sensitivity:** The performance of distillation losses (e.g., SiD vs. DMD) may depend on hyperparameter tuning, which was not fully explored.

**Questions:**

1. **Generalization to Non-Linear Operators:** Can DCD handle non-linear corruption operators (e.g., non-linear blur, sensor saturation)? If not, what are the limitations?
2. **Robustness to Mismatched Corruptions:** How does DCD perform if the corruption operator during distillation differs from that during pretraining?
3. **Scalability to Higher Resolutions:** Have you tested DCD on higher-resolution datasets (e.g., ImageNet, 256×256)? Are there any architectural or training adjustments needed?
4. **Comparison with GANs:** Have you considered comparing with GAN-based methods that also train on corrupted data?
5. **Real-World Deployment:** Are there any latency or memory constraints when deploying the one-step generator in real-time applications (e.g., MRI reconstruction)?

---

> ### Author Response · Authors · 2025-11-22
> **1/n rebuttal**
>
> [Highlight: A new Section 3.3 has been added in blue in the paper for a deeper and intuitive analysis on the mechanism of our method, titled “Why Score Distillation Beyond Acceleration?”]
>
> We sincerely thank the reviewer for your thoughtful and positive assessment of our work, especially regarding its originality in unifying corruption-aware diffusion training with score distillation, the robustness and breadth of our experimental results, the clarity of our presentation, and the practical significance for real-world scientific and medical imaging scenarios.
>
>
>
> > **Q1. Extension to non-linear operators**
>
>
>
> We evaluate DCD on well-established linear corruption tasks in order to **ensure a fair and direct comparison** with prior baselines such as Ambient Tweedie (suitable only for denoising) [1], SFBD (denoising) [3], Ambient Diffusion (inpainting only), and EM-Diffusion [2]. These methods serve as effective proxies for assessing the core contribution of our approach in a controlled and reproducible manner. Our experiments **cover all task types** used in these baselines, ensuring both fairness and completeness. Moreover, there is currently **no widely adopted benchmark for nonlinear corruption operators**, so we follow the standard linear settings used throughout the literature [1-3].
>
> [1] How much is a noisy image worth? data scaling laws for ambient diffusion. ICLR 2025.
>
> [2] An expectation-maximization algorithm for training clean diffusion models from corrupted observations. NeurIPS 2024.
>
> [3] Stochastic Forward-Backward Deconvolution: Training Diffusion Models with Finite Noisy Datasets. ICML 2025.
>
>
>
> Below, we clarify what would be required to extend DCD to nonlinear settings.
>
> 1. **Distillation stage:** The distillation stage can readily accommodate nonlinear operators, as it only requires corrupting samples in the form $\mathcal{A}(x) + \text{noise}$,  as shown in Line 6 of Algorithm 1.
> 2. **Pretraining stage:** While DCD can consistently improve upon weak teacher diffusion models without requiring a strong teacher, we believe that having reasonably pretrained diffusion models would result in more meaningful performance gains and a more useful learned prior. However, little prior work has explored how to train effective diffusion models directly from nonlinear observations, which often leads to poor teacher performance. We believe our work has the potential to further advance learning from nonlinear corruption, particularly if nonlinear corruption-aware diffusion training objectives are developed.
>
>
> We thank the reviewer for highlighting this more challenging nonlinear setting as an important direction for future research.
>
> > **Q2. Robustness to Mismatched Corruptions**
>
> In Table 8, we discussed the effect of mismatched noise level $\sigma$ **in both stages**. Our method DCD remains quite effective, e.g., surpassing the teacher by a large margin (FID 16.07 vs 42.99), even when the noise level is over-estimated.
> | σ̂ (Setting)           | Teacher-Full | Teacher-Trunc. | DCD |
> |------------------------|--------------|----------------|------|
> | **0.15 (under)**       | 80.60        | 49.78          | 103.55 |
> | **0.20 (exact)**        | 60.73        | 12.21      | **4.77** |
> | **0.25 (over)**        | 42.99        | 88.11          | **16.07** |
>
>
>
> We provide further experiments of mismatched noise level $\sigma$ **only in distilation stage** below:
> | Method          | FID  (↓) |
> |-----------------|------------|
> | Teacher         | 12.21  |
> | Underestimation $\sigma=0.15$  | 72.13      |
> | Exact $\sigma=0.2$ | **4.77** |
> | Overestimation $\sigma=0.25$ | **7.98** |
>
>
> **An FID evaluation plot has been added to the paper (Fig. 6 in the Appendix).** It shows that a *slight* over-estimation mildly degrades performance but still outperforms the teacher, demonstrating the robustness of our method.
>
> For **small mismatches**—such as slightly over-estimating the noise—we therefore expect DCD to remain robust: although performance may fall relative to the perfectly matched setting, the distilled generator still surpasses the weak teacher and successfully captures most of the structural information present in the corrupted observations.
>
> However, for **large mismatches**, DCD will perform poorly. Because DCD fundamentally relies on aligning the corrupted student score with the teacher score, a severely mis-specified corruption model disrupts this alignment. Intuitively and theoretically, the student then receives inconsistent training signals and fails to learn a meaningful prior.
>
>
> In addition, previous corruption-aware algorithms such as EM-Diffusion, Ambient Diffusion and Ambient Tweedie all require **exact** corruption operations. They are also expected to perform poorly under severe mismatches.

---

> ### Author Response · Authors · 2025-11-22
> **2/n rebuttal**
>
> > **Q3. Scalability to Higher Resolutions**
>
> We note that our multi-coil MRI experiments are conducted on brain images of size **320×380**, which is already larger than the standard ImageNet 256×256 resolution. Across all experiments, from delburring and super-resolution to Multi-coil MRI, we use the **original diffusion architectures without any modification** (e.g., EDM architecture and Muli-coil MRI diffusion models), and consistently observe performance improvements from our distillation method. We therefore expect similarly steady improvements on higher-resolution datasets. Furthermore, prior works on score distillation, such as DMD2 and SiD-LSG (experiments on ImageNet and Text-to-Image generation) indicate the potential good scalability of our method.
>
> DMD2: Improved Distribution Matching Distillation for Fast Image Synthesis
>
> SiD-LSG: Guided Score identity Distillation for Data-Free One-Step Text-to-Image Generation
>
> > **Q4. Comparison with GANs**
>
>
> 1. **[Additional experiments comparing with AmbientGAN]** We  compare against the most well-known GAN-based method for learning from corrupted data, AmbientGAN [1]. We follow the official codebase https://github.com/AshishBora/ambient-gan to implement the AmbientGAN on celebAHQ under the same setting as our paper's Table 5. DCD significantly outperforms AmbientGAN.
>
> | **Method**           | **p = 0.6** | **p = 0.8** |
> |----------------------|-------------|-------------|
> | **Observation**      | 275.04      | 383.82      |
> | **Ambient GAN**| 33.18        | 54.09      |
> | **Ambient Diffusion**| 6.08        | 11.19       |
> | **DCD**              | 4.44        | 7.10        |
>
>
>
>
> 2. **[Literature Review: DCD > AmbientDiffusion > AmbientGAN]** As shown in Table 5, our method surpasses Ambient Diffusion in the random masking setting across all missing rates $\\{0.6,0.8,0.9\\}$ by a large margin. Furthermore, Fig. 3 of Ambient Diffusion [2] demonstrates that Ambient Diffusion consistently outperforms AmbientGAN at all missing rates. Based on this established comparison in the literature, our results indicate that DCD also surpasses GAN-based methods.
>
>
> [1] AmbientGAN: Generative models from lossy measurements. ICLR 2018.
>
> [2] Ambient Diffusion: Learning Clean Distributions from Corrupted Data. NeurIPS 2023.
>
>
>
>
> > Q5. **Real-World Deployment: latency and memory**
>
> We thank the reviewer for highlighting the importance of real-world deployment. We believe DCD is particularly **well-suited** for such settings:
>
> 1. **Latency:** Because DCD yields a **one-step** generator, inference is dramatically faster than standard diffusion sampling. For example, on CIFAR-10 at $\sigma=0.2$, generating 50,000 samples decreases from 10 minutes using the teacher to 20 seconds with the distilled generator—an approximate **30× acceleration in wall-clock time**, as shown in Table 12 of the Appendix.
> 2. **Memory:** While training involves three models (teacher diffusion, fake diffusion, and student diffusion), **only the student generator is needed at deployment**. In our implementation, the student shares the exact same architecture as the teacher, so DCD delivers superior performance without increasing inference-time memory requirements.
>
>
> We additionally discuss **hardware considerations**, particularly for edge devices.
> For example, in the CIFAR-10 32×32 setting (batch size 1, single forward pass), both latency and memory usage fall well within the capabilities of modern edge hardware such as high-end smartphones. A UNet with roughly **225 MB** of parameters typically requires under **~0.5 GB of RAM** when accounting for intermediate activations, which fits comfortably within the **1–3 GB memory budget** that mobile operating systems commonly allocate to a single ML workload. The computational cost of one forward pass at this resolution is only on the order of a few to a few tens of GFLOPs, allowing inference to complete within a few to tens of milliseconds on recent NPUs/GPUs and enabling **real-time or near real-time performance**.
>
>
> **We sincerely hope our point-to-point responses address your concerns and help increase your confidence in our work, and we are willing to offer more clarification if needed.**

---

> > ### Comment · Reviewer_AX6j · 2025-11-26
> >
> > We thank the authors for their thorough and thoughtful responses to our questions. The additional experiments and clarifications provided—particularly the comparison with AmbientGAN, analysis of mismatched corruption settings, and discussion of scalability to higher resolutions—significantly strengthen the paper and address our initial concerns.
> >
> > We are especially pleased to see the new Section 3.3 (“Why Score Distillation Beyond Acceleration?”), which offers a more intuitive and accessible explanation of the distillation mechanism. This addition enhances the paper’s clarity and pedagogical value, making the core contribution more accessible to a broader audience.
> >
> > We also appreciate the authors’ candid discussion regarding nonlinear corruptions and the current limitations of corruption-aware diffusion training. Their clear articulation of the requirements for extending DCD to nonlinear settings provides a valuable roadmap for future work in this direction.
> >
> > Overall, we believe that DCD represents a meaningful and practical advance in generative modeling for corrupted-data regimes, with strong empirical results, a modular and flexible framework, and clear real-world applicability. The paper is well-executed, well-written, and makes a compelling case for the value of distillation beyond acceleration.
> >
> > We are happy to maintain our high rating and strongly support acceptance.

---

> > > ### Author Response · Authors · 2025-11-28
> > > **thank you!**
> > >
> > > Dear Reviewer,
> > >
> > > Thank you very much for your thoughtful and generous feedback. We sincerely appreciate the time and care you devoted to evaluating our work.
> > >
> > > We sincerely appreciate your recognition that DCD advances generative modeling in corrupted-data settings through strong empirical performance, a flexible modular framework, and clear practical relevance. We are equally grateful for your generous words, noting that the paper is well-executed, well-written, and makes a compelling case for distillation beyond acceleration.
> > >
> > > We hope our work brings fresh insights to diffusion distillation and delivers a practical, new tool for the corruption-learning community.
> > >
> > > Warm regards,
> > >
> > > Authors

---

### Official Review · Reviewer_4EL1 · 2025-11-01

**Soundness:** 3
**Presentation:** 3
**Contribution:** 2
**Rating:** 6
**Confidence:** 5

**Summary:**

The paper proposes a two-stage framework named DCD for learning a one-step generative model from corrupted observations. Stage one involves training a diffusion model teacher on the corrupted data using a suitable corruption-aware objective. Stage two distills this teacher into a one-step generator, where the key step is to apply the known corruption operator to the generator's output during the distillation loss computation. Extensive experiments show that the second stage not only makes the model fast but also significantly improves sample quality (FID) over the teacher model across various restoration tasks and MRI reconstruction.

**Strengths:**

1. The paper addresses an important practical problem. Learning from corrupted data is particularly useful in scientific areas like medical imaging. A notable contribution of this paper is that it shows that score distillation can improve generation quality from corrupted data in addition to accelerating sampling.

2. The method is empirically strong. The proposed method is tested on multiple datasets, diverse corruption types, and a real-world application. The gains over the teacher models are consistent and significant.

3. The two-phase design is practical to allow leveraging the rapidly growing literature on corruption-aware diffusion models for the teacher pretraining stage. This makes the framework adaptable.

**Weaknesses:**

1. The primary novelty of framework lies in specific composition of established techniques and the empirical validation that distillation is particularly effective in this setting. It leverages corruption-aware diffusion training and score distillation as constituent elements but is limited in inventing new components. The manuscript would be strengthened by more precise positioning of its contribution and clarifies its novelty and contribution in synthesizing existing methods to solve a new problem, rather than the introduction of fundamentally new techniques.

2. The central claim that distillation improves quality is empirical. A deeper and more intuitive analysis of the underlying mechanism is necessary to clarify the claim. In addition, the crucial question on the reason that enforcing self-consistency on a corrupted output space results in a more faithful representation of the clean data manifold remains unaddressed in the main text. Existing explanation for this phenomenon is cursory, and the core theoretical justification is deferred to the appendix.

**Questions:**

Please refer to the section of Weaknesses. I suggest that the authors provide a precise positioning of their contribution and a more intuitive analysis on the underlying mechanism to explain why distillation improves quality.

---

> ### Author Response · Authors · 2025-11-22
> **1/n rebuttal**
>
> We sincerely thank the reviewer for the constructive feedback on the writing of our paper, and for recognizing the practical importance of learning from corrupted data, the strong and consistent empirical performance across diverse settings, the adaptable two-phase design that leverages the rapidly growing literature on corruption-aware diffusion models, and our key insight that score distillation can improve generation quality beyond mere acceleration.
>
> > About the positioning of our contributions
>
> We thank the reveiwer for your constructive feedback on how to position our contributions more precisely. We restate our contributions as follows:
>
> - **A conceptual shift for the diffusion-distillation community:** Our results show, to our knowledge, a hidden benefit of score distillation. Whereas the original motivation of score distillation lied in accelerating sampling of generative models, we show in this work that distillation can also significantly improve generation quality when the teacher model is trained on corrupted data. **This is a new insight that we believe could inspire new research directions at the interface of corruption-aware learning and distillation.**
> - **A brand-new tool for the corruption-learning community:** This new insight also **provides a strong, competitive method in corruption-aware generative modeling.** In particular, DCD performs well across all tasks, with the following highlights:
>
>
>    - FID **4.77** learned purely from *noisy* CIFAR-10 (significantly better than current few-shot SOTA and close to DDPM’s 4.04).
>    - FID **4.44** on **60%-masked CelebA-HQ**.
>    - SOTA on **high-resolution (320×380) brain MRI**.
>
>
> We believe our tools will be widely adopted simply because they are elegant, concise, and deliver superior performance. If the reviewer agrees with the above restated contributions, we are happy to add this to the paper.
>
>
> ---
>
> **Then what’s the unique advantage of our method?**
>
> 1. **Superior performance that has never been achieved before with score distillation.**
>    As far as we can tell, this is the first time score-distillation-based models reach true SOTA in zero-shot learning from corruption *across all major tasks* in the literature:
>    - denoising vs. Ambient Tweedie (ICLR) + SFBD (ICML)
>    - inpainting vs. Ambient Diffusion (NeurIPS)
>    - multi-coil MRI vs. Ambient Diffusion Posterior Sampling (ICLR)
>    - deblurring / super-resolution vs. EM-Diffusion (NeurIPS)
>
>
>
> 2. **Flexibility.**
>    Unlike previous zero-shot methods that are basically one-task-only (SFBD only for denoising, Ambient Diffusion only for inpainting), our method is naturally flexible and can also be combined with these more specialized corruption-aware algorithms.
>
>      We would also like to emphasize that the flexibility of our framework is highly non-trivial, as we can adapt to corruption in image space, Fourier space, varying degrees of noise, and misspecification.
>
>
>
> 3. **Very fast inference (one step).**
>    An additional significant benefit of our approach is that we get both great image quality *and* a **30× speed-up** over standard diffusion.  As inspired by another reviewer, DCD is especially **well-suited** for **real-world deployment** because of its latency, memory benefits (please see our response to reviewer AX6j), and successful application to the Multi-Coil MRI problem (noted by reviewer 5ZSz).

---

> ### Author Response · Authors · 2025-11-22
> **2/n**
>
> > Deeper and more intuitive analysis of the underlying mechanism: Why Score Distillation Beyond Acceleration?
>
>
> We updated our paper with a new section 3.3, titled “Why Score Distillation Beyond Acceleration?”. We provide the same text for your convenience below:
>
> ### Why Score Distillation Beyond Acceleration?
>
> Score distillation is traditionally viewed as a means of *accelerating* sampling by compressing a multi-step diffusion process into a single forward pass. However, our paper reveals a more fundamental **conceptual shift**:  in corrupted or low-quality data regimes, score distillation acts as a *principled mechanism* for improving the sample quality of teacher diffusion models.
>
> ---
>
> #### **A Toy Example**
> In Fig. 2, the observation is a noisy spiral dataset with $\sigma$ = 0.05.  Teacher diffusion models on noisy datasets—(b) Teacher-Full and (c) Teacher-Truncated—must spread their probability mass across all regions supported by the corrupted data, making the learned density overly diffuse. In contrast, (d) DCD excels at denoising the original dataset, producing a narrow, concentrated, and sharp approximation.  This improvement occurs *without* ever providing the clean data during training.
>
>
>
> #### **Intuitive Understanding**
> Why does this sharpening occur? Diffusion models are trained using the ELBO objective for maximum likelihood, which minimizes the **forward KL divergence**—a *mode-covering* objective. Thus, teacher diffusions allocate probability mass across *all* plausible regions indicated by the noisy observations, even those that do not correspond to the true clean data distribution.
>
> In contrast, the **reverse distillation objective** (Eq. 5) is taken over the *student generator’s* distribution, naturally inducing **mode-seeking** behavior: the generator only needs to place mass where it actually samples. Importantly,  **encouraging mode seeking does not imply mode collapse.**  (FID and Recall in Tab. 11 as well as multiple qualitative examples in the Appendix support this.) This shift allows the distilled generator to concentrate its probability mass on the **high-density, well-supported regions** implied by the teacher’s score field, while discarding overly diffuse or low-density regions that teacher diffusion models include.
>
>
>
> #### **Theoretical Support**
> We provide detailed analysis in the appendix:
>
> - Section C.1 derives bounds on distilled generator performance under standard capacity and optimization assumptions.
> - Section C.2 specializes to a linear Gaussian setting, yielding explicit optimization landscapes, quantitative error bounds, and closed-form expressions for global minimizers (Eq. 10).
> - Section C.3 extends the theory to general linear corruption settings.
>
> ---
>
>
> We hope our explanation could address your concerns and increase your confidence in our work, and we are willing to offer more clarification if needed.

---

### Official Review · Reviewer_ZwLf · 2025-11-01

**Soundness:** 2
**Presentation:** 3
**Contribution:** 2
**Rating:** 4
**Confidence:** 4

**Summary:**

This paper proposes a framework named Distillation from Corrupted Data (DCD) for learning high-fidelity, one-step generative models from corrupted data. The framework is implemented in two stages. A diffusion model is first pretrained on the corrupted data as a teacher model, and the knowledge of the teacher model is then distilled into an efficient one-step generator. The contribution of this manuscript lies in applying score distillation technology to learning generative models from corrupted data, providing a new solution for the relevant field.

**Strengths:**

1. The paper introduces DCD, a novel framework for learning generative models from corrupted data without needing clean data. This approach is original in its unified treatment of diverse corruptions.

2. The research is of high quality, with a rigorous two-phase framework combining corruption-aware diffusion pretraining and score distillation. Theoretical analysis supports the method, and comprehensive experiments validate its effectiveness across various datasets and tasks.

3. The proposed DCD framework is potential to improve generative modeling in scenarios with limited clean data.

**Weaknesses:**

1. The DCD framework extends score distillation to corrupted data but lacks novel methodological contributions. Score distillation has already been widely used for learning generative models from clean data, and is extended to the scenario of corrupted data in this paper. However, it seems that DCD relies on existing diffusion models and score distillation techniques and offers limited innovations and does not provide unique paradigm for learning from corrupted data.

2. In the distillation phase, the authors used an auxiliary fake diffusion model to approximate the distribution induced by the generator. The use of an auxiliary fake diffusion model during distillation introduces potential approximation errors. This method may not accurately capture the distribution induced by the generator and affect the overall performance of the framework.

3. Experimental results are not sufficient to validate the effectiveness of DCD.

i) Evaluations are performed on small, single-type datasets like CIFAR-10 and CelebA-HQ. The results on these small, homogeneous datasets cannot demonstrate the generalization of DCD on different image sizes, diverse classes, or various corruption types/levels/schedules.

ii) In most experiments, FID is adopted as the main evaluation metric. There lacks of a comprehensive assessment of the performance of DCD under varying metrics.

iii) Ablation studies on the fake diffusion surrogate, shared initialization, and training stability are missing. Both the fake diffusion model and the generator are initialized from the teacher model. Since the teacher model itself may learn incorrect features, this could cause staying in local optima during training.

4. In the mathematical derivation of score distillation, the authors assume that the score fields between the teacher model and the generator could be aligned, but do not explain the rationality of the assumption or provide supporting evidence.

5. Minor issue. The names of the horizontal and vertical coordinates are missing in Figure 12.

**Questions:**

1. What are the innovative points of the DCD framework in terms of methodology and how do these innovative points give it a unique advantage in learning generative models from corrupted data?

2. What is the additional error introduced by approximation with auxiliary fake diffusion model?

3. How do the results change with resolution and class diversity?

4. What is the performance of DCD under more metrics like KID, Precision/Recall, and Density/Coverage?

5. Could the authors provide ablation studies on the fake diffusion surrogate, shared initialization, and training stability?

6. What is the rationality of the assumption that the score fields between the teacher model and the generator could be aligned?

---

> ### Author Response · Authors · 2025-11-22
> **1/n rebuttal**
>
> [Highlight: A new Section 3.3 has been added in blue in the paper for a deeper and intuitive analysis on the mechanism of our method, titled “Why Score Distillation Beyond Acceleration?”]
>
> We provide one-to-one responses as follows. References to prior works are at the end.
>
> > 1. Novelty in terms of methodology and unique advantages
>
> We thank the reviewer for raising this point. What defines the novelty of a paper? It depends. Sometimes it’s a new algorithm that solves a task extremely well. Other times it’s a brand-new *perspective* — the algorithm itself may be simple or even obvious in hindsight, but it lets you do something people simply never thought about before.
>
> We argue that our paper has **both kinds of novelty** as follows.
>
> - **A conceptual shift for the diffusion-distillation community:**  Our results show, to our knowledge, a hidden benefit of score distillation. Whereas the original motivation of score distillation lied in accelerating sampling of generative models, we show in this work that distillation can also significantly improve generation quality when the teacher model is trained on corrupted data. **This is a new insight that we believe could inspire new research directions at the interface of corruption-aware learning and distillation.**
> - **A brand-new tool for the corruption-learning community:** This new insight also **provides a strong, competitive method in corruption-aware generative modeling.** In particular, DCD performs well across all tasks, with the following highlights:
>
>
>    - FID **4.77** learned purely from *noisy* CIFAR-10 (significantly better than current few-shot SOTA and close to DDPM’s 4.04).
>    - FID **4.44** on **60%-masked CelebA-HQ**.
>    - SOTA on **high-resolution (320×380) brain MRI**.
>
>    We believe our tools will be widely adopted simply because they are elegant, concise, and deliver superior performance.
>
>
> ---
>
> **Then what’s the unique advantage of our method?**
>
> 1. **Superior performance that has never been achieved before with score distillation.**
>    As far as we can tell, this is the first time score-distillation-based models reach true SOTA in zero-shot learning from corruption *across all major tasks* in the literature:
>    - denoising vs. Ambient Tweedie (ICLR) + SFBD (ICML)
>    - inpainting vs. Ambient Diffusion (NeurIPS)
>    - multi-coil MRI vs. Ambient Diffusion Posterior Sampling (ICLR)
>    - deblurring / super-resolution vs. EM-Diffusion (NeurIPS)
>
>
>
> 2. **Flexibility.**
>    Unlike previous zero-shot methods that are basically one-task-only (SFBD only for denoising, Ambient Diffusion only for inpainting), our method is naturally flexible and can also be combined with these more specialized corruption-aware algorithms.
>
>      We would also like to emphasize that the flexibility of our framework is highly non-trivial, as we can adapt to corruption in image space, Fourier space, varying degrees of noise, and misspecification.
>
>
>
> 3. **Very fast inference (one step).**
>    An additional significant benefit of our approach is that we get both great image quality *and* a **30× speed-up** over standard diffusion.
>    As inspired by another reviewer, DCD is especially **well-suited** for **real-world deployment** because of its latency, memory benefits (please see our response to reviewer AX6j), and successful application to the Multi-Coil MRI problem (noted by reviewer 5ZSz).

---

> ### Author Response · Authors · 2025-11-22
> **2/n rebuttal**
>
> > 2. What is the additional error introduced by approximation with auxiliary fake diffusion model?
>
>
>
>  We believe the reviewer has **a misunderstanding** here. We appreciate the reviewer's concern about the possibility of model errors when introducing the fake diffusion model. The use of a fake diffusion model, however, **has been shown to bring great benefits in score distillation** as compared to not using a fake diffusion model. This has been shown in prior works (DMD, SiD, and Diff-Instruct). For example, the DMD paper shows that without the distribution matching loss (i.e., the alignment between the fake and teacher diffusions at the distributional level), the distilled model performs worse (see their Table 2, Ablation Study: “Without dist. matching”).
>
> We have also shown in our paper that direct distillation without a surrogate (e.g., SDS) performs poorly and does not converge. This trend is consistent across all corruption levels, as shown below (Table 14 in the paper). We add **ours without fake diffusion** experiments and also present the **distillation curve in Fig. 3 of our updated pdf**:
>
>
> **Table. CIFAR-10 denoising**
>
> | **Method**                   | **$\sigma$ = 0.1**     | **$\sigma$ = 0.2**     | **$\sigma$ = 0.4**     |
> |-----------------------------|-----------------|-----------------|-----------------|
> | SDS (no fake diffusion)     | >200            | >200            | >200            |
> | DMD (with fake diffusion, different distillation objective)  | 12.52 ± 0.04    | 7.48 ± 0.06     | 30.09 ± 0.23    |
> |Ours (without fake diffusion) | >200 | >200| >200|
> | **Ours**   (with fake diffuion)           | **3.98 ± 0.04** | **4.77 ± 0.03** | **21.63 ± 0.03**|
>
>
> These results clearly demonstrate that:
> - **Without** the fake diffusion surrogate (SDS+our without fake diffusion), optimization collapses in our corruption-learning settings.
> - **Our fake diffusion surrogate + our objective** yields the best performance across all corruption levels.
>
> Given the good performance of our current framework with fake diffusion, we consider the reviewer **is proposing a possible way to improve the performance of our framework by doing more gradient update of fake diffusion**.
> Our framework alternates updates between the generator and the fake diffusion model (1 step each), following standard practice in score distillation (e.g., DMD, SID). We experimented with performing **more** fake-diffusion updates per generator step and found:
>
> - **no improvement** in sample quality,
> - **no improvement** in stability or convergence, and
> - **higher** computational cost at each iteration.
>
> Therefore, our current framework with 1:1 update schedule is both **efficient and sufficient**, delivering strong performance at minimal compute cost.
>
>
>
> > 3. resolution and class diversity
>
>
> We understand the reviewer's concern on the diversity of our experimental datasets. We believe our chosen datasets suffice for the following reasons:
>
> 1) **Size and class diversity:** **Note that the fastMRI images we use are of resolution 320x380.** In total, we use 5 datasets (CIFAR10, CelebAHQ, FFHQ, AFHQv2, and fastMRI datasets) and analyze a wide variety of corruption operators (additive Gaussian noise, masking, blur, downsampling, and subsampled Fourier measurements). The images also vary in size, ranging from 32x32, 64x64, to 320x380. Some datasets, such as CIFAR10, have many classes. We also achieve superior performance to prior works across all 5 datasets.
>
> 2) **Similar resolution and diversity to prior work:** We covered all the datasets used in EM-Diffusion (CIFAR10, CelebAHQ), Ambient Diffusion (CelebAHQ, AFHQ, CIFAR10, brain images), Ambient Diffusion DPS (fastMRI), and SFBD (CIFAR10, CelebAHQ).
>
>
>  We believe that the **combination of numerous datasets ranging from natural to scientific images, diversity of corruption operators, to similar, if not more, diversity to prior works** suffices to showcase the generality of our approach.
>
>  We are happy to discuss more about what can be satisfying if the reviewer still has concerns about this point.

---

> ### Author Response · Authors · 2025-11-22
> **3/n rebuttal**
>
> > 4. more metrics: precision/recall/IS/KID
>
> We understand the reviewer's concern on we relying on FID to measure the performance. FID is useful, however, to not only measure sample quality but also sample coverage. We point out that well-known diffusion papers such as DDIM (ICLR), DMD (CVPR), and our baselines SFDB (ICML), Ambient Tweedie (ICLR), and EM-Diffusion (NeurIPS) only report FID for sample quality. We follow this trend and report FID for fair comparison. To address the reviewer's concern, we have also included additional metrics, such as precision, recall, IS, and KID in the following table:
>
>
>
>
>  *\*Results copied from Rectified flow (ICLR, https://arxiv.org/pdf/2209.03003) paper's Table 1.*
>
> | Method                       | σ    | FID ↓  | IS ↑   | Precision ↑ | Recall ↑ | KID ↓      |
> |------------------------------|------|--------|--------|--------------|-----------|------------|
> | **Teacher (σ = 0.2)**        | 0.2  | 12.21  | 8.312  | 0.595        | 0.416     | 0.00593    |
> | **Ours (σ = 0.4)**           | 0.2  | 21.63  | 7.934  | 0.536        | 0.384     | 0.01270    |
> | **Ours (σ = 0.2)**           | 0.2  | 4.77   | 9.165  | 0.650        | 0.564     | 0.00252    |
> | **Ours (σ = 0.1)**           | 0.2  | 3.98   | 9.346  | 0.643        | **0.578** | 0.00157    |
> | **DDPM***                    | 0.0  | 3.21   | 9.46   | N/A          | 0.57      | N/A        |
> | **Rectified Flow (ODE)***    | 0.0  | 2.58   | 9.60   | N/A          | 0.57      | N/A        |
>
>
>
> DDPM and Rectified flow are trained on clean data ($\sigma=0$), while our method only sees corrupted data during training ($\sigma=0.2$). **DCD can achieve almost the same good sample quality as DDPM and Rectified Flow in terms of all metrics.**
>
>
> The results show that DCD can achieve both sample quality (FID) and mode coverage (Recall).
>
>
> > 5. Ablation on fake diffusion, shared initilization: local minima
>
> We believe the reviewer has  **another misunderstanding** on shared initialization.
>  We appreciate the reviewer's concern regarding convergence to local minima. **We note that both the diffusion surrogate and shared initialization are standard practice in the score distillation literature** (as is done, e.g., in DMD, SiD, Diff-Instruct). A significant reason for this is that both practices have been observed to improve performance of distillation.
>
> To showcase this, we have added a new plot (Fig. 3) in the main text showing the distillation curves for:*
> (1) our full method,
> (2) removing shared initialization, and
> (3) removing the fake-diffusion surrogate.
>
> Removing shared initialization leads to a significant degradation in performance (FID ≈ 250), while removing the fake diffusion surrogate causes the training to become unstable and fail to converge.
>
>
>
> We also provide code references demonstrating the use of shared initialization + the use of fake diffusion:
>
> - SiD code https://github.com/mingyuanzhou/SiD/blob/01fbb13cf18411e5f70d187c8fb2559567e5ae1c/training/sid_training_loop.py#L217
> - Diff Instruct code
> https://github.com/pkulwj1994/diff_instruct/blob/0c37389fbfb1f5fc38708dec57d0dae257726460/training/di_training_loop.py#L185
> - DMD: They didn't release the code, but they specified shared initialization clearly in the Algo. 1 (Line 1) of their paper.
>
> **As indicated by the Recall + FID metrics above, as well as multiple qualitative examples at the end of the appendix, our method not only achieves good performance but also does not suffer from potential mode collapse issues.**

---

> ### Author Response · Authors · 2025-11-22
> **4/n rebuttal**
>
> > 6. theory explanation: how the scores of teacher and student align with each other
>
> We appreciate the reviewer’s question regarding the rationale behind aligning the scores of the teacher and student models. As the reviewer might not be fully familiar with score distillation, **the motivation for score alignment arises from standard theoretical foundations of score distillation.** To clarify this intuition, we provide a brief proof sketch below. Our objective is to minimize the divergence between the fake diffusion ($\phi$) and the teacher diffusion ($\psi$, frozen):
> $$
> \min_{\phi} \mathbb{E}_{t}\big[\mathcal{D}\big(p\_{\phi,t}(x_t) || p\_{\psi,t}(x_t)\big)\big].
> $$
> As proved in **Theorem 2 of Diffusion-GAN (ICLR 2022, https://arxiv.org/pdf/2206.02262)**, consider the setting
> $y = f(x) + h(\epsilon)$ (which corresponds to $y = \alpha_t x + \beta_t \epsilon$ in diffusion models) and $y_g = f(x_g) + h(\epsilon)$. If the distributions of $y$ and $y_g$ are the *same*, then the distributions of the underlying clean variables $x$ and $x_g$ are also the *same*.
>
> In other words: **if the noised scores match, then the clean scores match as well.**
> This is precisely the justification behind aligning noised distributions via score distillation.
>
>
> We are willing to provide more clarification if needed.
>
> **We have also added a new section 3.3 in the paper for a deeper and more intuitive analysis of the underlying mechanism of our method,  titled 'why score distillation beyond acceleration?'.**
>
>
> ---
>
> References:
>
> 1. DMD: One-step Diffusion with Distribution Matching Distillation. CVPR.
>
> 2. SiD: Score identity Distillation: Exponentially Fast Distillation of Pretrained Diffusion Models for One-Step Generation.  ICML.
>
>
> 5. EM-Diffusion: An Expectation-Maximization Algorithm for Training Clean Diffusion Models from Corrupted Observations. NeurIPS.
>
> 6. Ambient Diffusion: Ambient Diffusion: Learning Clean Distributions from Corrupted Data. NeurIPS.
>
> 7. Ambient Tweedie: How much is a noisy image worth? Data Scaling Laws for Ambient Diffusion. ICLR.
>
> 8. Diff-Instruct: A Universal Approach for Transferring Knowledge From Pre-trained Diffusion Models. NeurIPS.
>
> 9. SFBD: Stochastic Forward-Backward Deconvolution: Training Diffusion Models with Finite Noisy Datasets. ICML 2025.
>
> 10. Ambient Diffusion Posterior Sampling: Solving Inverse Problems with Diffusion Models Trained on Corrupted Data. ICLR 2025.
>
> Last but not least, the names of horizontal and vertical coordinates of Fig. 12 that the reviewer is referring to (now Fig 15) are simply x and y.  Hope this is not confusing.
>
> **We believe that our new tables, figures and clarification greatly address the reviewer’s concerns and help situate our work within the broader progress on learning from corrupted data and score distillation. If so, we would be grateful if the reviewer would consider increasing their score.**

---

### Author Response · Authors · 2025-11-27
**Dear Reviewers**

Dear Reviewers,

We would appreciate any feedback on our rebuttal, if there is any.

Best,
Authors

---

### Author Response · Authors · 2025-11-28
**Summary of our rebuttal**

Dear PCs, SACs, ACs, and Reviewers,

We sincerely appreciate the constructive and thoughtful feedback provided during the review and rebuttal process. In light of the main concerns raised, and given the limited remaining discussion window (**only four days left**), we have made a set of focused revisions to strengthen the paper. All changes in the revised submission are highlighted in blue.

The main updates are as follows:

- **Section 3.3:** Expanded the discussion *“Why score distillation beyond acceleration?”* with additional intuitive analysis and illustrative toy examples.
- **Figure 5:** Added an ablation on generation initialization and fake diffusion.

In addition, we have included further experiments and reported more metrics (**Precision, Recall, and KID**) in response to **Reviewer ZwLf**’s requests.

We hope these revisions directly address the main concerns raised by the reviewers and help clarify both the scope and the contributions of our work. We are, of course, happy to continue the discussion and further clarify any remaining issues.

Best regards,
Authors

---

### Meta-Review · Area_Chair_bdZi · 2026-01-05

**Summary:**

**Paper Summary:** This paper proposes a framework for learning generative models from corrupted observations. The framework consists of two training stages. In the first stage, a corruption-aware diffusion model is trained as a teacher. In the second stage, the teacher is distilled into a one-step generator using an auxiliary fake diffusion model. The authors evaluate the method on several image generation tasks and demonstrate superior performance with significantly faster image synthesis. Notably, the results show that score distillation not only accelerates generation but also improves performance under corrupted data.

**Concerns:** Reviewers raised concerns regarding the clarity of contributions, the justification for introducing the auxiliary fake diffusion model, the need for more extensive experimental analysis, the rationality of the underlying assumptions, the lack of deeper and more intuitive analysis, comparisons with GAN-based methods, and sensitivity to hyperparameters.

**Author-Reviewer Discussion:** After discussion, the AC finds that most of the reviewers’ concerns have been adequately addressed. However, the proposed theoretical analysis still lacks experimental validation to confirm the assumptions and conclusions, and it does not fully align with the intuitive interpretation in terms of mode-seeking KL divergence.

**Recommendation:** Despite the remaining concern, the AC acknowledges the notable finding that score distillation can both accelerate generation and improve performance, as well as the simplicity and effectiveness of the proposed training strategies. Therefore, the AC recommends **acceptance** of this paper and encourages the authors to further address the remaining concern in the revision.

**Reviewer Concerns:**

Reviewers raised concerns about the clarity of the contributions, the justification for the auxiliary fake diffusion model, the need for more comprehensive experiments, the validity of the underlying assumptions, the depth and intuitiveness of the analysis, comparisons with GAN-based methods, and sensitivity to hyperparameters.

After discussion, the AC finds that most concerns have been sufficiently addressed. However, the theoretical analysis still lacks experimental validation of its assumptions and conclusions, and its connection to the intuitive interpretation of mode-seeking KL divergence remains unclear.

**Reviewer Scores:**

Given that most concerns raised by the reviewers have been resolved, the expected post-rebuttal scores are: 6, 6, 8, and 6.

---

### Decision · Program_Chairs · 2026-01-26

Accept (Poster)